# Out-of-Equilibrium Full-Counting Statistics in Gaussian Theories of Quantum Magnets

Riccardo Senese[1*], Jacob H. Robertson[1] and Fabian H.L. Essler[1]

**1** The Rudolf Peierls Centre for Theoretical Physics, Oxford University, Oxford OX1 3NP, UK
*riccardo.senese@physics.ox.ac.uk

January 4, 2024

## Abstract

We consider the probability distributions of the subsystem (staggered) magnetization in ordered and disordered models of quantum magnets in D dimensions. We focus on Heisenberg antiferromagnets and long-range transverse-field Ising models as particular examples. By employing a range of self-consistent time-dependent mean-field approximations in conjunction with Holstein-Primakoff, Dyson-Maleev, Schwinger boson and modified spin-wave theory representations we obtain results in thermal equilibrium as well as during non-equilibrium evolution after quantum quenches. To extract probability distributions we derive a simple formula for the characteristic function of generic quadratic observables in any Gaussian theory of bosons.

## 1 Introduction

A fundamental tenet of quantum theory is that measuring an observable $\mathcal{O}$ in a given state $|\Psi\rangle$ generally leads to different measurement outcomes that are described by a probability distribution $P(\mathcal{O}, |\Psi\rangle)$. In the context of many-particle systems quantum mechanical probability distribution functions (PDFs) known as "Full Counting Statistics" (FCS) have had important applications in mesoscopic devices [1, 2], where shot noise experiments can determine the charge and statistics of the quasiparticles relevant for transport. More recently they became accessible in ultra-cold atomic gases [3–9], where it is possible to measure the PDFs of various observables defined on subsystems. These experiments in turn motivated theoretical efforts to compute PDFs both in [10–23] and out of [7,24–31] equilibrium, as well as in non-equilibrium steady states, see e.g. [32–34] and references therein. From a theoretical point of view calculating the FCS for a given subsystem observable in an interacting many-particle system poses a formidable problem. Consequently there are relatively few known results, in particular in two and three spatial dimensions. Here we consider PDFs of subsystem observables in models of quantum magnets, focusing mainly on the case $D > 1$. We employ a range of representations of quantum spins in terms of canonical boson operators and focus on physical regimes dominated by Gaussian fluctuations of these bosons. This allows us to obtain efficient determinant representations for the characteristic functions of the PDFs of interest, both in and out of equilibrium.

The paper is organized as follows. In Section 1.1 we introduce probability distribution functions (PDFs) of quantum observables and their characteristic functions in many-body quantum systems. In Section 1.2 we report a simple formula for the characteristic function of any observable expressible as at most a quadratic polynomial in bosonic variables, in any Gaussian state of said bosons. A derivation of the formula is presented in Appendix C.

We then turn to applications of this formula to extract PDFs in models of quantum magnets both in and out of equilibrium. In Section 2 we focus on systems possessing long-range

magnetic order and employ spin-wave theory to reduce the problem to the study of Gaussian theories of bosons. In Section 2.1 we discuss the 2D and 3D Heisenberg antiferromagnet and derive the PDF of the subsystem staggered magnetization in thermal equilibrium and during time evolutions after quantum quenches. In Section 2.2 we study the out-of-equilibrium dynamics of the long-range transverse field Ising chain (LRTFIC) in cases where the direction of magnetic order doesn't significantly change over time. To overcome this limitation, we review and extend in Section 2.3 a recently proposed method to implement spin-wave theory around a time-dependent direction, in a large class of models. This allows us to consider a more general class of quantum quenches in the LRTFIC, for which we derive the out-of-equilibrium PDF of the subsystem magnetization.

In Section 3 we consider disordered phases of magnets by studying the 2D Heisenberg antiferromagnet at low but finite temperatures and its dynamics after quantum quenches. In Section 3.1 we employ a Schwinger boson mean-field theory, which however turns out to be a poor approximation. We improve on this in Section 3.2 by means of a modified spin-wave theory originally proposed by Takahashi, which we genaralize to the out-of-equilibrium case.

Section 4 contains a summary of the results and our conclusions. Four appendices follow discussing details of the approximations employed and some mathematical derivations.

## 1.1 Characteristic function and probability distribution

Consider a generic system defined on a discrete lattice of dimension $d$ and focus on a subsystem $\mathcal{A}$ with total number of sites $|\mathcal{A}| = \ell$. Given any Hermitian operator $R_\mathcal{A}$ with support only in $\mathcal{A}$, we want to determine the probability distribution function (PDF) $P_\mathcal{A}(r)$ of $R_\mathcal{A}$ in a state of the system characterized by the reduced density matrix (RDM) $\rho_\mathcal{A}$

$$P_\mathcal{A}(r) = \text{Tr}_\mathcal{A}\left[\rho_\mathcal{A}\delta(r - R_\mathcal{A})\right] \qquad\qquad \rho_\mathcal{A} = \text{Tr}_{\overline{\mathcal{A}}}[\rho] \,, \tag{1}$$

where $\overline{\mathcal{A}}$ is the complement of $\mathcal{A}$. $P_\mathcal{A}(r)$ can be rewritten in terms of the characteristic function $\chi_\mathcal{A}(\lambda)$ [16, 27]

$$\chi_\mathcal{A}(\lambda) \equiv \text{Tr}_\mathcal{A}\left[\rho_\mathcal{A}\exp(i\lambda R_\mathcal{A})\right] \qquad\qquad P_\mathcal{A}(r) = \int_{-\infty}^{\infty}\frac{d\lambda}{2\pi}e^{-i\lambda r}\chi_\mathcal{A}(\lambda)\,, \tag{2}$$

for which the properties $\chi_\mathcal{A}(-\lambda) = \chi_\mathcal{A}^*(\lambda)$ and $|\chi_\mathcal{A}(\lambda)| \leq 1$ hold. The latter follows by resolving the trace in (2) over the eigenvectors of $\rho_\mathcal{A}$ and applying Schwarz inequality to bound the contribution of the unitary operator $\exp(i\lambda R_\mathcal{A})$.

We note that for the very specific class of operators $R_\mathcal{A}$ having discrete spectrum composed only of equispaced eigenvalues

$$R_\mathcal{A}|m\rangle = r_m|m\rangle \qquad\qquad r_m = r_0 + mR \qquad\qquad r_0, R \in \mathbb{R}, \quad m \in \mathbb{Z}\,, \tag{3}$$

the second formula in Eq. (2) is greatly simplified thanks to the periodicity

$$\chi_\mathcal{A}(\lambda + 2\pi n/R) = e^{i2\pi n r_0/R}\chi_\mathcal{A}(\lambda) \qquad\qquad n \in \mathbb{Z}\,. \tag{4}$$

Indeed, using (4), we can rewrite $P_\mathcal{A}(r)$ as

$$P_\mathcal{A}(r) = \sum_{n\in\mathbb{Z}}\int_{-\pi/R}^{\pi/R}\frac{d\lambda}{2\pi}e^{-i(\lambda + 2\pi n/R)r}\chi_\mathcal{A}(\lambda)e^{i2\pi n r_0/R} = \tilde{P}_\mathcal{A}(r)\sum_{m\in\mathbb{Z}}\delta(r - r_0 - mR) \tag{5}$$

where

$$\tilde{P}_\mathcal{A}(r) \equiv \frac{R}{2\pi}\int_{-\pi/R}^{\pi/R}d\lambda\,e^{-i\lambda r}\chi_\mathcal{A}(\lambda) = \frac{R}{\pi}\text{Re}\left[\int_{0}^{\pi/R}d\lambda\,e^{-i\lambda r}\chi_\mathcal{A}(\lambda)\right]\,, \tag{6}$$

is normalized to sum to 1 when viewed as a discrete PDF over the eigenvalues (3). The finite interval of integration in $\lambda$ is the main advantage of (6) over the original (2) when numerical integrations are required.

## 1.2 Full-counting statistics in Gaussian theories of bosons

We now present our key result for computing the characteristic function of a quantum observable expressed as at most a quadratic polynomial in bosonic variables, when the system state can be expressed as a Gaussian function of said variables. Concretely, let $\{a_i\}_{i=1}^{i=\ell}$ be canonical annihilation operators for $\ell$ bosonic harmonic oscillators. These can be assembled into a $2\ell$-component vector with commutation relations given by

$$\boldsymbol{a}^\dagger = \{a_1^\dagger, \ldots, a_\ell^\dagger, a_1, \ldots, a_\ell\} \,, \qquad \left[\boldsymbol{a}_i, \boldsymbol{a}_j^\dagger\right] = \Sigma_{ij}^z \,. \tag{7}$$

Here we have defined

$$\Sigma^x = \begin{pmatrix} 0 & \mathbb{I}_{\ell\times\ell} \\ \mathbb{I}_{\ell\times\ell} & 0 \end{pmatrix} \,, \qquad \Sigma^y = i \begin{pmatrix} 0 & -\mathbb{I}_{\ell\times\ell} \\ \mathbb{I}_{\ell\times\ell} & 0 \end{pmatrix} \,, \qquad \Sigma^z = \begin{pmatrix} \mathbb{I}_{\ell\times\ell} & 0 \\ 0 & -\mathbb{I}_{\ell\times\ell} \end{pmatrix} \,. \tag{8}$$

Let $\rho$ denote the bosonic Gaussian state

$$\rho = \frac{1}{Z} \exp\left[\frac{1}{2}\boldsymbol{a}^\dagger W \boldsymbol{a} + \boldsymbol{w}^\dagger \cdot \boldsymbol{a}\right] \,, \tag{9}$$

where $W$ is an Hermitian $2\ell \times 2\ell$ negative-definite matrix and $\boldsymbol{w}$ a vector of length $2\ell$ such that $\boldsymbol{w}^\dagger = \boldsymbol{w}^T \Sigma^x$. As a consequence of Wick's theorem the state $\rho$ is fully characterised by the one and two-point functions of bosons

$$\boldsymbol{\omega} = \text{Tr}[\rho \boldsymbol{a}] \,, \qquad \Delta = \text{Tr}\left[\rho \left(\boldsymbol{a} - \boldsymbol{\omega}\right)\left(\boldsymbol{a}^\dagger - \boldsymbol{\omega}^\dagger\right)\right] - \frac{1}{2}\Sigma^z \,. \tag{10}$$

Our main result can be stated as follows.

**Result:** Consider a Hermitian operator of the form

$$R = \frac{1}{2}\boldsymbol{a}^\dagger G \boldsymbol{a} + \boldsymbol{g}^\dagger \cdot \boldsymbol{a} \qquad\qquad \text{Det}(G) \neq 0 \qquad\qquad \Sigma^x G \Sigma^x = G^T. \tag{11}$$

The last relation can always be imposed, without loss of generality, given the definition of $\boldsymbol{a}$ in (7). The characteristic function of the associated quantum mechanical probability distribution for the operator $R$ in the state $\rho$ can be represented in terms of the one- and two-point functions as

$$\chi(\lambda) = \text{Tr}\left[\rho\, e^{i\lambda R}\right] = Z_G \frac{\exp\left[-\frac{1}{2}\left(\boldsymbol{\omega}^\dagger - \boldsymbol{\omega}_G^\dagger\right)\left(\Delta + \Delta_G\right)^{-1}\left(\boldsymbol{\omega} - \boldsymbol{\omega}_G\right)\right]}{\sqrt{\text{Det}(\Delta + \Delta_G)}} \,, \tag{12}$$

where

$$\boldsymbol{\omega}_G = -G^{-1}\boldsymbol{g} \,, \qquad \Delta_G(\lambda) = -\frac{1}{2}\coth\left(i\lambda\frac{1}{2}\Sigma^z G\right)\Sigma^z \,,$$

$$Z_G(\lambda) = \exp\left[-i\lambda\frac{1}{2}\boldsymbol{g}^\dagger G^{-1}\boldsymbol{g}\right]\text{Det}\left[2\,\Sigma^z\sinh\left(-i\lambda\frac{1}{2}\Sigma^z G\right)\right]^{-1/2} \,. \tag{13}$$

A derivation of (12) based on coherent states methods is presented in Appendix C. There we also discuss how the formula is modified in presence of a singular $G$. In Appendix D we briefly mention how the formula can be obtained by purely algebraic methods.

## 2 Magnetically ordered systems

### 2.1 2D and 3D Heisenberg antiferromagnet

The $d$-dimensional antiferromagnetic Heisenberg model is described by the $SU(2)$ invariant Hamiltonian

$$H = J \sum_{\langle i,j \rangle} \mathbf{S}_i \cdot \mathbf{S}_j \qquad J > 0 , \tag{14}$$

where $S_i^\gamma$ are spin-$s$ operators respecting the $SU(2)$ algebra $\left[ S_i^\gamma, S_j^\rho \right] = i\delta_{ij}\epsilon_{\gamma\rho\omega}S_i^\omega$ on a $d$-dimensional hypercubic lattice with $N = L^d$ sites, the interactions are limited to neighbouring sites and periodic boundary conditions (PBC) are assumed. The ground-state of this model is a non-degenerate SU(2) singlet [35,36] with a gap to the first excited states that vanishes in the thermodynamic limit [37,38]. This leads to the spontaneous breaking of the spin rotational $SU(2)$ symmetry in 2D at $T = 0$ [39]. The order parameter is the staggered magnetization

$$\Sigma = \sum_{i \in A} S_i^z - \sum_{j \in B} S_j^z , \tag{15}$$

where $A$ and $B$ are the "even" and "odd" sublattices respectively. In 3D the antiferromagnetic order persists also at $0 \leq T < T_c$ with $T_c/J \simeq 1$ [40–44]. We are interested in calculating the PDF of the staggered magnetization in the presence of long-range order for a local subsystem $\mathcal{A}$ with total number of sites $|\mathcal{A}| = \ell$. We will focus both on the system at equilibrium for a given temperature $T < T_c$ and on the non-equilibrium time evolution following a global quantum quench. For the latter, we will start both from the classical Néel state or the ground state of the XXZ model, and time-evolve according to (14).

To analytically study the model specified by (14), in the presence of long-range order, we employ the Holstein-Primakoff (HP) representation [45,46] of the spin operators $S_i^\gamma$, by introducing two families of bosons $a$ and $b$ respectively associated with sublattices $A$ and $B$

$$\begin{aligned} S_i^z &= s - a_i^\dagger a_i & S_i^+ &= \sqrt{2s} \left( 1 - \frac{1}{2s} a_i^\dagger a_i \right)^{1/2} a_i \\ S_j^z &= -s + b_j^\dagger b_j & S_j^- &= \sqrt{2s} \left( 1 - \frac{1}{2s} b_j^\dagger b_j \right)^{1/2} b_j , \end{aligned} \tag{16}$$

where we have $\mathbf{S}^2 = s(s+1)$ with $s \geq 1/2$, $S^\pm \equiv S^x \pm iS^y$. Taylor expanding the square-root and using the obtained representation in the Hamiltonian (14) results in an expansion in inverse powers of $s$, and spin-wave theory is based on truncating this series [39,46–51]. In the following we will truncate the Hamiltonian at $\mathcal{O}(s^{-2})$ and consider

$$\begin{aligned} H = J \sum_{\langle i,j \rangle} \Bigg[ &-s^2 + s \left( a_i^\dagger a_i + b_j^\dagger b_j + a_i b_j + a_i^\dagger b_j^\dagger \right) \\ &- a_i^\dagger a_i b_j^\dagger b_j - \frac{1}{4} \left( a_i^\dagger a_i^2 b_j + a_i b_j^\dagger b_j^2 + \text{h.c.} \right) \Bigg] . \end{aligned} \tag{17}$$

Given the presence of quartic interactions we will refer to this truncation as HP4, while the truncation at $\mathcal{O}(s^{-1})$ is the standard linear spin-wave theory (LSW). Alternative approaches, based on normal ordering the string of bosonic operators arising after the Taylor expansion of the square-root in (16), can generate exact truncation schemes [46,52]. However, as pointed out already by Kubo in [46], these representations lead to unphysical results when combined with mean-field approximations of the interacting terms, so that we do not consider them here.

### 2.1.1 Self-consistent mean-field approximation in thermal equilibrium

Decoupling the quartic terms in (17) in a mean-field approximation gives

$$ABCD \to \left( \langle AB \rangle CD + AB \langle CD \rangle - \langle AB \rangle \langle CD \rangle \right) + (B \leftrightarrow C) + (B \leftrightarrow D) , \tag{18}$$

where $\langle . \rangle$ denotes a thermal average. The resulting theory is Gaussian, and self-consistency is imposed by determining the thermal average within the mean-field theory

$$\langle AB \rangle \equiv \frac{\mathrm{Tr}\left[ \exp(-\beta H_{\mathrm{MF}}) AB \right]}{\mathrm{Tr}\left[ \exp(-\beta H_{\mathrm{MF}}) \right]} , \tag{19}$$

where $\beta$ is the inverse temperature and $H_{\mathrm{MF}}$ is the Hamiltonian after decoupling according to (18). The thermal Gaussian state obtained in this way is the one that minimizes the free energy associated with the interacting Hamiltonian in the subspace of all Gaussian states. We pass to Fourier space in the bosons by

$$\tilde{a}_k \equiv \sqrt{\frac{2}{N}} \sum_{i \in A} e^{-i \boldsymbol{k} \cdot \boldsymbol{x}_i} a_i \qquad\qquad \tilde{b}_k \equiv \sqrt{\frac{2}{N}} \sum_{j \in B} e^{-i \boldsymbol{k} \cdot \boldsymbol{x}_j} b_j , \tag{20}$$

from which we get the self-consistently decoupled Hamiltonian

$$H_{\mathrm{MF}} = 2dJ \sum_k \left[ \mathrm{Re}(P)(\tilde{a}_k^\dagger \tilde{a}_k + \tilde{b}_k^\dagger \tilde{b}_k) + \gamma(k)(P^* \tilde{a}_k \tilde{b}_{-k} + P \tilde{a}_k^\dagger \tilde{b}_{-k}^\dagger) \right] + C , \tag{21}$$

where $C$ is a constant, $P \equiv s - f - g$ and

$$\begin{aligned}
f &\equiv \langle a_i^\dagger a_i \rangle = \langle b_j^\dagger b_j \rangle \in \mathbb{R} & g &\equiv \langle a_i b_j \rangle \quad i, j \text{ nearest-neighbours} , \\
\gamma(k) &\equiv \frac{1}{2d} \sum_{\vec{\delta}_i} e^{i k \cdot \vec{\delta}_i} & |\gamma(k)| &\leq 1, \ \gamma(k) \in \mathbb{R} .
\end{aligned} \tag{22}$$

In the previous equation $\vec{\delta}_i$ runs over the vectors connecting a site to its nearest neighbours. In mean-field decoupling the interaction we have used the symmetries of the original Hamiltonian (17), i.e. we have set $\langle a^\dagger b \rangle$, $\langle aa \rangle$ and $\langle bb \rangle$ to zero because of the global $U(1)$ symmetry $a \to e^{i\phi} a$, $b \to e^{-i\phi} b$. Furthermore, $\langle a^\dagger a \rangle = \langle b^\dagger b \rangle$ because of the exchange symmetry $a \to b$. Note that setting the mean-fields $f$ and $g$ to zero brings us from the mean-field HP4 to LSW. We will assume that in equilibrium $g \in \mathbb{R}$ and check it self-consistently at the end. With this additional assumption we can diagonalize (21) by a simple canonical Bogoliubov transformation to a new set of bosons $\tilde{\alpha}_k$, $\tilde{\beta}_k$

$$\tilde{a}_k \equiv \cosh \theta_k \, \tilde{\alpha}_k - \sinh \theta_k \, \tilde{\beta}_{-k}^\dagger \qquad\qquad \tilde{b}_k \equiv \cosh \theta_k \, \tilde{\beta}_k - \sinh \theta_k \, \tilde{\alpha}_{-k}^\dagger , \tag{23}$$

with the angle $\theta_k$ defined by $\tanh 2\theta_k = \gamma(k)$. We arrive in this way to the diagonal form

$$H_{\mathrm{MF}} = \sum_k \varepsilon_k (\tilde{\alpha}_k^\dagger \tilde{\alpha}_k + \tilde{\beta}_k^\dagger \tilde{\beta}_k) + E \qquad\qquad \varepsilon_k = 2dJP \sqrt{1 - \gamma(k)^2} , \tag{24}$$

where $E$ is a constant equal to the ground-state energy. Given that $\varepsilon_k \to 0$ for $k \to 0$, the spectrum is gapless, as required by the presence of Goldstone modes associated with the spontaneous symmetry breaking of the $SU(2)$ symmetry. It can be easily checked that below the transition temperature, solutions to the self-consistent equations $f = 1/Z \, \mathrm{Tr}\left[ \exp(-\beta H_{\mathrm{MF}}) a_i^\dagger a_i \right]$ and $g = 1/Z \, \mathrm{Tr}\left[ \exp(-\beta H_{\mathrm{MF}}) a_i b_j \right]$ exist and that $g$ is indeed real. The thermal Gaussian state can be fully characterized by the set of all non-zero 2-point functions, i.e. $\Delta_k^{aa} \equiv \langle \tilde{a}_k^\dagger \tilde{a}_k \rangle$ and

$\Delta_k^{ab} \equiv \langle \tilde{a}_k \tilde{b}_{-k} \rangle$. Given the simplicity of (24), the expectation values of the $\Delta_k$ functions in the Gibbs ensemble, with inverse temperature $\beta$, are easily found to be

$$\Delta_k^{aa}(\beta) = \frac{1}{2} \frac{(2n_k(\beta) + 1)}{\sqrt{1 - \gamma(k)^2}} - \frac{1}{2} \qquad \Delta_k^{ab}(\beta) = -\gamma(k) \left( \Delta_k^{aa}(\beta) + \frac{1}{2} \right), \qquad (25)$$

where $n_k(\beta) = [\exp(\beta \varepsilon_k) - 1]^{-1}$ is the Bose occupation for the mode of momentum $k$.

### 2.1.2 Self-consistent mean-field approximation out of equilibrium I

The formalism in terms of HP bosons just introduced can be easily extended to the out-of-equilibrium scenario. If we start from an initial state that breaks the $SU(2)$ symmetry along the $z$ direction, i.e. $\langle S_i^x \rangle = \langle S_i^y \rangle = 0$, $\langle S_i^z \rangle \neq 0$ $\forall i$, and time evolve according to (14), we find $\langle S_i^x \rangle = \langle S_i^y \rangle = 0$ during the whole time evolution. Thus the HP representation (16) is suitable also out-of-equilibrium, as long as order doesn't melt. The more general case in which during a time evolution the vectorial order parameter $\langle \mathbf{S}_i \rangle$ changes both in direction and magnitude will be discussed in Section 2.3.

To study quench dynamics using the HP representation we apply the self-consistent time-dependent mean-field theory (SCTDMFT) [30, 53–65], reviewed in Appendix A, to the HP4 Hamiltonian (17). The initial states are by construction Gaussian and include the classical Néel state or the ground state of the XXZ Hamiltonian as determined by self-consistent mean-field theory in equilibrium. The time-dependent mean-field Hamiltonian $H_{\mathrm{MF}}(t)$ obtained applying the normal-ordering procedure of Appendix A is formally identical to (21), with the only difference in that the mean-field $P$ acquires an explicit time dependence following the generalization of (18),(19) to

$$\langle AB \rangle(t) \equiv \langle \psi | U_{\mathrm{MF}}^\dagger(t)(AB) U_{\mathrm{MF}}(t) | \psi \rangle \qquad U_{\mathrm{MF}}(t) \equiv \mathcal{T} \left[ \exp \left( -i \int_0^t dt' H_{\mathrm{MF}}(t') \right) \right]. \qquad (26)$$

We remark that even if there is no small parameter in front of the quartic interaction in (17) to justify the applicability of the SCTDMFT approximation, as long as the number of bosons per site is small interactions among them are effectively suppressed, and the approximation is expected to be good on short and intermediate time scales. The Heisenberg equations of motions (EOMs) for the Heisenberg picture operators $\mathcal{O}(t) = U_{\mathrm{MF}}^\dagger(t) \mathcal{O} U_{\mathrm{MF}}(t)$ are

$$\begin{aligned}
\frac{d}{dt} \tilde{a}_k(t) &= -i\, 2\, dJ \left[ \mathrm{Re}(P(t))\, \tilde{a}_k(t) + P(t) \gamma(k) \tilde{b}_{-k}^\dagger(t) \right], \\
\frac{d}{dt} \tilde{b}_k(t) &= -i\, 2\, dJ \left[ \mathrm{Re}(P(t))\, \tilde{b}_k(t) + P(t) \gamma(k) \tilde{a}_{-k}^\dagger(t) \right].
\end{aligned} \qquad (27)$$

From these we find

$$\frac{d}{dt} \Delta_k^{aa} = -4\, dJ \gamma(k) \mathrm{Im}(P^* \Delta_k^{ab}) \qquad \frac{d}{dt} \Delta_k^{ab} = -i\, 4\, dJ \left[ \mathrm{Re}(P) \Delta_k^{ab} + P \gamma(k) \left( \Delta_k^{aa} + \frac{1}{2} \right) \right]. \qquad (28)$$

Given that HP4 doesn't possess local conservation law other than the energy, it is expected to locally relax towards a Gibbs ensemble [66–68], whose effective temperature $1/\beta$ is fixed by the post-quench energy $E_0$ by requiring $E_0 = \langle H_{HP4} \rangle_\beta$. In the spirit of assessing the SCTDMFT of HP4 as an approximation to the time evolution of the non-integrable interacting Heisenberg Hamiltonian (14), we will compare its late time behaviour with the mean-field HP4 Gibbs ensemble at the appropriate effective temperatures. Even if the SCTDMFT is by construction not expected to yield good results at late times [59, 60], in models where local observables relax quickly, i.e. over short time scales where the SCTDMFT is expected to work well, it is possible to describe approximate thermalization by this simple mean-field approach [64, 65].

### 2.1.3 PDF of the staggered magnetization in thermal equilibrium

Thanks to (25) and (28) we have complete knowledge of the Gaussian state $|\psi\rangle$ describing the system both in equilibrium and during a non-equilibrium time evolution. Given the restriction to the local subset $\mathcal{A}$ of $\ell$ total sites, the knowledge of the full state $\rho = |\psi\rangle\langle\psi|$ is redundant and we therefore work with the RDM $\rho_{\mathcal{A}}$. As the partial trace of a Gaussian state is another Gaussian state [69], $\rho_{\mathcal{A}}$ has the form of (9) but without linear terms

$$\rho_{\mathcal{A}} = \frac{1}{Z_{\mathcal{A}}} \exp\left[\frac{1}{2}\boldsymbol{a}^{\dagger} W \boldsymbol{a}\right], \tag{29}$$

where $\boldsymbol{a}$ defined as in (7) is a $2\ell$ vector that accommodates all $a_i$, $a_i^{\dagger}$, $b_j$, $b_j^{\dagger}$ operators. To compute the PDF of the staggered magnetization $\Sigma_{\mathcal{A}}$ in $\mathcal{A}$, which we denote simply as $P_{\mathcal{A}}$, we start by expressing $\Sigma_{\mathcal{A}}$ in terms of the HP bosons

$$\Sigma_{\mathcal{A}} = \ell s - \sum_{i \in A \cap \mathcal{A}} a_i^{\dagger} a_i - \sum_{j \in B \cap \mathcal{A}} b_j^{\dagger} b_j . \tag{30}$$

It is evident that aside for the constant we can cast the previous quadratic observable in the general form (11). The PDF $P_{\mathcal{A}}$ is thus obtained by direct application of (6) and (12).

From now on we focus on the specific case $s = 1/2$, for which the effect of quantum fluctuations is the strongest in differentiating the quantum ground state from the classical ($s \to \infty$) Néel state. All the subsequent results are expected to become more accurate for higher values of $s$. As noted in Ref. [27], as a consequence of the cluster decomposition principle and the central limit theorem, in states with a finite correlation length $\xi$ and for large values of $\ell^{1/d} \gg \xi$, $P_{\mathcal{A}}$ approaches a Gaussian PDF with standard deviation that scales as the square root of the subsystem volume $\ell$. In states with power-law correlations the asymptotic PDF can be non-Gaussian if the decay is slow enough, see e.g. Refs [11, 17, 25].

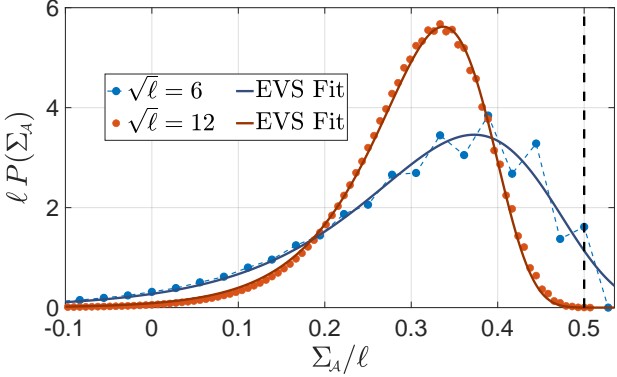

Figure 1: PDF of the staggered magnetization at $T/J = 0$ for the $s = 1/2$ 2D Heisenberg antiferromagnet in a square local subsystem $\mathcal{A}$ with $\ell$ total sites, for $\ell = 36, 144$. Extreme value statistics (EVS) Gumbel fits are also shown. The vertical dashed line indicates the physical threshold of $1/2$ for the staggered magnetization per site.

In Fig. 1 we report $P_{\mathcal{A}}$ for a square subsystem $\mathcal{A}$ in the ground-state of the $s = 1/2$ 2D Heisenberg antiferromagnet, where LSW and mean-field HP4 give the same result, given that the different scaling of $\varepsilon_k$ is irrelevant at $T/J = 0$. Both the probabilities shown yield an average of the staggered magnetization that exactly matches the value of $\langle\Sigma_{\mathcal{A}}\rangle/\ell \simeq 0.303$ reported in literature. As shown in Fig. 1, for intermediate values of $\ell$ $P_{\mathcal{A}}$ is well described by the extreme value statistics (EVS) Gumbel distribution

$$G(x|a, b) = \frac{1}{b} \exp\left[\frac{x-a}{b} - \exp\left(\frac{x-a}{b}\right)\right] . \tag{31}$$

We note that Gumbel distributions have previously appeared in the description of PDFs of fringe visibilities in interference experiments with 1D Bose liquids [25]. In the current context it is a phenomenological fit. The fact that our $P_\mathcal{A}$ is exactly zero beyond the physical threshold $\Sigma_\mathcal{A}/\ell = 1/2$ is a direct consequence of the HP representation for the operator $S^z$, which doesn't allow eigenvalues larger than 1/2. In contrast, the lower bound of $\Sigma_\mathcal{A}/\ell = -1/2$ is violated as the constraint of the boson occupancy being at most one is not strictly enforced. However, in ordered systems this effect is greatly reduced by the condition of having a small number of bosons per site and indeed both curves in Fig. 1 are appreciable only within the physical region. Finally, an even/odd effect in the values of $\Sigma_\mathcal{A}$ is evident, in particular for $\ell = 36$, where the even and odd eigenvalues of the staggered magnetization follow two different smooth curves. Note that such an effect, already reported in PDFs of magnetic models in [17,27], would have been difficult to infer from the sole knowledge of the first few moments of the distributions. As we will see below, the even/odd effect is usually associated with small temperatures and is suppressed by going to large values of $\ell$.

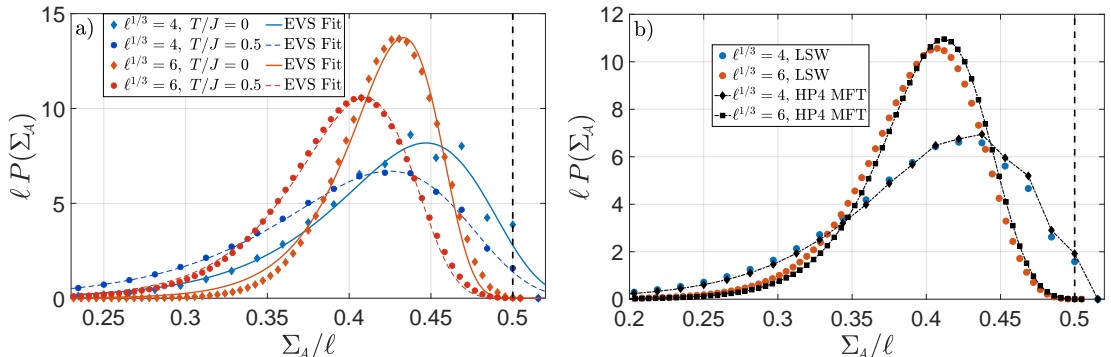

Figure 2: a) LSW PDFs of the staggered magnetization at $T/J = 0$ and $T/J = 0.5$ for the $s = 1/2$ 3D Heisenberg antiferromagnet in a cubic $\mathcal{A}$ with $\ell = 64, 216$, together with extreme value statistics (EVS) Gumbel fits. Being at $T = 0$, these are also the curves for mean-field HP4. b) Differences between LSW and HP4 for $T/J = 0.5$. The vertical dashed lines indicates the physical threshold of 1/2 for the staggered magnetization per site.

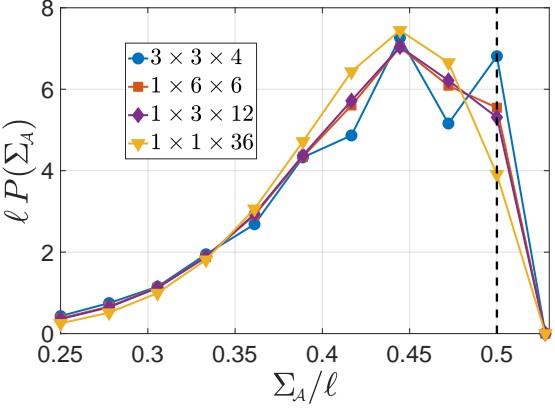

Figure 3: Probability distribution at $T/J = 0$ for the 3D $s = 1/2$ case with different shapes of the subsystem $\mathcal{A}$ having a total of $\ell = 36$ sites.

In Fig. 2 we show the $s = 1/2$ 3D Heisenberg antiferromagnet $P_\mathcal{A}$ in a cubic subsystem $\mathcal{A}$

both in the ground-state and for $T/J = 0.5$, as calculated using LSW and the mean-field HP4 for $\ell = 64, 216$. The probabilities associated with the ground-state give an average $\langle \Sigma_{\mathcal{A}} \rangle / \ell \simeq 0.422$, exactly matching the known value reported in literature. We see how for $T/J = 0.5$ the temperature fluctuations reduce the order possessed by the ground-state. Furthermore, the fact that for the mean-field of HP4 at $T/J = 0.5$ the term $P$ that rescales the dispersion $\varepsilon_k$ (with respect to LSW) takes a value slightly larger than $1/2$ is translated in a lower presence of excitations, which then results in the little right-shift of the mean-field HP4 curves with respect to the LSW ones in Fig. 2b. By comparing the two curves at $T/J = 0$ in Fig. 2a, it is evident the beginning of the transition from a Gumbel-like shape towards a Gaussian one, given that for $\ell = 216$ the fit is not as good as for $\ell = 64$. The plot for $\ell = 64$ and $T/J = 0.5$ also proves that the even/odd effect is not only suppressed by increasing $\ell$, but also from increasing the temperature. In Fig. 3 we compare $P_{\mathcal{A}}$ in four different shapes of a subsystem $\mathcal{A}$ with $\ell = 36$ sites. In low-dimensional shapes longer distances, i.e. less correlated regions, are involved within the subsystem. This causes a shift towards Gaussianity similar to the one obtained by increasing the total number of sites in a fixed shape, like in Fig. 2a. We observe that also the even/odd effect is greatly reduced by this reduction of dimensionality of the subsystem. The results for $0 < T < 0.5$ are very similar to Fig. 3 so that we do not report them explicitly.

### 2.1.4 PDF of the staggered magnetization after quantum quenches

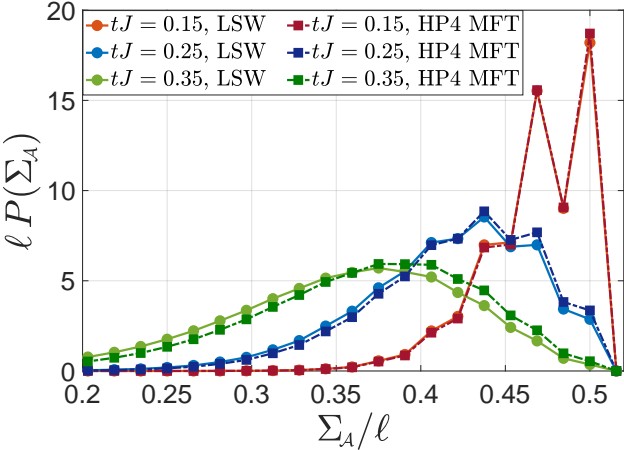

Figure 4: Early instants of the out-of-equilibrium evolution of $P_{\mathcal{A}}$ in a 3D cubic subsystem $\mathcal{A}$ with $\ell = 64$ total site and $s = 1/2$, following a quench from the classical Néel state according to LSW theory and the SCTDMFT of HP4.

We perform out-of-equilibrium time evolution following a global quantum quench starting from ground-states $|GS, \eta\rangle$ of the XXZ model with anisotropy parameter $0 \leq \eta < 1$

$$H_{\text{XXZ}}(\eta) = \sum_{\langle i,j \rangle} \left[ \eta \left( S_i^x S_j^x + S_i^y S_j^y \right) + S_i^z S_j^z \right] . \tag{32}$$

In Fig. 4 we plot the early instants of the time-evolution starting from the classical Néel state $|GS, 0\rangle$ for the 3D case with cubic subsystem $\mathcal{A}$ of $\ell = 64$ total sites and $s = 1/2$. The full order present at $t = 0$ is quickly destroyed by the energy injected in the system by the quench, whose associated effective temperature is of the order of $0.9J$ both for $1/\beta_{\text{LSW}}$ and $1/\beta_{\text{HP4}}$, i.e. very close to the transition temperature. Given that states too close to the transition, and more generally any state where the order is greatly reduced, lie beyond the limit of validity of our

approximation, there is no reason to believe our theory to be any good at later times in this specific case. We note that already at the short times considered small differences between LSW and the SCTDMFT of HP4 arise. Furthermore, the very strong even/odd effect that is present for times $t < 0.20/J$ dies out at times $t > 0.35/J$.

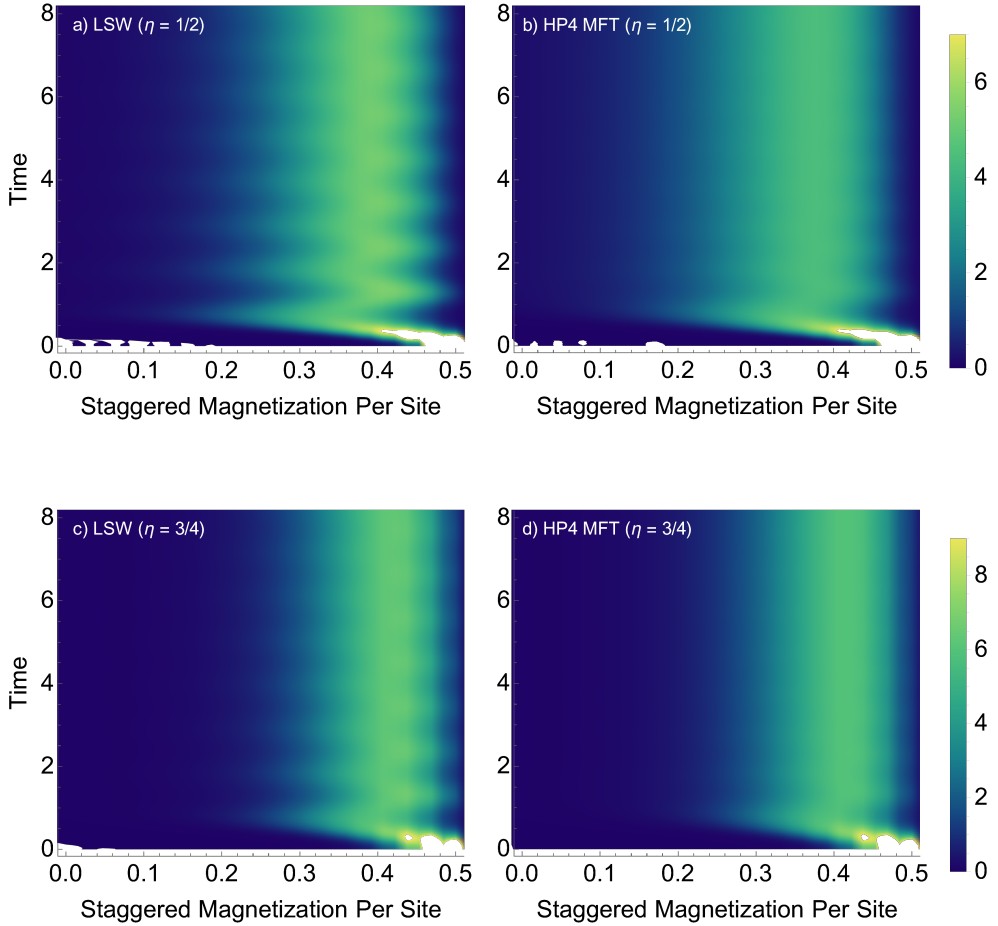

Figure 5: Intensity plots for the time evolution of $P_\mathcal{A}$ in the cubic subsystem $\mathcal{A}$ with $\ell = 64$ and $s = 1/2$, for the quenches from $\eta = 1/2$: a) LSW and b) SCTDMFT of HP4; $\eta = 3/4$: c) LSW and d) SCTDMFT of HP4.

In order to reduce the energy injected in the system by the quench, in the hope to obtain a time-evolution that is accessible to our approximation up to later times, we reduce the parameter jump, i.e. we start from $|GS, \eta\rangle$ with $\eta = 1/2, 3/4$ and $9/10$, $\eta = 1$ being the Heisenberg model. These quenches have, respectively, effective temperatures $1/(J\beta_{\mathrm{LSW}}) \simeq 0.64, 0.50, 0.37$ in LSW theory and $1/(J\beta_{\mathrm{HP4}}) \simeq 0.69, 0.54, 0.40$ in mean-field HP4. Note that in order to compare LSW and the SCTDMFT of HP4 we need to start from exactly the same Gaussian state at initial time, which we take to be the ground-state of the XXZ model according to LSW theory. In Fig. 5 we show the intensity plots for the time evolution of $P_\mathcal{A}$ for the quenches from $\eta = 1/2$ and $3/4$, both in LSW theory and the SCTDMFT of HP4 for the 3D cubic subsystem $\mathcal{A}$ with $\ell = 64$ and $s = 1/2$. In all cases the system starts out at $t = 0$ being strongly ordered, with $P_\mathcal{A}$ having a strong peak very close to $\Sigma_\mathcal{A}/\ell \sim 1/2$, while already at time scales of $tJ \sim 1$ the order is partially suppressed. Given that the effective temperature of the quench for $\eta = 1/2$ is larger than the one for $\eta = 3/4$, as expected we find that stationary state reached at late

times to be less ordered in the former. Fig. 5 also shows how LSW requires more time to relax than the SCTDMFT of HP4, with (slowly decaying) oscillations in $P_{\mathcal{A}}$ going on up to times $Jt = 8$. This is a consequence of the fact that LSW out-of-equilibrium represents a free time evolution, while the SCTDMFT of HP4 takes approximately into account interactions, which favour relaxation.

In Fig. 6 we plot $P_{\mathcal{A}}$ from the SCTDMFT of HP4 with again $s = 1/2$ and 3D cubic $\mathcal{A}$ with $\ell = 64$ sites, at initial, intermediate and late times of the quench dynamics from $\eta = 1/2, 3/4$, $9/10$, and compare it with the thermal state at the appropriate effective temperature. The $\eta = 3/4$ quench is also produced for $\ell = 216$. The deep quench $\eta = 1/2$, characterized as seen by an effective temperature which is beyond half of the transition one, presents a strong difference between the initial PDF and the late-time stationary one. It is thus surprising that even for such a high energy injected through the quench the agreement between the late-time $P_{\mathcal{A}}$ and the thermal one is reasonable. In the case $\eta = 9/10$ the match between late time behaviour and Gibbs ensemble is essentially exact on the scale of Fig. 6c, which includes the initial probability. The even/odd effect present in every initial state of Fig. 6 is seen to disappear in the late time dynamics, except for the shallow quench $\eta = 9/10$, where a slightly suppressed even/odd effect is still visible at times $Jt \sim 5$. Finally, by comparing the two quenches for $\eta = 3/4$ of size $\ell = 64$ in Fig. 6b and $\ell = 216$ in 6d, we note that the agreement thermal state/stationary state in the latter is slightly worse. Given that in producing the PDFs all that matters are the 2-point functions in real space within the subsystem $\mathcal{A}$, this effect is consequence of the fact that the match in the 2-point functions between thermal and stationary state is slightly better for shorter distances, like the ones involved in the $\ell = 64$ plot.

## 2.2 Long-range transverse field Ising chain

So far we have considered the non-equilibrium evolution after quantum quenches from initial states that are magnetically ordered along the same direction as the stationary state reached at late times. We will now consider situations where the initial state is magnetically ordered along a different direction than the stationary state. In order to make contact with previous results in the literature we focus on the one-dimensional Ising model with long-range interactions

$$H = -\frac{1}{2s} \sum_{i,j=1}^{N} J_{ij} S_i^x S_j^x - h \sum_{i=1}^{N} S_i^z \,, \tag{33}$$

where $h$ is the transverse field that contrasts the alignment along the $\hat{x}$ direction and $S_i^\gamma$ are spin $s$ degrees of freedom. In presence of open boundary conditions the long-range spin interaction $J_{ij}$ is given by

$$J_{ij} = \frac{1}{Q(\alpha)} \frac{J}{|i-j|^\alpha} \qquad Q(\alpha) = \frac{1}{(N-1)} \frac{1}{2} \sum_{i \neq j} \frac{1}{|i-j|^\alpha} \,, \tag{34}$$

where $\alpha$ is a non-negative real parameter, the self-interaction $J_{ii}$ is set to zero for every $i$ and $Q(\alpha)$ is a normalization factor needed to ensure intensive scaling of the energy density for $\alpha \leq 1$ in the thermodynamic limit. Translational invariance can be recovered by modifying the coupling to be explicitly periodic on the lattice [70]

$$J_{ij} \equiv J \left| \frac{N}{\pi} \sin\left[ \frac{\pi(i-j)}{N} \right] \right|^{-\alpha} \xrightarrow{N \to \infty} \frac{J}{|i-j|^\alpha} \,, \tag{35}$$

but retaining the original form in the thermodynamic limit for $|i-j|$ finite. At $\alpha = \infty$ the Hamiltonian (33) possesses only nearest neighbours interactions and the model is exactly solvable by a Jordan-Wigner transformation [71] that maps the interacting spin model into a model

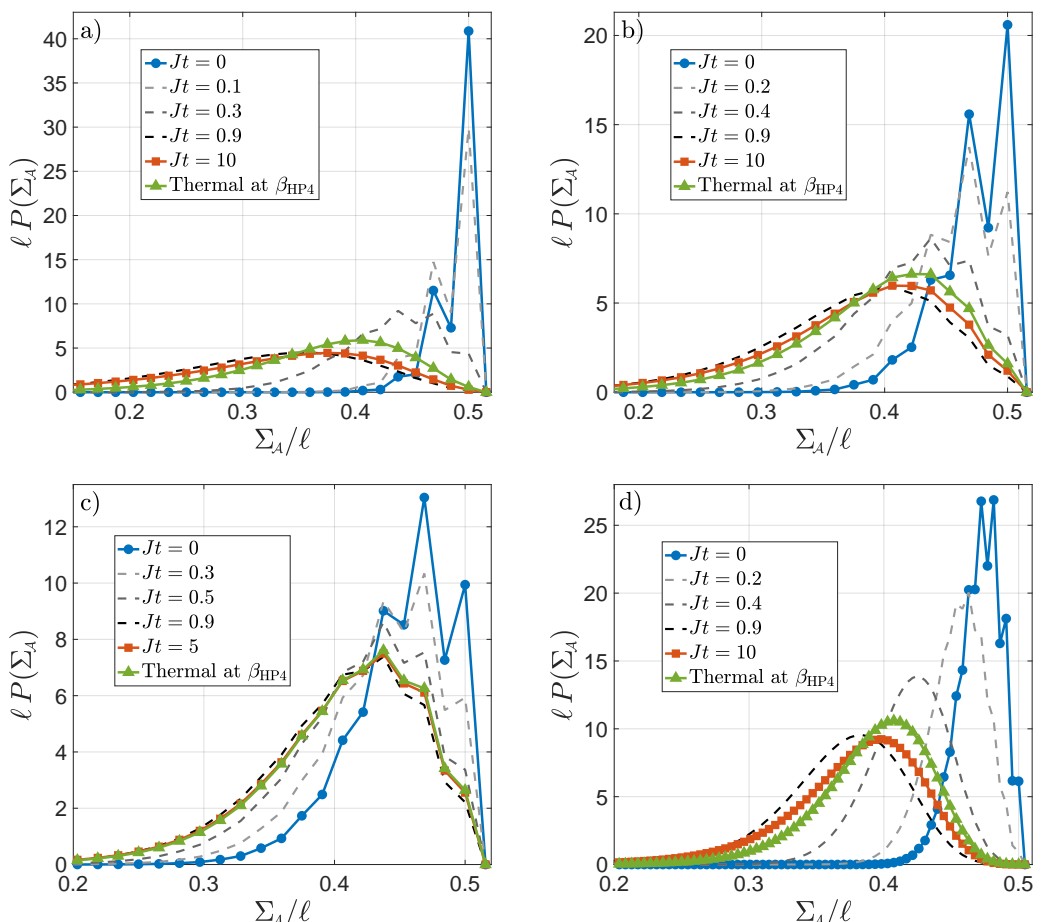

Figure 6: Initial time, intermediate and late times $P_{\mathcal{A}}$, in the $s$-1/2 3D cubic subsystem with $\ell = 64$, for quenches from the ground-state of the XXZ model with a) $\eta = 1/2$, b) $\eta = 3/4$, c) $\eta = 9/10$. d) Same plot with $\ell = 216$ and $\eta = 3/4$. The late time probability distributions are compared with the equilibrium ones at appropriate effective temperature $\beta_{\text{HP4}}$.

of free fermions [35]. At $\alpha = 0$ the model reduces to the fully connected transverse field Ising model and the Hamiltonian (33) is expressible in terms of rescaled total spin operators

$$s^\gamma = \frac{1}{N}\sum_{i=1}^{N} S_i^\gamma \qquad \gamma = x, y, z \ , \tag{36}$$

whose algebra, in the thermodynamic limit, reduces to the one of classical spins

$$[s^i, s^j] = i\frac{1}{N}\epsilon_{ijk}s^k \quad \xrightarrow{N\to\infty} \quad 0 \ . \tag{37}$$

For $0 < \alpha < \infty$ the model is non-integrable and is therefore expected to thermalize. In equilibrium and for sufficiently long-ranged interactions $\alpha < 2$ ferromagnetic long-range order exists below a critical temperature $T < T_c$ for $|h| < h_c$ [72]. Out-of-equilibrium, the long-range model has been extensively studied in the literature and shown to exhibit a number of interesting phenomena [70, 73–79]. Given the existence of magnetic order at finite temperatures, we expect quantum quench dynamics from initial states with sufficiently low energy density above the ground state (which correspond to effective temperatures $T_{\text{eff}} < T_c$) to result in relaxation to thermal states with ferromagnetic order. The presence of magnetic order during

the entire time evolution in such cases has been verified for $\alpha = 0$ [79–81] and is observed numerically for $\alpha > 0$ [28, 76].

### 2.2.1 Self-consistent mean-field approximation out of equilibrium II

For the sake of simplicity we now constrain our discussion to translationally invariant situations. The classical ground state of (33) is a ferromagnetic state where the spins are ordered along the direction

$$\vec{n} = \begin{pmatrix} \sin\theta \cos\phi \\ \sin\theta \sin\phi \\ \cos\theta \end{pmatrix} , \tag{38}$$

where

$$\phi = 0 , \qquad \theta = \arccos(h/\Gamma) , \quad \Gamma = \frac{1}{N} \sum_{i,j} J_{i,j} . \tag{39}$$

Here the spin-rotational symmetry by an angle $\pi$ around the z-axis is broken spontaneously and we have, without loss of generality, chosen the ground state with $\phi = 0$. The dynamical properties of this model can be analysed by spin-wave theory. In particular, we expect that quench-dynamics is amenable to a spin-wave analysis as long as the initial state belongs to the low-energy part of the Hilbert space of states that is described by spin-wave theory. As an example we consider a quench from a saturated ferromagnetic state $|\psi(0)\rangle$ along the direction

$$\vec{n}_0 = \begin{pmatrix} \sin\theta_0 \cos\phi_0 \\ \sin\theta_0 \sin\phi_0 \\ \cos\theta_0 \end{pmatrix} , \quad |\theta - \theta_0|, |\phi - \phi_0| \ll 1 . \tag{40}$$

We introduce the matrix

$$R(\theta, \phi) = \begin{bmatrix} \cos\theta \cos\phi & -\sin\phi & \sin\theta \cos\phi \\ \cos\theta \sin\phi & \cos\phi & \sin\theta \sin\phi \\ -\sin\theta & 0 & \cos\theta \end{bmatrix} , \tag{41}$$

and proceed by rotating the spin-quantization axis to be along $\vec{n}_0$, which amounts to defining new spin operators through

$$S_j^\alpha = R_0^{\alpha\beta} \sigma_j^\beta \qquad R_0 \equiv R(\theta_0, \phi_0) . \tag{42}$$

We then employ a Holstein-Primakoff representation for $\sigma_j^\alpha$

$$\sigma_j^z = s - a_j^\dagger a_j , \qquad \sigma_j^+ = \sqrt{2s - a_j^\dagger a_j} \, a_j . \tag{43}$$

This generates a $1/s$-expansion of the Hamiltonian

$$H = s\,h_0 + \sqrt{s}\,h_1 + s^0\,h_2 + s^{-1/2}\,h_3 + \dots , \tag{44}$$

where $h_0$ is a constant and

$$h_1 = \sum_j \lambda \, a_j + \text{h.c.} ,$$
$$h_2 = \sum_{i,j} \left[ t_{ij} a_i^\dagger a_j + \left( \Delta_{ij} a_i a_j + \text{h.c.} \right) \right] ,$$
$$h_3 = \sum_{i,j} V_{ij} a_i^\dagger a_j^\dagger a_j + \text{h.c.} . \tag{45}$$

Here we have defined

$$\lambda = -\frac{h}{\sqrt{2}}R_0^{zx} - \frac{\Gamma}{\sqrt{2}}\big(R_0^{xx} - iR_0^{xy}\big)R_0^{xz},$$

$$\Delta_{jk} = \Delta_{j-k} = -\frac{J_{jk}}{4}\big[\big(R_0^{xx}\big)^2 - 2iR_0^{xx}R_0^{xy} - \big(R_0^{xy}\big)^2\big],$$

$$t_{jk} = t_{j-k} = -\frac{J_{jk}}{2}\big[\big(R_0^{xx}\big)^2 + \big(R_0^{xy}\big)^2\big] + \delta_{j,k}\big[hR_0^{zz} + \Gamma\big(R_0^{xz}\big)^2\big],$$

$$V_{jk} = V_{j-k} = \delta_{j,k}\frac{h}{4\sqrt{2}}R_0^{zx} + \frac{1}{\sqrt{2}}\big(R_0^{xx} + iR_0^{xy}\big)R_0^{xz}\big[J_{jk} + \frac{\Gamma}{4}\delta_{j,k}\big]. \tag{46}$$

Importantly we have by virtue of (40) that

$$|\lambda| \ll 1, \tag{47}$$

which makes the $\mathcal{O}(\sqrt{s})$ term in the Hamiltonian parametrically small and precludes the generation of a large bosonic occupation $n_j = \langle a_j^\dagger a_j\rangle$ over time. We now take into account the interaction terms $h_3$ in the framework of the SCTDMFT of Appendix A. Denoting by $^\times_\times\mathcal{O}^\times_\times$ normal ordering with respect to the Gaussian mean-field state $|\psi_{\mathrm{MF}}(t)\rangle$ and $\langle\mathcal{O}\rangle_t = \langle\psi_{\mathrm{MF}}(t)|\mathcal{O}|\psi_{\mathrm{MF}}(t)\rangle$, we have for example

$$a_i^\dagger a_j^\dagger a_j = {}^\times_\times a_i^\dagger a_j^\dagger a_j{}^\times_\times + m\,a_i^\dagger a_j^\dagger + m^*a_i^\dagger a_j + m^*\hat{n}_j + a_i^\dagger(f_{jj} - 2|m|^2) + a_j^\dagger(f_{ij} - 2|m|^2)$$

$$+ a_j(g_{ij}^* - 2m^*m^*) - mg_{ij}^* - m^*f_{ij} - m^*f_{jj} + 4m^*|m|^2, \tag{48}$$

where we have defined

$$m(t) = \langle a_j\rangle_t, \quad f_{ij}(t) = f_{i-j}(t) = \langle a_i^\dagger a_j\rangle_t, \quad g_{ij}(t) = g_{i-j}(t) = \langle a_i a_j\rangle_t. \tag{49}$$

In the SCTDMFT we drop the cubic normal ordered term. We then ought to proceed analogously with the infinitely many interaction terms in the $1/s$-expansion (44) of the Hamiltonian, which is a difficult problem. In practice we instead truncate the $1/s$-expansion by discarding all contributions beyond $h_3$. This leads to the self-consistent mean-field Hamiltonian of the form

$$H_{\mathrm{MF}}(t) = \sum_{i,j}\big[A_{ij}(t)a_i^\dagger a_j + (B_{ij}(t)a_i a_j + \mathrm{h.c.})\big] + \sum_j\big(\Lambda(t)a_j + \mathrm{h.c.}\big) + C(t), \tag{50}$$

where $C(t)$ is a constant and

$$\Lambda(t) = \sqrt{2s}\lambda + \frac{1}{\sqrt{2s}}\sum_\ell\Big[V_{\ell j}^*\big(f_{j\ell}(t) - 2|m(t)|^2\big)$$

$$+ V_{j\ell}^*\big(f_{\ell\ell}(t) - 2|m(t)|^2\big) + V_{\ell j}\big(g_{\ell j}^*(t) - 2m^*(t)m^*(t)\big)\Big],$$

$$A_{jk}(t) = t_{jk} + \frac{1}{\sqrt{2s}}\big(V_{jk}m^*(t) + \mathrm{c.c.}\big) + \frac{1}{\sqrt{2s}}\delta_{jk}\Big(m^*(t)\sum_\ell V_{\ell k} + \mathrm{c.c.}\Big),$$

$$B_{jk}(t) = \Delta_{jk} + \frac{1}{\sqrt{2s}}V_{jk}^*m^*(t). \tag{51}$$

The self-consistency conditions follow straightforwardly from the EOMs

$$\dot{m} = -i\sum_\ell\big(A_{j\ell}m + 2B_{j\ell}^*m^*\big) - i\Lambda^*,$$

$$\dot{f}_{jk} = i\sum_\ell\big(A_{j\ell}f_{\ell k} - A_{k\ell}f_{j\ell} + 2B_{j\ell}g_{\ell k} - 2B_{k\ell}^*g_{j\ell}^*\big) + i\big(\Lambda m - \Lambda^*m^*\big),$$

$$\dot{g}_{jk} = -i\sum_\ell\big(A_{j\ell}g_{\ell k} + A_{k\ell}g_{j\ell} + 2B_{j\ell}^*f_{\ell k} + 2B_{k\ell}^*(\delta_{j,\ell} + f_{\ell j})\big) - 2i\Lambda^*m. \tag{52}$$

These EOMs conserve energy, as proved in full generality in Appendix A.

In order to benchmark this approximation we focus on the $\alpha = 0$ case for finite sizes of the system, where quantum effects are important even though the problem is equivalent to the time evolution of the single large spin (36). Choosing as initial configuration a fully ordered state restricts us to the sector of maximum total angular momentum $\boldsymbol{s}^2 = s(s + 1/N)$, which is conserved by the fully connected Hamiltonian[1]. Thus, the dimension of the part of the Hilbert space involved in the time evolution is simply $2W + 1$ where $W \equiv Ns$ and we are able to obtain exact results from diagonalization of the Hamiltonian in the restricted sector. In the fully connected case the formalism of the previous section can also be expressed in terms of a single boson $a$ introduced by the HP transformation whose $z$-component reads $\sum_j \sigma_j^z = W - a^\dagger a$. From this is clear that the large parameter $s$ from last section is here replaced by $W$, i.e. the total number of lattice sites contribute to the main variable that controls our truncation scheme. Fig. 7 shows the exact time evolution for quenches starting from $\cos \theta_0 = \delta \cdot h/\Gamma$ for several values of $\delta$ and compares to the results from our truncated spin-wave theory. As expected the approximation becomes worse as we increase the angle between the initial direction and the stationary one. Fig. 7c shows how for small values of $\delta$ the approximation remains reasonable up to fairly large times. Comparison of Fig. 7a and 7c shows that the goodness of the approximation primarily depends on the value of $\delta$ and is not strongly influenced by the large parameter $W$.

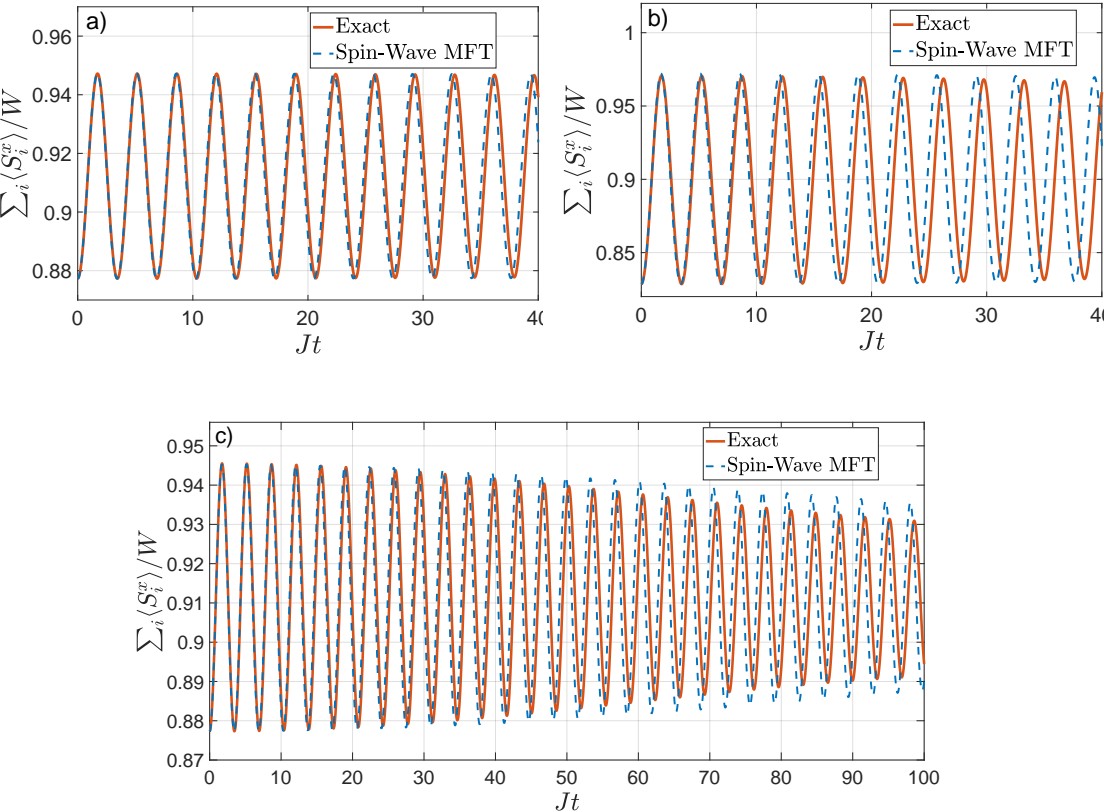

Figure 7: Exact time evolution of $\sum_i \langle S_i^x \rangle$ in the fully connected TFIC with $h = 0.8$, $\Gamma = 2J$, $\cos \theta_0 = \delta \cdot h/\Gamma$ and comparison with the SCTDMFT of the cubic spin-wave approximation. a) $W = 1000$ and $\delta = 1.2$; b) $W = 1000$ and $\delta = 1.4$; c) $W = 100$ and $\delta = 1.2$.

---

[1]For spin variables with $s \geq 1$ this is true only if in the fully connected TFIC we re-insert the self-interactions $J_{ii} \neq 0$, which introduces minor changes to the equations derived in this section.

## 2.3 Spin-wave theory around a time-dependent direction

As we have seen in the previous subsection, if the magnetic order in the initial state is oriented along a different direction compared to the order in the stationary state reached at late times, a self-consistent spin-wave approximation in a fixed frame is quantitatively accurate only if the angle between the two orders is small. A possible path around this limitation was proposed in Refs [78, 79] for the example of an infinite-range transverse-field Ising chain (TFIC) with additional finite-range perturbations, by considering spin-wave theory in a self-consistently determined rotating frame. The method has been further developed in the infinite-range case [82] and extended to generic models with spatially decaying long-range interactions in [73]. However, it still possesses a number of serious complications that arise when one tries to implement a SCTDMFT scheme in a rotating frame – for example it generally does not conserve energy[2].

We will now revisit the approach of Refs [78,79] and show that a more accurate approximation can be obtained by employing a systematic $1/s$-expansion. This will in particular expose fundamental differences with spin-wave theory in equilibrium, such as the necessity of $s$ being large when the interactions are sufficiently short-ranged.

We consider a generic translationally invariant spin-$s$ Hamiltonian in $d$ dimensions of the form

$$H = -\frac{1}{2s} \sum_{i,j} \sum_{\alpha \leq \beta} J_{ij}^{\alpha\beta} S_i^\alpha S_j^\beta - \sum_i \sum_\alpha h^\alpha S_i^\alpha \, , \tag{53}$$

where the self-interaction $J_{ii}^{\alpha\beta}$ is set to zero and

$$J_{ij}^{\alpha\beta} := J^{\alpha\beta}(\boldsymbol{r}_i - \boldsymbol{r}_j) \qquad\qquad J_{ij}^{\alpha\beta} = J_{ji}^{\alpha\beta} \qquad\qquad i \neq j \, . \tag{54}$$

Here $\boldsymbol{r}_i$ is the position of site $i$ on the $d$-dimensional lattice. The basic idea of Ref. [79] is to introduce a rotating reference frame, in which the $z$-axis is aligned with the direction specified by the time-evolving expectation value of the spin operators $\langle \boldsymbol{S}_i \rangle$. This amounts to defining new spin operators $\sigma_{t,i}^\gamma$ fulfilling SU(2) commutation relations by

$$\sigma_{t,i}^\gamma = S_i^\mu R^{\mu\gamma}(t) = V(t)^\dagger S_i^\gamma V(t) \, , \tag{55}$$

where $R(t) = R(\theta(t), \phi(t))$ is the matrix (41) and

$$V(t) = \exp\left(i\theta(t) \sum_j S_j^y\right) \exp\left(i\phi(t) \sum_j S_j^z\right) \, . \tag{56}$$

In the approach of [78, 79] the rotation matrix $R(t)$ is to be determined self-consistently by requiring

$$\langle \psi_i | \sigma_{t,i}^\beta(t) | \psi_i \rangle = 0 \qquad\qquad \forall t, \ \beta = x, y \quad , \tag{57}$$

where $|\psi_i\rangle$ is the known initial state at time $t = 0$ and $\sigma_{t,i}^\beta(t) = e^{iHt} \sigma_{t,i}^\beta e^{-iHt}$. In the following we will relax the condition (57) for convenience and require the expectation value to be merely small instead of zero. However, we stress that our conclusions remain unchanged if we impose (57) strictly. Using the definition (55) it is straightforward to show that the new spin operators fulfil equations of motion

$$\frac{d}{dt} \sigma_{t,i}^\gamma(t) = i\left[\tilde{H}(t), \sigma_{t,i}^\gamma(t)\right] \, , \tag{58}$$

---

[2]In practice there can of course be situations where the change in energy is extremely small on short and intermediate time scales, and the method will provide a good approximation.

where

$$\tilde{H}(t) = -\frac{1}{2s}\sum_{i,j}\sum_{\alpha\leq\beta}\sum_{\mu,\nu}J_{ij}^{\alpha\beta}R^{\alpha\mu}(t)R^{\beta\nu}(t)\sigma_{t,i}^{\mu}(t)\sigma_{t,j}^{\nu}(t) - \sum_{i}\sum_{\alpha,\mu}h^{\alpha}R^{\alpha\mu}(t)\sigma_{t,i}^{\mu}(t),$$
$$-\dot{\theta}(t)\sum_{i}\sigma_{t,i}^{y}(t) - \dot{\phi}(t)\sum_{i}\sum_{\mu}R^{z\mu}(t)\sigma_{t,i}^{\mu}(t). \tag{59}$$

As in the rotating frame the spins are approximately aligned along the z-axis, *cf.* (57), it is natural to employ a HP representation of the new spin-operators

$$\sigma_{t,i}^{z}(t) = s - a_{i}^{\dagger}(t)a_{i}(t) \qquad\qquad \sigma_{t,i}^{+}(t) = \sqrt{2s - a_{i}^{\dagger}(t)a_{i}(t)}\,a_{i}(t). \tag{60}$$

The ability to define bosonic variables $a_{i}(t)$ whose occupation number is small is the main rationale behind the rotating frame approach. Combining the equations of motion (58) with (60) results in a BBGKY hierarchy for the equal time expectation values of products of HP boson operators. In order to exhibit this structure we find it convenient to express the spin operators $\sigma_{t,i}^{\gamma}(t)$ as

$$\sigma_{t,i}^{\gamma}(t) = \tilde{U}(t)^{\dagger}\sigma_{0,i}^{\gamma}\tilde{U}(t), \qquad \sigma_{0,i}^{\gamma} = S_{i}^{\mu}R^{\mu\gamma}(0). \tag{61}$$

Here the unitary operator $\tilde{U}(t)$ combines the effects of the rotating frame and the time evolution, and is given by

$$\tilde{U}(t) = \mathcal{T}\left[\exp\left(-i\int_{0}^{t}dt'\tilde{h}(t')\right)\right], \qquad \tilde{h}(t) \equiv \tilde{H}(t)\Big|_{\sigma_{t,i}^{\gamma}(t)\to\sigma_{0,i}^{\gamma}}. \tag{62}$$

Defining a state

$$|\tilde{\psi}(t)\rangle \equiv \tilde{U}(t)|\psi_{i}\rangle, \tag{63}$$

which is different from the standard time-evolved state $|\psi(t)\rangle \equiv \exp(-iHt)|\psi_{i}\rangle$, the requirement (57) becomes

$$\langle\tilde{\psi}(t)|\sigma_{0,i}^{\beta}|\tilde{\psi}(t)\rangle = 0 \qquad\qquad \forall t,\ \beta = x, y \quad, \tag{64}$$

with now $\langle\tilde{\psi}(t)|\sigma_{0,i}^{z}|\tilde{\psi}(t)\rangle$ being close to $s$. This leads us to employ the time-independent HP representation

$$\sigma_{0,i}^{z} = s - a_{i}^{\dagger}a_{i} \qquad\qquad \sigma_{0,i}^{+} = \sqrt{2s - a_{i}^{\dagger}a_{i}}\,a_{i}. \tag{65}$$

In the following we will determine the time evolution of the spins $\sigma_{0,j}^{\alpha}$ using spin-wave methods. Equal time expectation values of the physical spins are then obtained by

$$\langle\psi(t)|S_{j_{1}}^{\alpha_{1}}\ldots S_{j_{n}}^{\alpha_{n}}|\psi(t)\rangle = R^{\alpha_{1}\beta_{1}}(t)\ldots R^{\alpha_{n}\beta_{n}}(t)\,\langle\tilde{\psi}(t)|\sigma_{0,j_{1}}^{\beta_{1}}\ldots\sigma_{0,j_{n}}^{\beta_{n}}|\tilde{\psi}(t)\rangle. \tag{66}$$

The formalism just presented can be easily generalized to non-translationally invariant cases by introducing a site-dependence on the rotation angles.

### 2.3.1 Holstein-Primakoff representation and $1/s$-expansion

When expressed using Holstein-Primakoff bosons the Hamiltonian $\tilde{h}(t)$ takes the form

$$\tilde{h}(t) = s\,\tilde{h}_{0}(t) + s^{1/2}\tilde{h}_{1}(t) + s^{0}\tilde{h}_{2}(t) + s^{-1/2}\tilde{h}_{3}(t) + \ldots, \tag{67}$$

where in close similarity with (45) we have

$$\tilde{h}_1(t) = \sum_j \lambda(t) a_j + \text{h.c.},$$

$$\tilde{h}_2(t) = \sum_{j\ell} \left[ t_{j\ell}(t) a_j^\dagger a_\ell + \left( \Delta_{j\ell}(t) a_j a_\ell + \text{h.c.} \right) \right],$$

$$\tilde{h}_3(t) = \sum_{j\ell} V_{j\ell}(t) a_j^\dagger a_\ell^\dagger a_\ell + \text{h.c.} . \tag{68}$$

By defining $2M_{\alpha\beta}^{\mu\nu} \equiv R^{\alpha\mu} R^{\beta\nu} + R^{\alpha\nu} R^{\beta\mu}$ and $\Gamma^{\alpha\beta} \equiv \sum_\ell J_{j\ell}^{\alpha\beta}$, the coefficients appearing in the previous two equations read

$$\tilde{h}_0(t) = -\frac{N}{2} \sum_{\alpha\leq\beta} \Gamma^{\alpha\beta} M_{\alpha\beta}^{zz}(t) - N \sum_\alpha \left( h^\alpha + \delta^{\alpha z} \dot{\phi}(t) \right) R^{\alpha z}(t),$$

$$\lambda(t) = -\sum_{\alpha\leq\beta} \Gamma^{\alpha\beta} \frac{M_{\alpha\beta}^{xz}(t) - i M_{\alpha\beta}^{yz}(t)}{\sqrt{2}} - \sum_\alpha \frac{h^\alpha + \delta^{\alpha z} \dot{\phi}(t)}{\sqrt{2}} \left[ R^{\alpha x}(t) - i R^{\alpha y}(t) \right] + \frac{i}{\sqrt{2}} \dot{\theta}(t),$$

$$\Delta_{j\ell}(t) = -\frac{1}{4} \sum_{\alpha\leq\beta} J_{j\ell}^{\alpha\beta} \left[ M_{\alpha\beta}^{xx}(t) - 2i M_{\alpha\beta}^{xy}(t) - M_{\alpha\beta}^{yy}(t) \right],$$

$$t_{j\ell}(t) = \delta_{j,l} \Big[ \sum_{\alpha\leq\beta} \Gamma^{\alpha\beta} M_{\alpha\beta}^{zz}(t) + \sum_\alpha \left( h^\alpha + \delta^{\alpha z} \dot{\phi}(t) \right) R^{\alpha z}(t) \Big] - \sum_{\alpha\leq\beta} \frac{J_{j\ell}^{\alpha\beta}}{2} \left[ M_{\alpha\beta}^{xx}(t) + M_{\alpha\beta}^{yy}(t) \right],$$

$$V_{j\ell}(t) = \frac{1}{\sqrt{2}} \sum_{\alpha\leq\beta} \left[ \frac{1}{4} \delta_{j,\ell} \Gamma^{\alpha\beta} + J_{j\ell}^{\alpha\beta} \right] \left( M_{\alpha\beta}^{xz}(t) + i M_{\alpha\beta}^{yz}(t) \right)$$

$$+ \frac{1}{4\sqrt{2}} \delta_{j,\ell} \Big[ \sum_\alpha \left( h^\alpha + \delta^{\alpha z} \dot{\phi}(t) \right) \left( R^{\alpha x}(t) + i R^{\alpha y}(t) \right) + i \dot{\theta}(t) \Big]. \tag{69}$$

We note that (69) reduce to (46) if we set $\dot{\theta} = \dot{\phi} = 0$ and choose the $J_{j\ell}^{\alpha\beta}$, $h^\alpha$ associated with the long-range TFIC of (33).

### 2.3.2 Gaussian approximation and energy non-conservation

The next step after going to the rotating frame and expressing the spins in terms of boson operators is to approximate $|\tilde{\psi}(t)\rangle$ by a SCTDMFT Gaussian bosonic state

$$|\tilde{\psi}_{\text{MF}}(t)\rangle = \mathcal{T} \left[ \exp\left( -i \int_0^t dt' \tilde{h}_{\text{MF}}(t') \right) \right] |\tilde{\psi}(0)\rangle, \tag{70}$$

where the quadratic $\tilde{h}_{\text{MF}}(t)$ is obtained starting with $\tilde{h}(t)$ and applying the normal ordering procedure of Appendix A at every order in the infinite $1/s$ expansion[3]. An important issue that arises in this step is whether energy is conserved. From (66) we have

$$E = \langle \psi(t)| H |\psi(t)\rangle = \langle \tilde{\psi}(t)| \tilde{h}(t) - \tilde{h}^{\theta\phi}(t) |\tilde{\psi}(t)\rangle, \tag{71}$$

where

$$\tilde{h}^{\theta\phi}(t) = -\dot{\theta}(t) \sum_i \sigma_{0,i}^y - \dot{\phi}(t) \sum_i \sum_\mu R^{z\mu}(t) \sigma_{0,i}^\mu. \tag{72}$$

---

[3]As already mentioned before, in practice the mean-field decoupling of this infinite series is a difficult problem and one usually starts by truncating the original Hamiltonian. However, our conclusions on the non-conservation of energy clearly also hold in truncated versions of the theory.

In the SCTDMFT we replace $|\tilde{\psi}(t)\rangle$ by $|\tilde{\psi}_{\mathrm{MF}}(t)\rangle$ and the mean-field energy then is

$$E_{\mathrm{MF}}(t) = \langle\tilde{\psi}_{\mathrm{MF}}(t)|\tilde{h}(t) - \tilde{h}^{\theta\phi}(t)|\tilde{\psi}_{\mathrm{MF}}(t)\rangle . \tag{73}$$

By construction we take our initial state to be Gaussian, so that $E_{\mathrm{MF}}(0) = E$. Making use of the underlying normal ordering in the definition of any self-consistent mean-field decoupled term, the time derivative gives

$$\dot{E}_{\mathrm{MF}}(t) = -i\langle\tilde{\psi}_{\mathrm{MF}}(t)|[\tilde{h}^{\theta\phi}(t) - \tilde{h}_{\mathrm{MF}}^{\theta\phi}(t), \tilde{h}(t) - \tilde{h}_{\mathrm{MF}}(t)]|\tilde{\psi}_{\mathrm{MF}}(t)\rangle. \tag{74}$$

In contrast to the general proof of conservation of energy in any "fixed-frame" formalism (Appendix A), the previous term will generally not vanish. Hence we either need to adjust the rotation $R(t)$ to ensure that $\dot{E}_{\mathrm{MF}}(t) = 0$, or restrict the application of the SCTDMFT to a time window in which the change in energy is very small. We have investigated the former for the example of the long-range TFIM but found that the resulting approximation becomes poor after a relatively short time. We therefore restrict our discussion to the latter in the following.

### 2.3.3 Cubic SCTDMFT ("Cubic I")

In practice it is necessary to truncate the $1/s$-expansion expansion and we now consider an approximation in which we drop all terms $\tilde{h}_{n\geq 4}(t)$. All higher truncations can be treated in the same way. We start by fixing the rotation matrices $R^{\alpha\beta}(t)$ such that $\lambda(t) = 0$ in (68). The corresponding matrix is denoted by $R_0(t)$ and $\theta_0(t)$, $\phi_0(t)$ coincide with the classical rotation angles. It is convenient to choose $\theta_0(0)$ and $\phi_0(0)$ such to align the z-axis of the rotating frame with the direction of magnetic order in the initial state. We then treat the cubic term $\tilde{h}_3(t)$ in the same way as in the fixed-frame approach of Section 2.2 to arrive at a mean-field Hamiltonian

$$\tilde{h}_{\mathrm{MF}}(t) = \sum_{i,j}\left[A_{ij}(t)a_i^\dagger a_j + (B_{ij}(t)a_i a_j + \mathrm{h.c.})\right] + \sum_j\left(\Lambda(t)a_j + \mathrm{h.c.}\right) + C(t) . \tag{75}$$

Here $C(t)$ is a constant and

$$\Lambda(t) = \frac{1}{\sqrt{2s}}\sum_\ell\left[V_{\ell j}^*(t)\left(f_{j\ell}(t) - 2|m(t)|^2\right) + V_{j\ell}^*(t)\left(f_{\ell\ell}(t) - 2|m(t)|^2\right)\right.$$
$$\left. + V_{\ell j}(t)\left(g_{\ell j}^*(t) - 2m^*(t)m^*(t)\right)\right],$$
$$A_{jk}(t) = t_{jk}(t) + \frac{1}{\sqrt{2s}}\left(V_{jk}(t)m^*(t) + V_{kj}^*(t)m(t)\right) + \frac{1}{\sqrt{2s}}\delta_{jk}\left(m^*(t)\sum_\ell V_{\ell k}(t) + \mathrm{c.c.}\right),$$
$$B_{jk}(t) = \Delta_{jk}(t) + \frac{1}{\sqrt{2s}}V_{jk}^*(t)m^*(t) . \tag{76}$$

The mean fields $m(t)$, $f_{jk}(t)$ and $g_{j,k}(t)$ are defined in the same way as in the fixed-frame approach (49), and the equations of motion arising from (75) are identical to (52). The leading contribution to the expectation values of the spin operators in the original frame is obtained from (66)

$$\langle S_j^\alpha(t)\rangle \approx s R_0^{\alpha z}(t) + \sqrt{2s}\left[R_0^{\alpha x}(t)\mathrm{Re}(m(t)) + R_0^{\alpha y}(t)\mathrm{Im}(m(t))\right] - R_0^{\alpha z}(t)f_{jj}(t) . \tag{77}$$

We note that the second term on the r.h.s. of (77) is in fact $\mathcal{O}(s^0)$ because the mean field $m(t) = \mathcal{O}(s^{-1/2})$ as a consequence of $\lambda(t) = 0$ and our choice of $\theta_0(0)$, $\phi_0(0)$. As expected, the approximation constructed in this way does not conserve energy and we find $\dot{E}_{\mathrm{MF}}(t) = \mathcal{O}(s^{-1/2})$. This remains true if we impose $\langle\tilde{\psi}(t)|\sigma_{0,i}^+|\tilde{\psi}(t)\rangle = 0$ as in Refs [78,79] rather than setting $\lambda(t) = 0$ in (68), thus introducing rotation angles that slightly deviate from the classical ones.

### 2.3.4 A different cubic TDMFT ("Cubic II")

The cubic SCTDMFT discussed above is based on normal ordering with respect to the time evolving Gaussian state $|\tilde{\psi}_{\text{MF}}(t)\rangle$ and dropping normal-ordered terms that are cubic (or higher order) in bosons. By construction this results in time-dependent mean-fields that include terms of all orders in $1/s$. As we will see for the example of the long-range transverse field Ising model below this will generate an approximation that becomes poor at finite times. It turns out that a better approximation is obtained by avoiding resummations to all orders in $1/s$. This can be done in the case of cubic interactions by choosing the rotation angles such that $\langle\tilde{\psi}(t)|\sigma^+_{0,i}|\tilde{\psi}(t)\rangle = 0$ up to a certain order in $1/s$. To do so we expand $\theta(t)$, $\phi(t)$ as

$$\theta(t) = \theta_0(t) + \frac{1}{s}\theta_2(t) + \dots \qquad \phi(t) = \phi_0(t) + \frac{1}{s}\phi_2(t) + \dots \, , \tag{78}$$

from which analogous expansions of $R^\mu(t)$ and $M^{\mu\nu}_{\alpha\beta}(t)$ follow. We then consider the following Gaussian approximation

$$|\tilde{\psi}(t)\rangle \approx |\tilde{\psi}_G(t)\rangle = \mathcal{T}\left[\exp\left(-i\int_0^t ds\,\tilde{h}_2(\theta_0(s),\phi_0(s))\right)\right]|\tilde{\psi}(0)\rangle, \tag{79}$$

where the corrections to the classical angles are generated by the presence of $\tilde{h}_3(t)$ through the equations of motion

$$\dot{\theta}_2(t) = -\sum_{\alpha\leq\beta}\Gamma^{\alpha\beta}M^{yz}_{\alpha\beta|2} + \sum_j\sum_{\alpha\leq\beta}J^{\alpha\beta}_{ij}\left[M^{xz}_{\alpha\beta|0}\text{Im}\left(Q^+_{ij}+P^+_i\right)+M^{yz}_{\alpha\beta|0}\text{Re}\left(Q^-_{ij}+P^-_i\right)\right]$$
$$+\sum_\alpha(h^\alpha+\delta^{\alpha z}\dot{\phi}_0)\left[-R^{\alpha y}_2+R^{\alpha x}_0\text{Im}\left(P^+_i\right)+R^{\alpha y}_0\text{Re}\left(P^-_i\right)\right]+\dot{\theta}_0\text{Re}\left(P^-_i\right)\,,$$

$$\dot{\phi}_2(t) = \frac{1}{R^{zx}_0}\left\{-\sum_{\alpha\leq\beta}\Gamma^{\alpha\beta}M^{xz}_{\alpha\beta|2}+\sum_j\sum_{\alpha\leq\beta}J^{\alpha\beta}_{ij}\left[M^{xz}_{\alpha\beta|0}\text{Re}\left(Q^+_{ij}+P^+_i\right)-M^{yz}_{\alpha\beta|0}\text{Im}\left(Q^-_{ij}+P^-_i\right)\right]\right.$$
$$\left.+\sum_\alpha(h^\alpha+\delta^{\alpha z}\dot{\phi}_0)\left[-R^{\alpha x}_2+R^{\alpha x}_0\text{Re}\left(P^+_i\right)-R^{\alpha y}_0\text{Im}\left(P^-_i\right)\right]-\dot{\theta}_0\text{Im}\left(P^-_i\right)\right\}. \tag{80}$$

Here we have defined

$$Q^\pm_{ij} \equiv \langle\tilde{\psi}_G(t)|a^\dagger_j a_j + a^\dagger_j a_i \pm a_i a_j|\tilde{\psi}_G(t)\rangle\,, \qquad P^\pm_i \equiv \frac{1}{4}\langle\tilde{\psi}_G(t)|2\,a^\dagger_i a_i \pm a_i a_i|\tilde{\psi}_G(t)\rangle. \tag{81}$$

The expectation values of spin operators are approximated by

$$\langle S^\alpha_j\rangle_t \approx sR^{\alpha z}_0(t)+\left[R^{\alpha z}_2(t)-R^{\alpha z}_0(t)\langle\tilde{\psi}_G(t)|a^\dagger_j a_j|\tilde{\psi}_G(t)\rangle\right]\,. \tag{82}$$

If we fix $\theta(0)$, $\phi(0)$ so that the z-axis in rotating-frame z-axis points along the initial direction of magnetic order one can show that (79) and (80) reproduce the leading orders in a "bare" $1/s$ expansion of the exact EOMs for the one and two point functions[4]. This implies that (82) coincides with the exact result for $\langle S^\alpha_j\rangle_t$ up to $\mathcal{O}(s^0)$ and doesn't possess any term beyond this order. Clearly an equivalent result for $\langle S^\alpha_j\rangle_t$ can be obtained also starting from fully classical angles which set $\lambda(t) = 0$.

---

[4]In particular, they generate the exact 1-point functions up to $\mathcal{O}(s^{-1/2})$, which vanish, and the exact 2-point functions up to $\mathcal{O}(s^0)$.

### 2.3.5 Full counting statistics of the subsystem magnetization

For the quench setup considered above there are at least two magnetizations of interest in a subsystem $\mathcal{A}$ with $\ell$ total sites. The first are the magnetizations in the rotating frame. The associated characteristic functions are

$$
\begin{aligned}
\chi_{\mathcal{A}}^{\gamma}(\lambda, t) &= \langle \psi(t) | \exp\Big( i \frac{\lambda}{s\ell} \sum_{i \in \mathcal{A}} \sigma_{t,i}^{\gamma} \Big) | \psi(t) \rangle \\
&\approx \langle \tilde{\psi}_{\mathrm{MF}}(t) | \exp\Big( i \frac{\lambda}{s\ell} \sum_{i \in \mathcal{A}} \sigma_{0,i}^{\gamma} \Big) | \tilde{\psi}_{\mathrm{MF}}(t) \rangle .
\end{aligned}
\tag{83}
$$

If we impose $\langle \tilde{\psi}(t) | \sigma_{0,i}^{+} | \tilde{\psi}(t) \rangle = 0$ in our SCTDMFT the rotating frame follows the average magnetization and $\chi_{\mathcal{A}}^{\gamma}(\lambda, t)$ are then the characteristic functions of the subsystem magnetizations in this frame. For simplicity we focus on

$$
\chi_{\mathcal{A}}^{z}(\lambda, t) \approx \langle \tilde{\psi}_{\mathrm{MF}}(t) | \exp\Big( i \frac{\lambda}{s\ell} \sum_{i \in \mathcal{A}} \big( s - a_i^{\dagger} a_i \big) \Big) | \tilde{\psi}_{\mathrm{MF}}(t) \rangle ,
\tag{84}
$$

which is obtained by direct application of (12). The associated PDF then follows from (6).

The other magnetizations of interest are the ones associated with original spins $S_i^{\gamma}$. Using (66) their characteristic functions are given by

$$
\begin{aligned}
\Pi_{\mathcal{A}}^{\gamma}(\lambda, t) &= \langle \psi(t) | \exp\Big( i \frac{\lambda}{s\ell} \sum_{i \in \mathcal{A}} S_i^{\gamma} \Big) | \psi(t) \rangle \\
&\approx \langle \tilde{\psi}_{\mathrm{MF}}(t) | \exp\Big( i \frac{\lambda}{s\ell} \sum_{i \in \mathcal{A}} \sum_{\mu} R^{\gamma\mu}(t) \sigma_{0,i}^{\mu} \Big) | \tilde{\psi}_{\mathrm{MF}}(t) \rangle .
\end{aligned}
\tag{85}
$$

Here application of (12) is less straightforward, given that the observables contain terms beyond quadratic order in the bosons. One way of proceeding is to truncate the spin operators at second order in $1/s$

$$
\frac{1}{s} \sum_{\mu} R^{\gamma\mu} \sigma_{0,i}^{\mu} \quad \longrightarrow \quad R_0^{\gamma z} + \frac{1}{\sqrt{2s}} \big[ \big( R_0^{\gamma x} - i R_0^{\gamma y} \big) a_i + \text{h.c.} \big] + \frac{1}{s} \big( R_2^{\gamma z} - R_0^{\gamma z} a_i^{\dagger} a_i \big) ,
\tag{86}
$$

where $R_2(t)$ is the first order beyond the classical rotation when we set $\langle \tilde{\psi}(t) | \sigma_{0,i}^{+} | \tilde{\psi}(t) \rangle = 0$. The characteristic function $\Pi_{\mathcal{A},T}^{\gamma}$ obtained in this way possesses a periodicity of $2\pi s \ell / R_0^{\gamma z}(t)$, reflecting the fact that the truncated observable (86) and the original one don't share the same eigenvalues. However, for the specific case of the long-range TFIM we will see in the next section that it is still a very good approximation to calculate the PDF over the truncated observable's spectrum using $\Pi_{\mathcal{A},T}^{\gamma}$ inside (6) and then interpolate the values of the probability over the exact eigenvalues.

### 2.3.6 Long-range transverse field Ising chain

We now apply the rotating-frame formalism to the long-range transverse field Ising chain (33). As in Section 2.2.1 we benchmark the method by comparing it with the exact time-evolution of the $\alpha = 0$ case in a finite-size system, where the large control parameter of the theory is $W = N s$ and quantum fluctuations arise from a global bosonic variable $a$ defined by $\sum_j \sigma_{0,j}^z = W - a^{\dagger} a$. As we are no longer restricted to initial states with magnetic order along a direction close to (38), we start from the fully polarized state along the $x$ direction and retain a finite value of $h$. In Fig. 8 we compare the exact time evolution of $\langle \sum_j S_j^x \rangle / W$ and

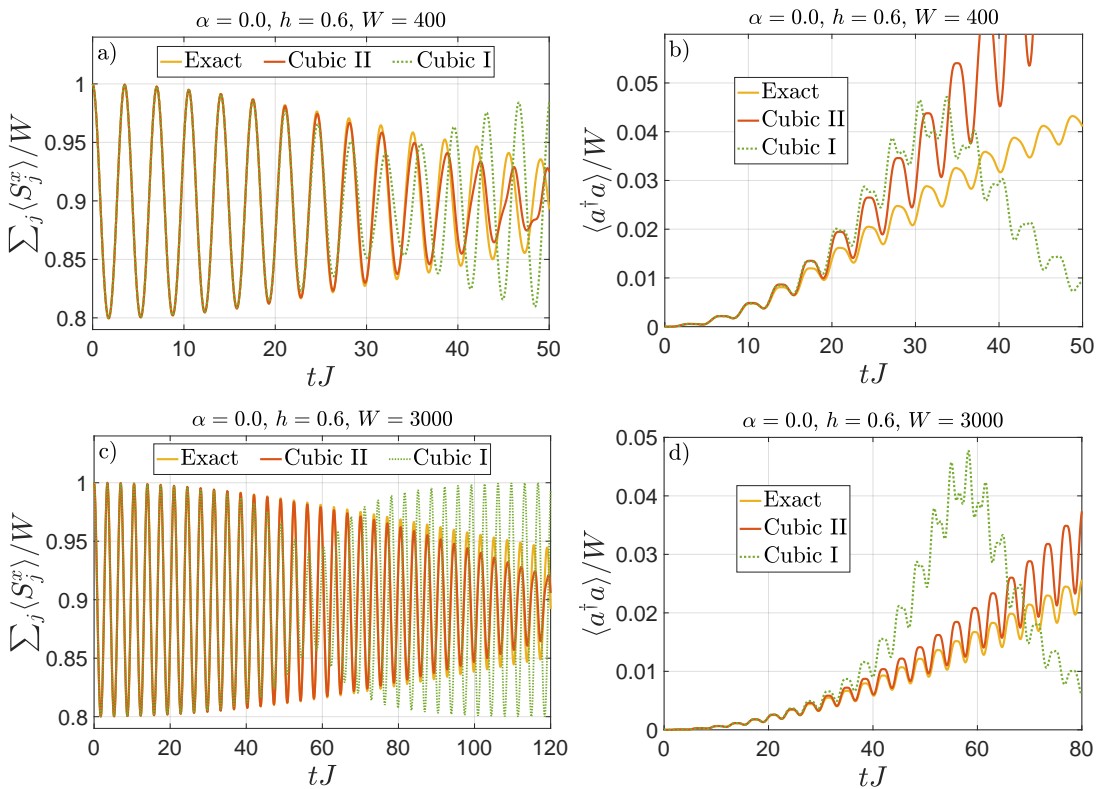

Figure 8: Time evolution of $\langle \sum_j S_j^x \rangle / W$ and $\langle a^\dagger a \rangle$ in the fully connected transverse field Ising model for $h = 0.6$ and $W$ values of 400 in ($a$) and ($b$), 3000 in ($c$) and ($d$). The plots compare exact results with cubic I and II time evolutions.

$|\langle \sum_j \boldsymbol{S}_j \rangle| = W - \langle a^\dagger a \rangle$ in the two cubic SCTDMFTs of subsections 2.3.3 (with rotation angles chosen to impose $\langle \tilde{\psi}(t)| \sigma_{0,i}^+ | \tilde{\psi}(t) \rangle = 0$) and 2.3.4. We refer to these as "cubic I" and "cubic II" respectively. The results of cubic II are very good up to fairly large times $t \sim W^{1/2}$. In contrast, cubic I is accurate only on significantly shorter times[5]. We expect Cubic I and Cubic II to retain comparable levels of accuracy in the entire range $0 < \alpha < 1$, where in the thermodynamic limit the model is still expected to yield a classical time evolution whenever starting from a fully polarized state [75].

In the region $\alpha > 1$ the cubic SCTDMFTs formally require $s \gg 1$ in order to possess a small expansion parameter. However, one might wonder whether the theory provides a fair approximation even for small values of $s$, analogously to what happens in the spin-wave theory of the Heisenberg AFM. To check this we have compared predictions from cubic I and II with the TDVP numerics of [76] in the case of $s = 1/2$ in a chain with open-boundary conditions. All methods turn out to be poor after a very short time scale, thus highlighting some intrinsic limitations of the approach.

In the large-s limit another well-known approach to the problem is to truncate the BBGKY hierarchy of the original spin variables. For the sake of completeness we summarize the simplest variant of this approach in Appendix B. This method has comparable accuracy to cubic II at $\alpha = 0$ (*cfr*. Appendix B) but still fails at very early times when $\alpha > 1$.

The exact diagonalization of the $\alpha = 0$ Hamiltonian in the relevant symmetry sector and associated time evolution provide us directly with the weights that form the probability distribution of any observable involving the total spin $s$. Consider the $\alpha = 0$ system with $s = 1/2$

---

[5]We have verified that the variant of cubic I that sets to zero the linear part of $\tilde{h}(t)$ is equally poor.

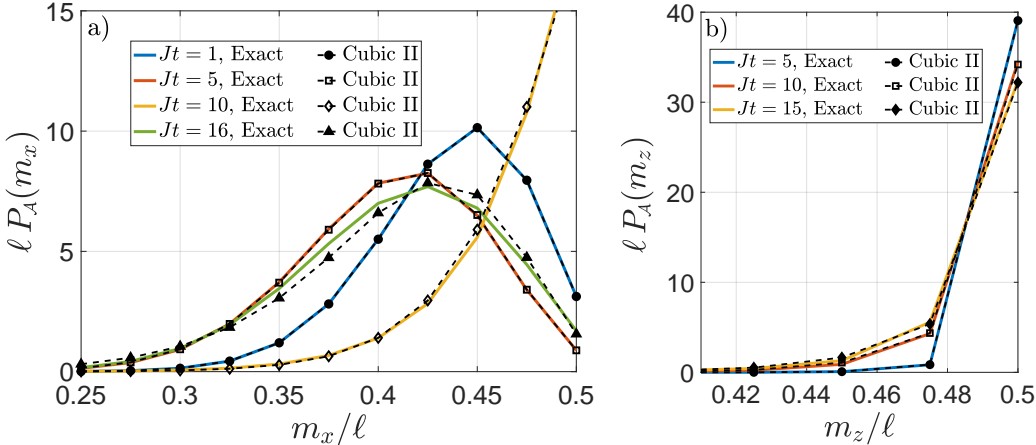

Figure 9: PDFs in the subsystem $\mathcal{A}$ for $s = 1/2$ at different times of (a) the $x$-magnetization $\sum_{i \in \mathcal{A}} S_i^x$ and (b) the rotated-frame $z$-magnetization $\sum_{i \in \mathcal{A}} \sigma_{t,i}^z = \sum_{i \in \mathcal{A}} S_i^\mu R^{\mu z}(t)$. Here $h = 0.6$, $N = 400$ and $\ell = 40$. The plots compare the exact PDFs with the ones obtained by combining the cubic II Gaussian state with the method described in Section 2.3.5.

spin variables and $N$ total sites, i.e. $W = N/2$. Given the probability distribution $P_N$ of any $s^\gamma$ total spin over the time-evolving state, the probability distribution $P_{\mathcal{A}}$ of $\sum_{i \in \mathcal{A}} S_i^\gamma$ over a subset $\mathcal{A}$ with $\ell$ total sites is given by

$$P_{\mathcal{A}}(n) = \sum_{i=1}^{N} \binom{i}{n} P_N(i) \left(\frac{\ell}{N}\right)^n \left(\frac{N-\ell}{N}\right)^{i-n} , \tag{87}$$

where $P_{\mathcal{A}}(i)$, $P_N(i)$ indicate the probability of the configuration possessing $i$ spin-flips compared to the fully ordered state in the direction $\gamma$. In Fig. 9 we compare the exact PDFs of $\sum_{i \in \mathcal{A}} S_i^x$ and $\sum_{i \in \mathcal{A}} \sigma_{t,i}^z = \sum_{i \in \mathcal{A}} S_i^\mu R^{\mu z}(t)$, with the ones obtained combining the cubic II time evolution with the Gaussian method described in the previous section, for a quench governed by the fully connected Hamiltonian with $h = 0.6$ and $N = 400$ (i.e. $W = 200$). We find very good agreement at the level of probability distributions in the time window for which the expectation value of the magnetization computed in spin-wave theory agrees with the exact result, see Fig. 8.

## 3 2D disordered Heisenberg antiferromagnet

So far our discussion focused on cases where our quantum state was characterized by well-formed long-range magnetic order. We now turn to situations where the quantum state of interest is *disordered*. We focus on the full counting statistics of the staggered magnetization in the isotropic 2-dimensional spin-1/2 Heisenberg antiferromagnet in thermal equilibrium and after quantum quenches from disordered initial states.

### 3.1 Schwinger boson mean-field theory (SBMFT)

We begin by considering the spin$-1/2$ Heisenberg antiferromagnet in two dimensions at $T > 0$, i.e. in the disordered phase. The simplest approach to this problem is by means of the Schwinger bosons mean-field theory developed by Arovas and Auerbach in [83], which

we briefly review below. The spin correlation length $\xi$ predicted by this theory is

$$\xi \propto c_1/T \, \exp(c_2/T) \, , \tag{88}$$

where $c_1$, $c_2$ are $O(1)$ positive constants [84]. This functional dependence dependence matches the one-loop renormalization group result for the nonlinear $\sigma$ model, but the $1/T$ pre-factor disagrees with the two-loop RG calculation [85] and Monte Carlo results [39]. Our goal is to assess the ability of this theory to produce a satisfactory approximation for the PDF of the staggered subsystem magnetization, both in and out of equilibrium. We introduce Schwinger bosons (SB) on the "even" (A) and "odd" (B) sublattices by

$$S_i^z = \frac{1}{2} \left( u_i^\dagger u_i - d_i^\dagger d_i \right) \qquad S_i^+ = u_i^\dagger d_i \qquad i \in A \, ,$$
$$S_j^z = \frac{1}{2} \left( d_j^\dagger d_j - u_j^\dagger u_j \right) \qquad S_j^+ = -d_j^\dagger u_j \qquad j \in B \, . \tag{89}$$

The single occupancy constraint for physical states $|\psi\rangle$ reads

$$\hat{C}_i |\psi\rangle = |\psi\rangle \, , \quad \hat{C}_i \equiv u_i^\dagger u_i + d_i^\dagger d_i \, . \tag{90}$$

The operators $\hat{C}_i$ are integrals of motion for any spin Hamiltonian, giving rise to a local $U(1)$ gauge symmetry. The isotropic Heisenberg Hamiltonian is expressed in Schwinger bosons as

$$H = J \sum_{\langle i,j \rangle} \mathbf{S}_i \cdot \mathbf{S}_j = -\frac{J}{2} \sum_{\langle i,j \rangle} \left( A_{ij}^\dagger A_{ij} - \frac{1}{2} \right) \, , \tag{91}$$

where $A_{ij} \equiv u_i u_j + d_i d_j$ and in the last equality the exact constraint $\hat{C}_i = 1$ on the Hilbert space of physical states has been used. The mean-field approximation assumes that

$$\Delta \equiv \langle A_{ij} \rangle = \langle A_{ij}^\dagger \rangle \neq 0 \qquad i, j \text{ nearest neighbours} \, , \tag{92}$$

which explicitly breaks the gauge invariance and involves unphysical boson states with site occupation different from one. To approximately recover (90) one imposes the constraint only on average $\langle \hat{C}_i \rangle = 1$. The unbroken $SU(2)$-invariance implies

$$\langle u_i^\dagger d_j \rangle = \langle u_i d_j \rangle = 0 \qquad i, j \in A \cup B$$
$$\langle u_i^\dagger u_j \rangle = \langle d_i^\dagger d_j \rangle = 0 \qquad \langle u_i u_j \rangle = \langle d_i d_j \rangle \qquad i \in A, \, j \in B \tag{93}$$
$$\langle u_i u_j \rangle = \langle d_i d_j \rangle = 0 \qquad \langle u_i^\dagger u_j \rangle = \langle d_i^\dagger d_j \rangle \qquad i, j \in A \text{ or } i, j \in B \, .$$

Combining the last equation in (93) with $\langle \hat{C}_i \rangle = 1$ leads to $\langle u_i^\dagger u_i \rangle = \langle d_i^\dagger d_i \rangle = 1/2$ for any $i \in A \cup B$. The resulting mean-field Hamiltonian reads

$$H_{MF} = Q \sum_{\langle i,j \rangle} \left( u_i u_j + d_i d_j + \text{h.c.} \right) + \mu \sum_{i \in A \cup B} \left( u_i^\dagger u_i + d_i^\dagger d_i \right) + \left( \frac{4}{J} Q^2 - \mu \right) L^2 \, , \tag{94}$$

where the Lagrange multiplier $\mu$ imposes the constraint on average and $Q \equiv -J\Delta/2$. The mean-field Hamiltonian is diagonalized by going to momentum space and carrying out a Bogoliubov transformation

$$\tilde{u}_k = \cosh \theta_k \, \tilde{\alpha}_k + \sinh \theta_k \, \tilde{\alpha}_{-k}^\dagger \qquad \tilde{d}_k = \cosh \theta_k \, \tilde{\beta}_k + \sinh \theta_k \, \tilde{\beta}_{-k}^\dagger \, ,$$
$$\tanh 2\theta_k = -\frac{4Q\gamma(k)}{\mu} \, , \qquad \qquad \gamma(k) \equiv \frac{1}{2} \left( \cos k_x + \cos k_y \right) \, . \tag{95}$$

This gives

$$H_{MF} = \sum_k \varepsilon_k (\tilde{\alpha}_k^\dagger \tilde{\alpha}_k + \tilde{\beta}_k^\dagger \tilde{\beta}_k) + E_0 \,, \qquad \varepsilon_k = \mu \sqrt{1 - \left(\frac{4Q\,\gamma(k)}{\mu}\right)^2} \qquad \mu > 0 \,, \tag{96}$$

with the requirement $\mu \geq 4Q$. Imposing self-consistency of the expectation values (92) fixes $\mu$ and $Q$ via

$$Q = \frac{2J}{L^2} \sum_k \frac{Q\,\gamma^2(k)}{\varepsilon_k}(1 + 2n_k) \qquad 1 = \frac{\mu}{2L^2} \sum_k \frac{(1 + 2n_k)}{\varepsilon_k} \,, \tag{97}$$

where $n_k$ is the Bose occupation factor associated with the mode of energy $\varepsilon_k$ and inverse temperature $\beta$. Very similar equations are obtained when dropping the assumption that $\Delta$ is real. For $T > 0.91J$ the only solution to the self-consistent equations is the trivial one $Q = 0$ [86]. This reflects the inability of the theory to describe the high-temperature disordered phase of the 2D Heisenberg antiferromagnet, where nearest neighbours correlations are strongly suppressed. At finite temperatures $0 < T < 0.91J$ non-trivial solutions to (97) exist.

### 3.1.1 PDFs in equilibrium

We are now in a position to determine the probability distributions of some observables in SBMFT. It is instructive to consider the PDF of the constraint operator $\hat{C}_i = u_i^\dagger u_i + d_i^\dagger d_i$. Using the identities (93) and the imposition of the constraint on average, the one-site reduced density matrix $\rho_1$ in SBMFT takes the simple temperature-indepedent form

$$\rho_1 = \frac{4}{9} \exp\left[-\log(3)\,\hat{C}_1\right] \,. \tag{98}$$

Using (2) and (6) we then obtain the PDF for $\hat{C}_1$

$$P_{\hat{C}_1}(c) = \int_{-\pi}^{\pi} \frac{d\lambda}{2\pi} e^{-ic\lambda} \frac{4}{(3 - e^{i\lambda})^2} \,. \tag{99}$$

The probability distribution of $\hat{C}_i$ in SBMFT is shown in Fig. 10a. We see that satisfying the constraint only on average introduces a large number of unphysical states in the theory, with the unphysical vacuum of bosons being the most probable state. This already suggests that SBMFT generally cannot be a quantitatively accurate approximation for physical observables. In Fig. 11 we show the PDF of the staggered magnetization $\Sigma_{\mathcal{A}}$ on a disc-shape sub-system $\mathcal{A}$ of $\ell = 80$ total sites computed in SBMFT for $J\beta = 1.59, 2.22$. The PDF is obtained by expressing $\Sigma_{\mathcal{A}}$ in terms of bosons by the SB mapping (89) and applying the general formula (12) for the characteristic function. The geometry of the subsystem is chosen in order to facilitate comparisons with previous numerical and experimental works [9, 20]. The Monte-Carlo results of [20] were obtained for a disc-shape subsystem in a total system of linear size $L = 32$, but for the temperatures and subsystem sizes considered here finite-size effects are expected to be small. For the higher temperature ($\beta = 1.59/J$) shown in Fig. 11a the Monte Carlo results are well described by a Gaussian PDF, which as noted in Section 2.1 is expected to arise asymptotically when the linear subsystem size becomes much larger than the correlation length. At the lower temperature ($\beta = 2.22/J$) shown in Fig. 11b the correlation length is larger and the Monte-Carlo results are no longer well described by a Gaussian. The PDF obtained from SBMFT is seen to be a poor approximation to the Monte-Carlo data. In particular, the tails of the SBMFT distributions decay much slower than the Monte Carlo results and extend beyond the physical thresholds $\Sigma_{\mathcal{A}}/\ell = \pm 1/2$. The disagreement is especially

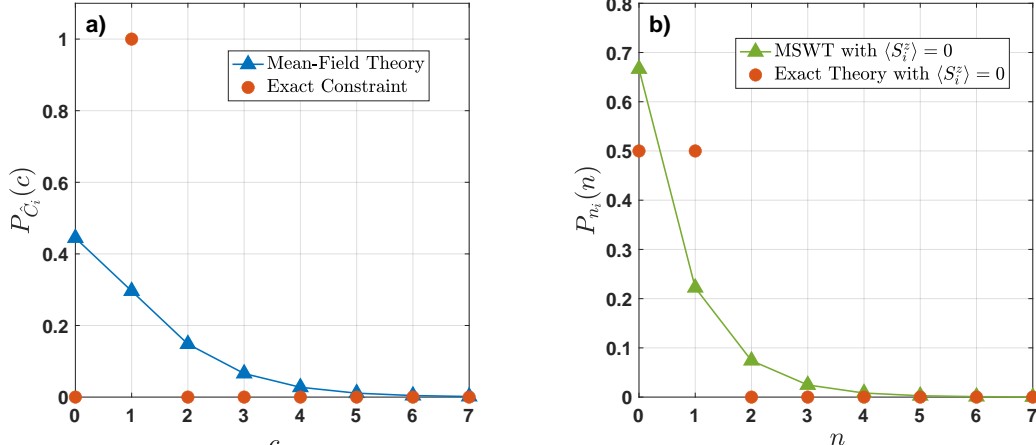

Figure 10: a) PDF of the constraint $\hat{C}_i = u_i^\dagger u_i + d_i^\dagger d_i$ in SBMFT and comparison with the exact constraint. b) PDF from mean-field MSWT (Section 3.2) of the occupation number $\hat{n}_i = a_i^\dagger a_i$ and comparison with the exact distribution of the disordered phase.

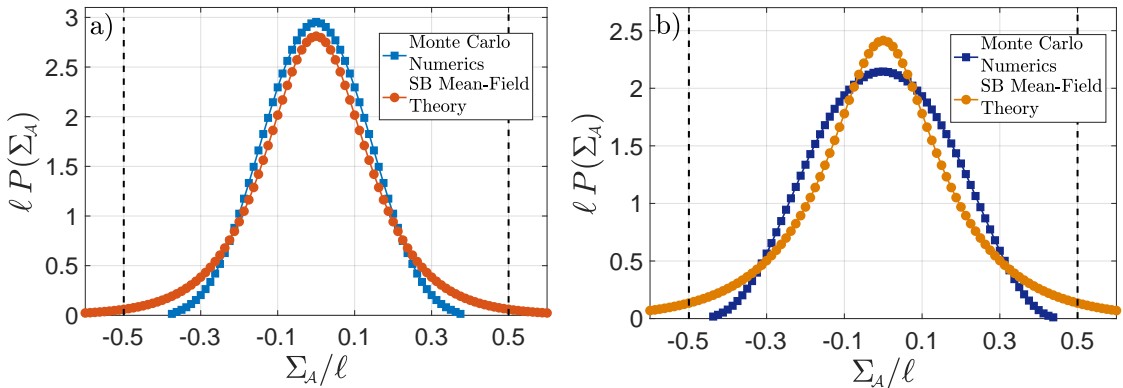

Figure 11: PDF for the staggered magnetization $\Sigma_\mathcal{A}$ in a disc-shape subsystem $\mathcal{A}$ of $\ell = 80$ sites as computed in SBMFT at a) $\beta = 1.59/J$ and b) $\beta = 2.22/J$. Comparison with the exact Monte Carlo results from [20] are shown. The black dashed lines represent the physical thresholds on the eigenvalues of $\Sigma_\mathcal{A}$.

pronounced at lower temperatures where the exact distribution is broader. We remark that given the presence of unphysical states, the eigenvalues of the staggered magnetization over 80 sites expressed using the SB representation (89) include both integer and half-integer numbers. In Fig. 11 we have excluded the unphysical half-integer eigenvalues and renormalized the distribution accordingly. The spirit of this is exactly the same discussed in Section 2.3 for the probability distribution of the $S^x$ magnetization.

### 3.1.2 Non-equilibrium dynamics in SBMFT

We now turn to non-equilibrium dynamics in the SBMFT. More precisely, we aim to generalize SBMFT to study quantum quenches in the SU(2) invariant Heisenberg model starting in initial states $|\psi_i\rangle$ that exhibit no magnetic order. We are particularly interested in situations where $|\psi_i\rangle$ is characterized by a short correlation length and low energy density relative to the ground state. We then expect the correlation length to grow under time evolution and the PDF of the staggered subsystem magnetization (along any direction) to become broader and flatter. In

order to generalize SBMFT to out-of-equilibrium settings we

1. Introduce a complex time-dependent mean field which is determined self-consistently

$$Q(t) = -J \langle \psi_i | U_{\text{MF}}^\dagger(t) u_j u_h U_{\text{MF}}^\dagger(t) | \psi_i \rangle , \quad j, h \text{ nearest neighbour sites}$$

$$= -\frac{J}{L^2} \sum_k \gamma(k) \langle \psi(t) | \tilde{u}_k \tilde{u}_{-k} | \psi(t) \rangle , \tag{100}$$

where $U_{\text{MF}}(t)$ is defined as in (26) and $|\psi(t)\rangle = U_{\text{MF}}(t)|\psi_i\rangle$.

2. Drop the Lagrange multiplier term that imposes the constraint on average. This is possible because when starting from an initial state that satisfies $\langle \hat{C}_i \rangle = 1$, time evolution with a time-dependent SBMFT Hamiltonian automatically ensures that the average constraint is fulfilled at all times.

The resulting time-dependent SBMFT Hamiltonian takes the form

$$H_{\text{MF}}(t) = 2 Q^*(t) \sum_k \gamma(k) \left[ \tilde{u}_k \tilde{u}_{-k} + \tilde{d}_k \tilde{d}_{-k} \right] + \text{h.c.} + \frac{4L^2}{J} |Q(t)|^2 . \tag{101}$$

The equation of motion for the constraint operator $\hat{C}_i$ is

$$\frac{d}{dt} \langle \hat{C}_i \rangle = i \left\langle [H_{\text{MF}}(t), \hat{C}_i] \right\rangle = i \sum_j' \left\langle Q^*(t)(u_i u_j + d_i d_j) - Q(t)(u_i^\dagger u_j^\dagger + d_i^\dagger d_j^\dagger) \right\rangle = 0 , \tag{102}$$

where the primed sum is restricted to the nearest neighbours of site $i$ and the last equality follows from the self-consistent nature of $Q(t)$ and the spatial symmetries of the problem. This justifies point 2. above. It is important to note that the above time-dependent SBMFT is consistent in the sense that the time evolution of physical observables is unaffected if we add a term to $H_{\text{MF}}(t)$ that is proportional to the constraint

$$\Gamma \sum_k \left( \tilde{u}_k^\dagger \tilde{u}_k + \tilde{d}_k^\dagger \tilde{d}_k \right). \tag{103}$$

This can be seen by considering the equations of motion

$$\frac{d}{dt} \langle \tilde{u}_k^\dagger \tilde{u}_k \rangle = -8 \gamma(k) \operatorname{Im} \left[ Q^*(t) \langle \tilde{u}_k \tilde{u}_{-k} \rangle \right], \tag{104}$$

$$\frac{d}{dt} \langle \tilde{u}_k \tilde{u}_{-k} \rangle = -4 i \gamma(k) Q(t) \left( 2 \langle \tilde{u}_k^\dagger \tilde{u}_k \rangle + 1 \right) - 2 i \Gamma \langle \tilde{u}_k \tilde{u}_{-k} \rangle , \tag{105}$$

and similarly for $d$-bosons by replacing $u \to d$. Using (100) we see that the only effect of $\Gamma$ is to change $\langle \tilde{u}_k \tilde{u}_{-k} \rangle$ by a phase factor $\exp(-2 i \Gamma t)$, which however does not affect expectation values of spin operators. Hence we are free to set $\Gamma = 0$. We note that (104) and (105) imply that $|Q(t)|$ is constant in time, which ensures conservation of energy. The equation of motion for its phase $\phi(t)$ is derived from

$$\frac{d}{dt} Q(t) \equiv i Q(t) \dot{\phi}(t) = i Q(t) \left[ \frac{4J}{L^2} \sum_k \left( \gamma(k)^2 \left( 1 + 2 \langle \tilde{u}_k^\dagger \tilde{u}_k \rangle \right) \right) \right] . \tag{106}$$

To solve the Heisenberg equations of motion we still need to specify an initial Gaussian state. We choose the latter to be of the form

$$|\psi_i\rangle = C \exp \left[ \frac{1}{4} \sum_{i \in A, j \in B} \eta_{ij} \left( u_i^\dagger u_j^\dagger + d_i^\dagger d_j^\dagger \right) \right] |0\rangle , \tag{107}$$

where $|0\rangle$ is the vacuum of bosons and $\eta_{ij} := \eta(|\boldsymbol{x}_i - \boldsymbol{x}_j|)$ imposes spatial symmetries. Given that $S_{\text{tot}}^\gamma$ for $\gamma = x, y, z$ commutes with $u_i^\dagger u_j^\dagger + d_i^\dagger d_j^\dagger$ and that every spin-operator annihilates the vacuum of bosons, $|\psi_i\rangle$ is annhilated by all $S_{\text{tot}}^\gamma$, which ensures invariance under $SU(2)$. We choose the parameters $\eta_{ij}$ such that the constraint is satisfied on average and that the initial state is normalizable and possesses a finite correlation length $\xi_i$ for spin-spin correlations. We stress that by satisfying the constraint only on average the Gaussian state (107) is not a physical state, but it is expected to generate a dynamics with features similar to those obtained starting from physical singlet states or $SU(2)$-invariant density matrices with correlation length close to $\xi_i$. At late times the system will thermalize at an effective temperature set by the energy density in the initial state. Spin correlations in this thermal state can approximately be described by SBMFT and will exhibit a finite correlation length $\xi_f$. To clearly exhibit the growth of the spin correlation length under time evolution we adjust the parameters $\eta_{ij}$ in order to achieve $\xi_f \gg \xi_i$. In practice, we consider non vanishing $\eta_{ij}$ for the first seven consecutive classes of nearest-neighbours ($\eta_i$ for $i = 1, ..., 7$) and tune them to maximize the correlation length growth.

Expectation values of operators that do not conserve total boson number exhibit persistent oscillations at a frequency which is determined by the thermal value of $\dot{\phi}(t)$ in (106). In contrast, physical spin observables relax at late times, *cf.* (103)-(105). In Fig. 12a we show results for the time evolution of the (equal-time) spin-spin correlation function $\langle \boldsymbol{S}_i \cdot \boldsymbol{S}_j \rangle$ in time-dependent SBMFT, where $i$ and $j$ are taken to belong to different sub-lattices. As expected we observe a light-cone effect [87,88], and at late times relaxation towards approximately thermal values. It is well-understood that while a simple self-consistent time-dependent mean-field approximation like the one employed here cannot correctly describe thermalization [59, 60, 62, 63, 89], it can describe relaxation to a steady state that is approximately thermal, see e.g. Ref. [64].

In Fig. 12b we show the initial and late time PDF of the staggered magnetization in a rectangular subsystem $\mathcal{A}$ with $\ell = 4 \times 40$ total sites obtained by employing our FCS formula (12) together with (6). This correctly shows the expected broadening arising from the increase in correlation length over time. However, by comparison with the equilibrium Monte Carlo results of Fig. 11, the shape of the distribution is seen to be still poor due to the heavy tails.

## 3.2 Takahashi's modified spin-wave theory (MSWT)

A different approach to the problem of determining equilibrium properties of the 2D Heisenberg antiferromagnet in the disordered phase was proposed by Takahashi [90, 91]. This method yields the same functional form (88) for the temperature dependence of the correlation length. However, as we highlight below, it improves over the SBMFT results by giving less weight to unphysical states. The first step is to employ a Dyson-Maleev (DM) representation of spin operators [92, 93]

$$
\begin{aligned}
S_i^z &= s - a_i^\dagger a_i & S_i^- &= a_i^\dagger & S_i^+ &= (2s - a_i^\dagger a_i) a_i & i &\in A \\
S_j^z &= -s + b_j^\dagger b_j & S_j^- &= b_j & S_j^+ &= b_j^\dagger (2s - b_j^\dagger b_j) & j &\in B \,,
\end{aligned}
\tag{108}
$$

where $a_i$ and $b_j$ are bosonic annihilation operators and $[a_i, b_j] = [a_i, b_j^\dagger] = 0$. This gives the following expression for the $s = 1/2$ Heisenberg Hamiltonian

$$
H = J \sum_{\langle i,j \rangle} \boldsymbol{S}_i \cdot \boldsymbol{S}_j = \frac{J}{2} \sum_{\langle i,j \rangle} \left( a_i^\dagger a_i + b_j^\dagger b_j + a_i b_j + a_i^\dagger b_j^\dagger - a_i^\dagger (a_i + b_j^\dagger)^2 b_j \right) - \frac{J}{2} L^2 \,.
\tag{109}
$$

Note that by construction the DM representation is non-Hermitian with respect to the standard inner product and a constraint on the boson occupation is implicit. Alternatively, one can use

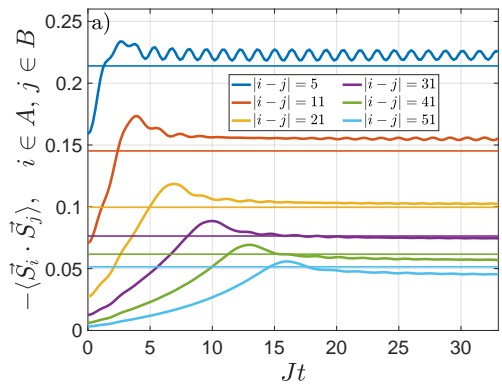 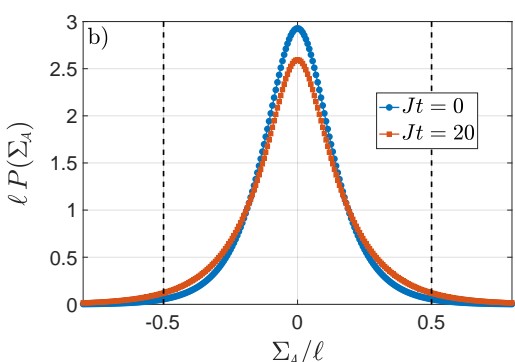

Figure 12: SBMFT time evolution from an initial state of the form (107) with $\eta_1 = 0.35$, $\eta_2 = \eta_3 \sim 0.024$, $\eta_4 \sim 0.012$, $\eta_5 = 0$, $\eta_6 \sim 0.008$, $\eta_7 \sim 0.015$ ($\xi_i \sim 15$, $\xi_f(\beta,\mu) \sim 200$). a) Spin-spin correlation functions $-\langle \boldsymbol{S}_i \cdot \boldsymbol{S}_j \rangle$ (on different sublattices) and comparison with the values obtained by a Gibbs ensemble with values of $\beta$ and $\mu$ set by the initial state (horizontal lines). b) Initial and late time PDF for the staggered magnetization $\Sigma_{\mathcal{A}}$ in a rectangular subsystem $\mathcal{A}$ of size $\ell = 4 \times 40$. The black dashed line represent the physical thresholds on the eigenvalues of $\Sigma_{\mathcal{A}}/\ell$.

a Holstein-Primakoff representation of spin operators and retain only terms involving at most 2n bosons in $H$, as done up to $n = 2$ in Section 2.1. This results in an approximation that we refer to as "MSWT/HP2n" in the following.

The next step is to mean-field decouple the quartic interaction in (109) while ensuring a disordered state with $\langle S_j^\gamma \rangle = 0 \; \forall \gamma$. To obtain $\langle S_j^z \rangle = 0$ Takahashi introduces a chemical potential term in the Hamiltonian that fixes the average occupation numbers $\langle a_i^\dagger a_i \rangle = \langle b_j^\dagger b_j \rangle = 1/2$, while the $z$-rotational invariance[6] ensures $\langle S_j^x \rangle = \langle S_j^y \rangle = 0$, together with

$$\langle a_i^\dagger b_j \rangle = \langle a_i a_{i'} \rangle = \langle b_j b_{j'} \rangle = 0 \qquad \forall \; i, i' \in A, \; j, j' \in B \;. \tag{110}$$

Imposing $\langle a_i^\dagger a_{i+q} \rangle = \langle b_j^\dagger b_{j+q} \rangle$ given the physical equivalence between the two sublattices and $\langle a_i b_j \rangle \in \mathbb{R}$ to obtain an Hermitian theory results in the MSWT/DM Hamiltonian

$$H_{\mathrm{MF}} = -J\Delta \sum_{\langle i,j \rangle} \left( a_i^\dagger a_i + b_j^\dagger b_j + a_i b_j + a_i^\dagger b_j^\dagger \right) - \mu \sum_{i \in A, j \in B} \left( a_i^\dagger a_i + b_j^\dagger b_j \right) + E_{\mathrm{MF}}$$
$$= -(4J\Delta + \mu) \sum_k \left( \tilde{a}_k^\dagger \tilde{a}_k + \tilde{b}_k^\dagger \tilde{b}_k \right) - 4J\Delta \sum_k \gamma(k)(\tilde{a}_k \tilde{b}_{-k} + \tilde{a}_k^\dagger \tilde{b}_{-k}^\dagger) + E_{\mathrm{MF}} \;, \tag{111}$$

where $\Delta \equiv \langle a_i b_j \rangle \in \mathbb{R}$ for $i, j$ nearest neighbours, $E_{\mathrm{MF}}$ is a constant and $\gamma(k)$ is defined as in (22). We note that MSWT/HP4 results formally in the same mean-field Hamiltonian (111) for $\Delta \in \mathbb{R}$, *cf.* (21). We diagonalize (111) by a Bogoliubov transformation

$$\tilde{a}_k = \cosh \theta_k \, \tilde{\alpha}_k - \sinh \theta_k \, \tilde{\beta}_{-k}^\dagger \qquad \tilde{b}_k = \cosh \theta_k \, \tilde{\beta}_k - \sinh \theta_k \, \tilde{\alpha}_{-k}^\dagger$$
$$\tanh 2\theta_k \equiv \eta \, \gamma(k) \qquad \qquad \eta \equiv \frac{4J\Delta}{4J\Delta + \mu} \;, \tag{112}$$

thus obtaining

$$H_{\mathrm{MF}} = \sum_k \varepsilon_k \left( \tilde{\alpha}_k^\dagger \tilde{\alpha}_k + \tilde{\beta}_k^\dagger \tilde{\beta}_k \right) + E_0 \qquad \varepsilon_k = -(4J\Delta + \mu)\sqrt{1 - \eta^2 \gamma(k)^2} \;. \tag{113}$$

---

[6]In terms of bosons, a rotation around the $z$-axis is obtained by $a \to a e^{i\phi}$, $b \to b e^{-i\phi}$.

Here we have $|\eta| < 1$ and $(4J\Delta + \mu) < 0$. In equilibrium at inverse temperature $\beta$ the self-consistency equations for $\Delta$ and $\mu$ read

$$\Delta = -\frac{1}{L^2} \sum_k \frac{\eta\,\gamma(k)^2}{\sqrt{1-\eta^2\gamma(k)^2}}(1+2n_k)\,, \qquad 1 = \frac{1}{L^2}\sum_k \frac{(1+2n_k)}{\sqrt{1-\eta^2\gamma(k)^2}}\,, \tag{114}$$

where $n_k$ is a Bose occupation number for a mode of energy $\varepsilon_k$ at inverse temperature $\beta$. Equations (114) have the same form of (97) and the same considerations regarding the existence of non-trivial solutions and the possibility of promoting $\Delta$ to a complex variable apply.

The mean-field theory derived in this way violates spin rotational symmetry as a result of the particular choice of quantization axis. In order to address this issue Takahashi introduces an explicit rotational averaging [91]. This involves the replacement of expectation values of (products of) spin operators by explicitly spin rotationally invariant ones

$$\langle\Psi|\mathcal{O}|\Psi\rangle \to \overline{\langle\Psi|\mathcal{O}|\Psi\rangle} = \frac{1}{8\pi^2}\int_0^\pi d\theta\,\sin\theta \int_0^{2\pi} d\phi \int_0^{2\pi} d\psi\,\langle\Psi|U^\dagger\,\mathcal{O}\,U|\Psi\rangle\,, \tag{115}$$

where $U$ is a general global $SO(3)$ rotation

$$U(\theta,\phi,\psi) = \exp\left(i\theta\sum_i S_i^y\right)\exp\left(i\phi\sum_i S_i^z\right)\exp\left(i\psi\,\hat{n}\cdot\sum_i \boldsymbol{S}_i\right)\,, \tag{116}$$

and $\hat{n}$ is the direction specified by the $\theta$ and $\phi$ angles. This prescription ensures that for any $SO(3)$ rotation $V$

$$\overline{\langle\Psi|V^\dagger\,\mathcal{O}\,V|\Psi\rangle} = \overline{\langle\Psi|\mathcal{O}|\Psi\rangle}\,, \tag{117}$$

and for $SU(2)$ singlet states $|\Psi_0\rangle$

$$\langle\Psi_0|\mathcal{O}|\Psi_0\rangle = \overline{\langle\Psi_0|\mathcal{O}|\Psi_0\rangle}\,. \tag{118}$$

Some examples of this rotational averaging are

$$\overline{\langle\Psi|S_j^\alpha|\Psi\rangle} = 0\,, \qquad \overline{\langle\Psi|S_j^\alpha S_k^\gamma|\Psi\rangle} = \frac{\delta_{\alpha,\gamma}}{3}\langle\Psi|\boldsymbol{S}_j\cdot\boldsymbol{S}_k|\Psi\rangle\,,$$

$$\overline{\langle\Psi|(S_i^\gamma S_j^\gamma S_k^\gamma S_\ell^\gamma)|\Psi\rangle} = \frac{1}{15}\langle\Psi|\big(\boldsymbol{S}_i\cdot\boldsymbol{S}_j\big)\big(\boldsymbol{S}_k\cdot\boldsymbol{S}_\ell\big)+\big(\boldsymbol{S}_i\cdot\boldsymbol{S}_k\big)\big(\boldsymbol{S}_j\cdot\boldsymbol{S}_\ell\big)+\big(\boldsymbol{S}_i\cdot\boldsymbol{S}_\ell\big)\big(\boldsymbol{S}_j\cdot\boldsymbol{S}_k\big)|\Psi\rangle\,. \tag{119}$$

### 3.2.1 PDFs in equilibrium

We start by considering the PDF for the occupation number $n_i = a_i^\dagger a_i$ (the same will be true for $b_j$) in a thermal state at inverse temperature $\beta$ as this provides a useful diagnostic for the role of unphysical boson states that have been introduced by not treating the constraint exactly. As for the case of the constraint in SBMFT, it is possible to obtain an analytical temperature-independent expression

$$P_{n_i}(n) = \int_{-\pi}^\pi \frac{d\lambda}{2\pi}e^{-in\lambda}\frac{2}{3-e^{i\lambda}}\,. \tag{120}$$

In Fig. 10b we plot $P_{n_i}(n)$ as a function of $n$. While there are significant deviations from the exact PDF, the agreement is significantly better than for SBMFT. This is a first indication that MSWT is a better approximation than SBMFT.

When considering PDFs of the staggered magnetization $\Sigma_{\mathcal{A}}$ (without loss of generality along the $\hat{z}$ direction) in a subsystem $\mathcal{A}$ we encounter a drawback of modified spin-wave theory: the rotational averaging precludes us from applying our FCS formula (12). This is because

| $\overline{\kappa}_2/\ell$ | | | | |
|---|---|---|---|---|
| $\beta$ | Monte Carlo | MSWT/HP6 | MSWT/DM | SBMFT |
| 1.59 | 1.32 | 1.56 | 1.59 | 2.38 |
| 2.22 | 2.19 | 2.51 | 2.71 | 3.89 |

Table 1: 2nd cumulant per site of the staggered magnetization $\Sigma_A$ in the disc-shape subsystem $\mathcal{A}$ with $\ell = 80$ sites in the various mean-field approximations and Monte Carlo results from [20].

$\overline{\langle \exp(i\lambda\Sigma_A) \rangle}$ is not given by the expectation value of a Gaussian bosonic operator. We therefore proceed by computing the first few cumulants of the PDF by brute force, i.e. employing Wick's theorem. The first and third cumulants vanish identically as a result of the rotational averaging. The second cumulant is given by

$$\overline{\kappa}_2 \equiv \overline{\langle \Sigma_A^2 \rangle} = \frac{1}{3} \sum_{i,j \in \mathcal{A}} v_{ij} \langle \mathbf{S}_i \cdot \mathbf{S}_j \rangle \,, \tag{121}$$

where $v_{ij}$ is equal to 1 if $i,j$ are both in the same sublattice and $-1$ otherwise. Expressing the spin operators by the DM representation (108) and using (111) with the condition $\Delta \in \mathbb{R}$ gives

$$\overline{\kappa}_2 = \frac{1}{3} \Big[ 2 \sum_{i,i' \in A} \Big( \langle a_i^\dagger a_{i'} \rangle^2 + \frac{1}{2}\delta_{i,i'} \Big) + 2 \sum_{i \in A,\, j \in B} \langle a_i b_j \rangle^2 \Big] \qquad i,i',j \in \mathcal{A} \,. \tag{122}$$

The same result is obtained by expressing the spins in terms of HP bosons and discarding terms beyond quartic interactions. Thus $\overline{\kappa}_2$ is the same in MSWT/DM and MSWT/HP4. We

| $\overline{\kappa}_4/\ell$ | | | | |
|---|---|---|---|---|
| $\beta$ | Monte Carlo | MSWT/DM | MSWT/HP4 | SBMFT |
| 1.59 | $-51.5$ | $-42.0$ | 243.0 | 748.2 |
| 2.22 | $-240.2$ | $-427.7$ | $-60.8$ | 2059.9 |

Table 2: 4th cumulant per site of the staggered magnetization $\Sigma_A$ in the disc-shape subsystem $\mathcal{A}$ of $\ell = 80$ sites.

explicitly computed $\overline{\kappa}_2$ for the disc-shape subsystem $\mathcal{A}$ of $\ell = 80$ sites and the two temperatures $J\beta = 1.59, 2.22$ already considered above. In Table 1 we report values from MSWT/DM as well as the next order in the HP expansion (MSWT/HP6), obtained by including and mean-field decoupling sextic interactions in the Hamiltonian and taking into account sextic terms arising in (121). We further report the results of Monte-Carlo simulations [20] and the SBMFT approach considered above. We see that the MSWT results are in fair agreement with Monte-Carlo simulations, in contrast to SBMFT.

The 4th cumulant of $\Sigma_A$ is given by

$$\overline{\kappa}_4 \equiv \overline{\langle \Sigma_A^4 \rangle} - 3\overline{\kappa}_2^2 \,, \tag{123}$$

and its evaluation requires Wick decompising a fairly large number of terms, as evident from (119). We thus limit the evaluation of $\overline{\kappa}_4$ to the cases of MSWT/DM and MSWT/HP4, for which we obtain the two different results reported in Table 2, together with the corresponding 4th cumulants from the Monte Carlo simulations and SBMFT. We see that MSWT/DM is better than MSWT/HP4 in this case, and that both are better than SBMFT.

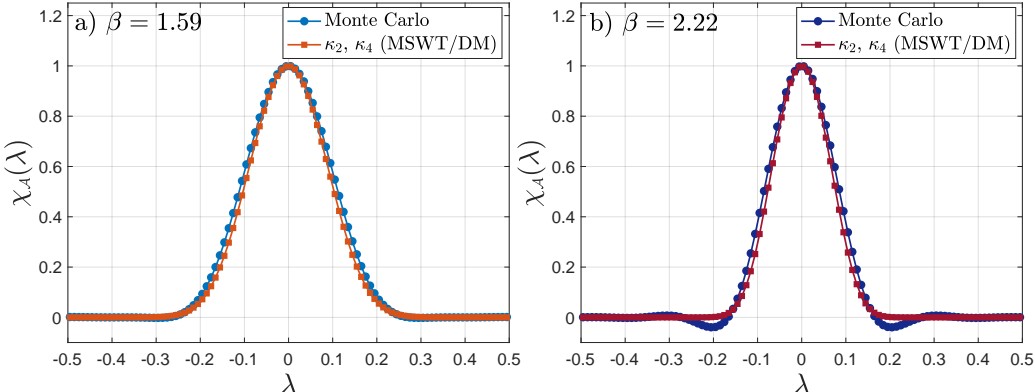

Figure 13: MSWT/DM approximate characteristic functions from (125) in the disc-shape subsystem $\mathcal{A}$ with $\ell = 80$ sites and comparison with $\chi_{\mathcal{A}}(\lambda)$ extracted from the full Monte Carlo PDFs. a) $\beta = 1.59$. b) $\beta = 2.22$.

The cumulants $\overline{\kappa}_n$ are related to the characteristic function by

$$\chi_{\mathcal{A}}(\lambda) = \exp\left[\sum_{n=1}^{\infty} \overline{\kappa}_n \frac{(i\lambda)^n}{n!}\right]. \tag{124}$$

In order to compare with the Monte Carlo simulations of [20] we construct approximate PDFs using our results for the first 4 cumulants and neglecting all $\overline{\kappa}_{n>4}$ by

$$\chi_{\mathcal{A}}(\lambda) \approx \exp\left[-\frac{1}{2}\overline{\kappa}_2 \lambda^2 + \frac{1}{24}\overline{\kappa}_4 \lambda^4\right]. \tag{125}$$

As shown in Fig. 13 the MSWT/DM approximation is in fairly good agreement with the Monte-Carlo results. The approximate PDFs[7] generated by integrating (125) are shown in Fig. 14, with $\overline{\kappa}_2, \overline{\kappa}_4$ from MSWT/DM or extracted directly from the Monte Carlo results. The agreement with the full Monte Carlo PDFs is significantly better than in the SBMFT case. We note that the approximate construction of PDFs just presented works only under the condition $\overline{\kappa}_4 < 0$ and thus cannot be applied to MSWT/HP4 at $\beta = 1.59$, *cf.* Table 2.

### 3.2.2 Non-equilibrium dynamics in MSWT

We now generalize MSWT to non-equilibrium dynamics. We consider the same class of quantum quenches as in Section 3.1.2 above. Given that the DM Hamiltonian (109) is non-Hermitian we define the Heisenberg-picture time evolution from an initial state $|\psi_i\rangle$ following [94]

$$\langle\psi_i|e^{iHt}\,O\,e^{-iHt}|\psi_i\rangle. \tag{126}$$

Applying the normal ordering prescription of Appendix A to (126) leads to the following time-dependent mean-field Hamiltonian

$$
\begin{aligned}
H_{\mathrm{MF}}(t) = {}&-4J\sum_k \left(\Delta(t)\,\tilde{a}_k^\dagger \tilde{a}_k + \overline{\Delta}(t)\,\tilde{b}_k^\dagger \tilde{b}_k\right) \\
&-4J\sum_k \gamma(k)\left(\overline{\Delta}(t)\,\tilde{a}_k \tilde{b}_{-k} + \Delta(t)\,\tilde{a}_k^\dagger \tilde{b}_{-k}^\dagger\right) + E_{\mathrm{MF}}(t).
\end{aligned}
\tag{127}
$$

---

[7]The PDFs obtained assuming $\overline{\kappa}_n = 0$ for $n > 4$ contain in general some negative values and are not properly normalized, but these effects are negligible in the examples considered.

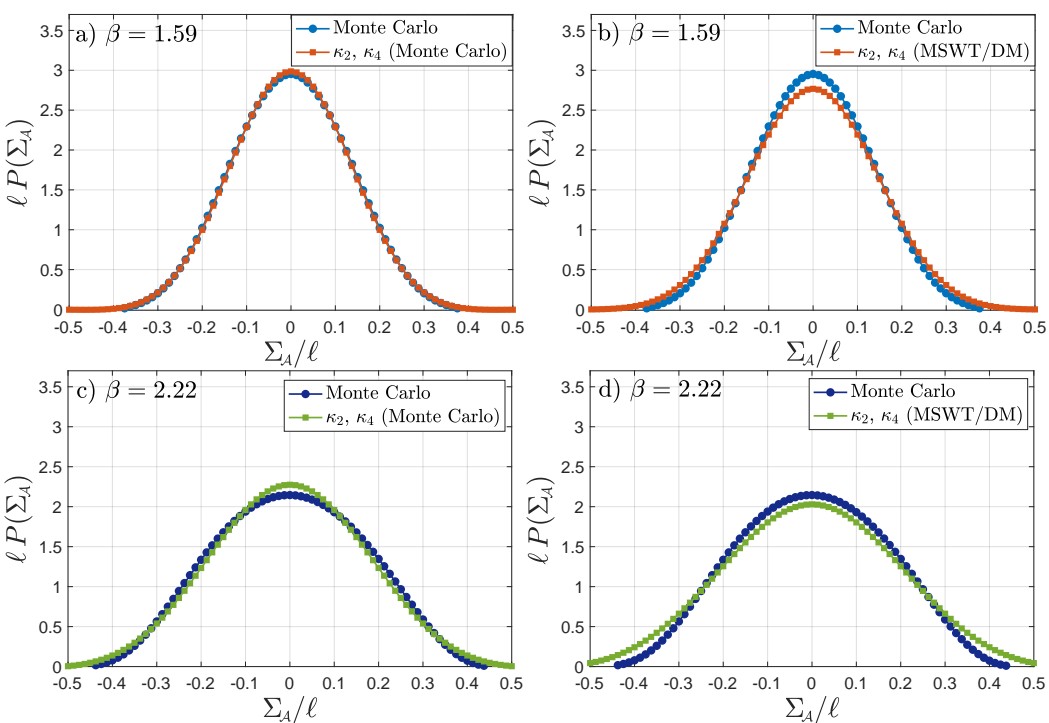

Figure 14: Staggered magnetization PDFs in the disc-shape subsystem $\mathcal{A}$ with $\ell = 80$ sites as reconstructed from 2nd and 4th cumulants using (125), and comparison with full Monte Carlo PDFs. a) $\beta = 1.59$ with $\overline{\kappa}_2, \overline{\kappa}_4$ extracted from full Monte Carlo PDFs. b) $\beta = 1.59$ with $\overline{\kappa}_2, \overline{\kappa}_4$ from MSWT/DM. c) $\beta = 2.22$ with $\overline{\kappa}_2, \overline{\kappa}_4$ extracted from full Monte Carlo PDFs. d) $\beta = 2.22$ with $\overline{\kappa}_2, \overline{\kappa}_4$ from MSWT/DM.

Here we have defined two complex mean fields by

$$\langle O \rangle_t := \langle \psi_i | U_{MF}(t)^{-1} \, O \, U_{MF}(t) | \psi_i \rangle \,, \qquad U_{\mathrm{MF}}(t) = \mathcal{T}\left[ e^{-\int_0^t ds \, H_{\mathrm{MF}}(s)} \right] \,,$$
$$\Delta(t) = \langle a_i b_j \rangle_t \,, \quad \overline{\Delta}(t) = \langle a_i^\dagger b_j^\dagger \rangle_t \,, \quad i,j \text{ nearest neighbours.} \tag{128}$$

In deriving the expression (127) we have already made use of $\langle a_i^\dagger a_i \rangle_t = \langle b_i^\dagger b_i \rangle_t = 1/2 \; \forall t$. Indeed, like in the time-dependent SBMFT, there is no need to introduce a Lagrange multiplier to impose the average constraint at all times, as it is automatically satisfied if we start from an initial state in which it holds. The EOMs for the momentum space space 2-point functions derived from (127), (128) are

$$\frac{d}{dt}\langle \tilde{a}_k \tilde{b}_{-k} \rangle_t = i4J\left[ (\Delta + \overline{\Delta})\langle \tilde{a}_k \tilde{b}_{-k} \rangle_t + \Delta\,\gamma(k)\big( \langle \tilde{a}_k^\dagger \tilde{a}_k \rangle_t + \langle \tilde{b}_{-k}^\dagger \tilde{b}_{-k} \rangle_t + 1 \big) \right] \,,$$
$$\frac{d}{dt}\langle \tilde{a}_k^\dagger \tilde{b}_{-k}^\dagger \rangle_t = -i4J\left[ (\Delta + \overline{\Delta})\langle \tilde{a}_k^\dagger \tilde{b}_{-k}^\dagger \rangle_t + \overline{\Delta}\,\gamma(k)\big( \langle \tilde{a}_k^\dagger \tilde{a}_k \rangle_t + \langle \tilde{b}_{-k}^\dagger \tilde{b}_{-k} \rangle_t + 1 \big) \right] \,,$$
$$\frac{d}{dt}\langle \tilde{a}_k^\dagger \tilde{a}_k \rangle_t = i4J\gamma(k)\left[ \Delta\langle \tilde{a}_k^\dagger \tilde{b}_{-k}^\dagger \rangle_t - \overline{\Delta}\langle \tilde{a}_k \tilde{b}_{-k} \rangle_t \right] = \frac{d}{dt}\langle \tilde{b}_{-k}^\dagger \tilde{b}_{-k} \rangle_t \,. \tag{129}$$

The structure of (129) ensures that the initial conditions $\overline{\Delta}(0) = \Delta(0)^*$, $\langle \tilde{a}_k \tilde{b}_{-k} \rangle_0^* = \langle \tilde{a}_k^\dagger \tilde{b}_{-k}^\dagger \rangle_0$ and $\langle \tilde{a}_k^\dagger \tilde{a}_k \rangle_0$, $\langle \tilde{b}_k^\dagger \tilde{b}_k \rangle_0 \in \mathbb{R}$ remain valid at all times. Thus at the level of 2-point functions[8], the non-Hermitian time evolution defined by (128) preserves the relations expected from an

---

[8]This is in contrast to the EOMs for 1-point functions, which expose the non-unitary nature of $U_{\mathrm{MF}}(t)$ in (128). However, all 1-point functions vanish due to the $z$-rotational invariance of the DM and HP4 Hamiltonian.

Hermitian theory. Indeed, one can verify that by starting from the Hermitian HP4 Hamiltonian and performing the usual SCTDMFT decoupling of Appendix A, one arrives exactly at the EOMs (129) with the identification $\overline{\Delta}(t) \equiv \Delta^*(t)$.

From (129) we easily derive that $|\Delta|$ is a conserved quantity, and this ensures conservation of energy. The equation of motion for the phase $\phi$ of $\Delta$ is

$$\frac{d\phi}{dt} = 8J \left[ \text{Re}(\Delta) + \frac{1}{L^2} \sum_k \gamma(k)^2 \left( \langle \tilde{a}_k^\dagger \tilde{a}_k \rangle + \langle \tilde{b}_{-k}^\dagger \tilde{b}_{-k} \rangle + 1 \right) \right] . \qquad (130)$$

The initial state is determined similarly to the SBMFT case. By requiring it to respect the $\hat{z}$ rotational invariance and the spatial symmetries of the Hamiltonian, we are constrained to the functional form

$$|\psi_i\rangle = C \exp \left[ -\frac{1}{2} \sum_{i \in A, j \in B} \eta_{ij} a_i^\dagger b_j^\dagger \right] |0\rangle , \qquad (131)$$

where $|0\rangle$ is the vacuum of bosons and $\eta_{ij} := \eta(|\boldsymbol{x}_i - \boldsymbol{x}_j|)$ in order to enforce the spatial symmetries. We note that even though (131) breaks $SU(2)$ invariance, the latter is recovered after applying the averaging procedure (115). The $\eta_{ij}$ are chosen by requiring the state to be normalizable, to possess an average bosonic occupation on a site equal to $1/2$ and to have initial energy density low enough to ensure $\xi_i \ll \xi_f$, where $\xi_i, \xi_f$ are respectively the initial and final correlations lengths. As before, we have considered as non-vanishing only the first seven consecutive classes of nearest neighbours $\eta_i$ $i = 1, ..., 7$.

An important difference compared to the time-evolution in SBMFT is that here all bosonic operators relax at late times, as can be seen e.g. in the inset in Fig. 16, which shows relaxation of the imaginary part of the anomalous correlator $\Delta = \langle a_i b_j \rangle$ with $i, j$ nearest-neighbours. This is expected because $\langle a_i b_j \rangle$ represents the leading order in the $1/s$-expansion of the two-point function of spins $\langle S_i^+ S_j^- \rangle$, which must relax at late times. In Fig. 15 we plot the time evolution, starting from (131), of $SU(2)$-invariant 2-point functions of spins and their approximate thermalization. Note that expressing the operators $\boldsymbol{S}_i \cdot \boldsymbol{S}_j$ in terms of bosons and using Wick's contractions to compute their expectation value in the SCTDMFT yields identical results for the DM and HP representations. The same is *not* true for $2n$-point spin-functions with $n \geq 2$. As expected, correlators between spins on different sublattices possess antiferromagnetic nature, as opposed to those within the same sublattice. Fig. 16 proves that the time evolution for the 2-point spin functions agrees in functional form with the Lieb-Robinson bound for correlation functions [87, 88].

### 3.2.3 Non-equilibrium FCS in MSWT

In Fig. 17 we show the DM time evolution, starting in two different initial states, of the 2nd and 4th cumulant of the staggered magnetization in a square subsystem $\mathcal{A}$ with $\ell = 30 \times 30$ sites. We compare the late time values with the DM thermal values at appropriate $\beta$ and $\mu$. The DM time evolution of the 2nd cumulant coincides with the HP4 result, but this is not true for the 4th cumulant. We have determined the HP4 time evolution of the 4th cumulant and noticed that it results in a poorer agreement between stationary values and thermal ones. This is perhaps not surprising given that HP4 represents a truncation of the exact HP mapping, while the DM mapping requires no truncation. One may expect that considering higher truncations HP2$n$ with $n \geq 3$ would lead to results closer to DM. As seen in Fig. 17 both the 2nd and 4th cumulants increase in absolute value after the quench. This is a consequence of the fact that the final correlation length is larger than the initial one and thus antiferromagnetic correlations grow in strength. We also notice that the agreement between stationary values and thermal

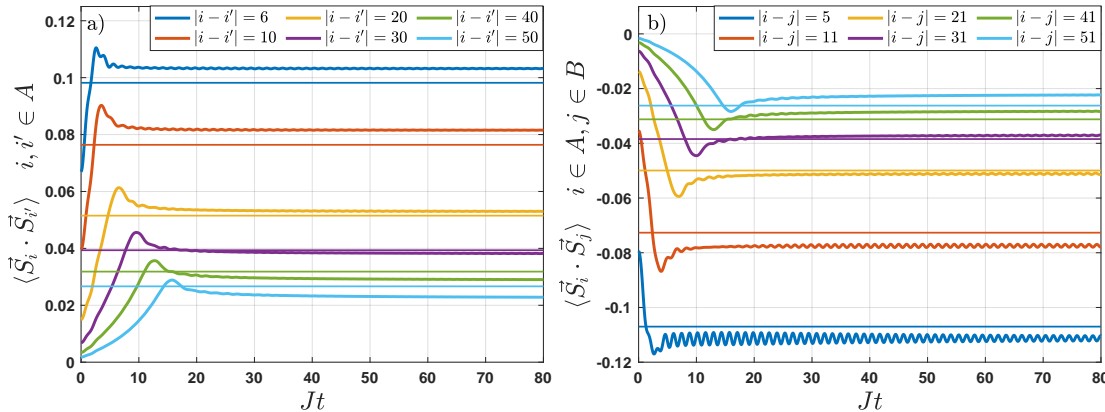

Figure 15: Time evolution of 2-point spin correlation functions in MSWT/DM–HP4 starting from the initial state (131) with $\eta_1 = 0.35$, $\eta_2 = \eta_3 \sim 0.024$, $\eta_4 \sim 0.012$, $\eta_5 = 0$, $\eta_6 \sim 0.008$, $\eta_7 \sim 0.015$ ($\xi_i \sim 20$, $\xi_f(\beta, \mu) \sim 200$), and comparison with thermal values at appropriate $\beta$ and $\mu$. a) Same sublattice. b) Different sublattices.

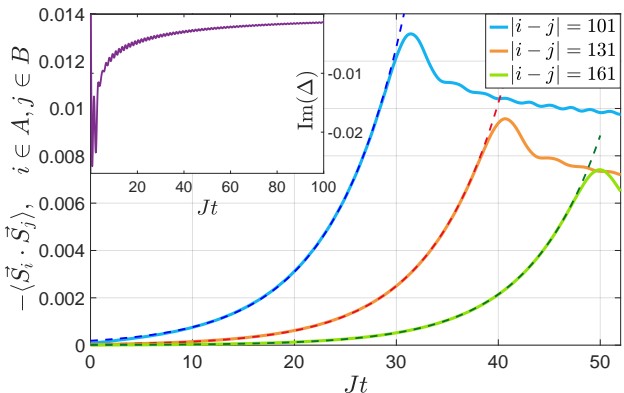

Figure 16: Light cone effect in MSWT/DM–HP4 time-evolution as seen from the spin-spin correlation functions and exponential fit $a \cdot \exp\left(-(L - 2vt)/b\right)$ predicted by the Lieb-Robinson bound. Inset: relaxation of $\text{Im}(\Delta)$ at late times. Same initial state of Fig. 15.

ones is better for the quench possessing a larger final correlation length. This is expected as by construction MSWT works better at low energy densities.

Given that $\overline{\kappa}_4 < 0$ during the whole time evolution, we can generate approximate PDFs obtained from the knowledge of $\overline{\kappa}_2$ and $\overline{\kappa}_4$ as discussed before. Results are shown in Fig. 18. As expected the PDFs broaden in time and eventually approach the appropriate thermal values. The agreement with the latter is better the smaller the energy density is. Moreover, the shapes of the approximate PDFs are closer to what one might expect after a quench in which the correlation length strongly increases than those obtained in SBMFT.

## 4 Summary and Conclusions

In this work we have considered the problem of computing the quantum mechanical PDFs of observables defined on subsystems in lattice models of quantum spins both in thermal equilibrium and after global quantum quenches. We have focused on situations in which the physi-

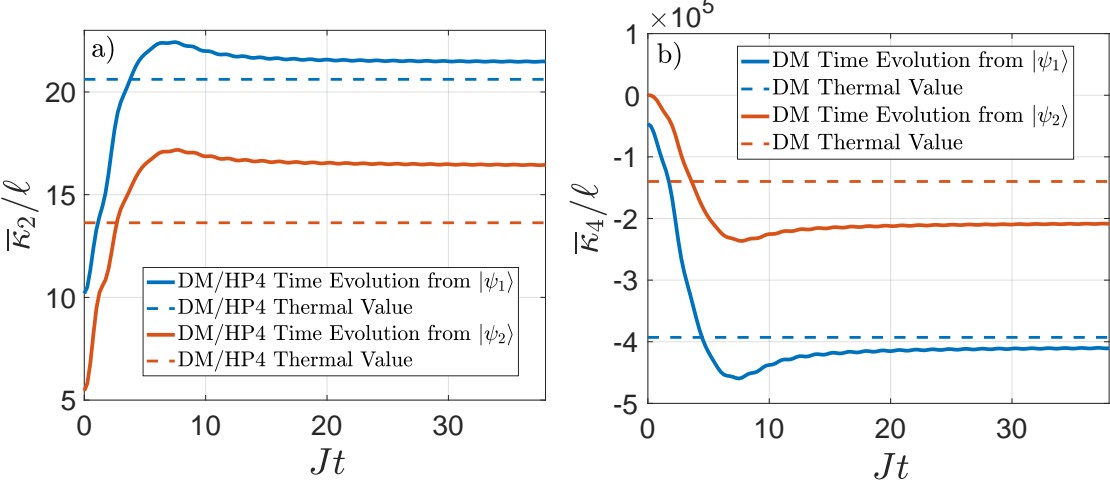

Figure 17: Time evolution of DM cumulants starting from the state $|\psi_1\rangle$ specified by the $\eta_i$ of Fig. 15 and the state $|\psi_2\rangle$ specified by $\eta_1 \sim 0.386$, $\eta_2 \sim 0.017$, $\eta_3 \sim 0.012$, $\eta_4 \sim 0.007$, $\eta_5 \sim 0.001$, $\eta_6 \sim 0.0$, $\eta_7 \sim 0.025$ ($\xi_i \sim 10$, $\xi_f(\beta,\mu) \sim 45$). The subsystem $\mathcal{A}$ is a square with $\ell = 30 \times 30$ sites. Comparison with thermal values is shown. a) 2nd cumulant $\overline{\kappa}_2(t)$. b) 4th cumulant $\overline{\kappa}_4(t)$.

cal properties are well described by Gaussian fluctuations around some appropriately defined mean fields. In situations characterised by the presence of magnetic order we have employed the Holstein-Primakoff representation of spin operators in conjunction with a self-consistent time-dependent mean-field approximation. In absence of magnetic order we have instead used non-equilibrium versions of Schwinger-boson mean field theory and Takahashi's modified spin-wave theory based on the Dyson-Maleev representation, to obtain Gaussian approximations of the spin dynamics. We then determined the PDFs of the subsystem (staggered) magnetizations in the framework of these Gaussian theories. This was achieved by means of the determinant representation (12) for the corresponding characteristic functions, which provides an efficient way for computing the PDFs numerically. Our main findings are as follows. (i) In the ordered phase in equilibrium of the antiferromagnetic Heisenberg model ($T = 0$ in $d = 2$ and $T < T_c$ in $d = 3$) the PDF of the subsystem order parameter at intermediate subsystem sizes is strongly skewed and can be reasonably well fitted to extreme value distributions. The time evolution of the subsystem order parameter in the 3D Heisenberg model after a quench from a Néel-ordered state at an energy density below the one corresponding to $T_c$ relaxes to the thermal value in an oscillating fashion, as shown in e.g. Fig. 5a, 5b. A characteristic feature of these quenches is that the direction of the order parameter is fixed throughout the time evolution. (ii) In order to investigate situations where the direction of the order parameter changes during the thermalization process after a quantum quench we have considered the transverse-field spin-$s$ Ising chain with long-range interactions, which ensure the existence of long-range magnetic order at low energy densities even though the model is one dimensional. We have shown that the dynamics after quenches from initial states characterised by order parameters at small angles $(\theta_0, \phi_0)$ relative to the order parameter in the stationary states reached at late times can be obtained by a standard self-consistent time-dependent mean field (Gaussian) approximation, even if $s$ is small. In order to address the case of large $(\theta_0, \phi_0)$ we have considered spin-wave expansions in a time-dependent frame along the lines of Refs [78, 79]. We have shown that these are quantitatively accurate only for large $s$ and sufficiently short times, and under these restrictions we have computed the time evolution of the subsystem order parameter af-

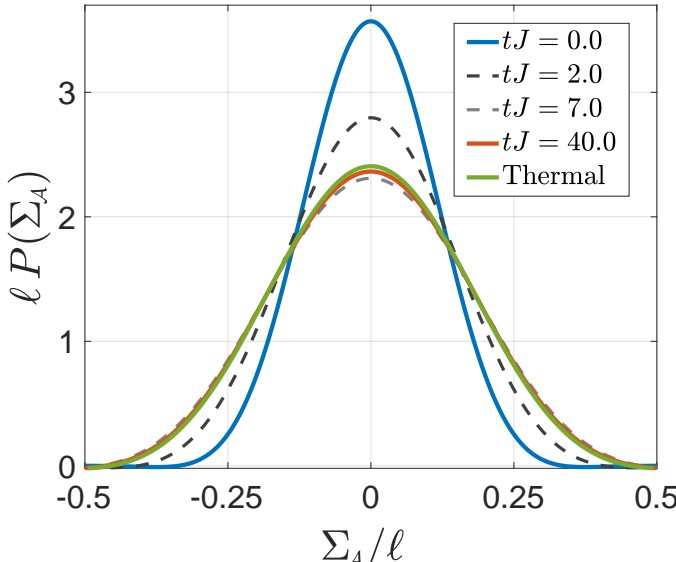

Figure 18: Approximate PDFs of $\Sigma_{\mathcal{A}}$ constructed from the DM 2nd and 4th cumulants at different times, in the time evolution from initial state specified by the $\eta_i$ of Fig. 15. The subsystem $\mathcal{A}$ is the $30 \times 30$ square of Fig. 17. Comparison with the PDF constructed from the DM thermal $\overline{\kappa}_2$ and $\overline{\kappa}_4$ is shown.

ter a quantum quench. (iii) Finally, we have considered the case of the 2D antiferromagnetic Heisenberg model at finite temperature and after quantum quenches. As long-range magnetic order is absent in these cases we have resorted to Schwinger-boson mean field theory and Takahashi's modified spin-wave theory in order to obtain Gaussian approximations in terms of lattice bosons. We found that modified spin-wave theory appears to provide a more accurate approximation both in thermal equilibrium and after quantum quenches. In the quenches we consider, the correlation length grows in time and the PDF of the staggered magnetization becomes flatter.

Overall, the PDFs of subsystem (staggered) magnetizations in the Heisenberg and Ising models we investigated exhibit rich behaviours after quantum quenches. It would be interesting to investigate these experimentally, e.g. in the setup of [9].

## Acknowledgements

This work was supported by the EPSRC under grant EP/S020527/1 (FHLE). We are grateful to E. Demler for helpful comments.

## A Self-consistent time-dependent mean-field theory (SCTDMFT)

Consider a generic many-body Hamiltonian $H$ written in terms of bosonic creation and annihilation operators $\boldsymbol{a}^{\dagger} = (a_1^{\dagger}, \ldots, a_n^{\dagger}, a_1, \ldots, a_n)$. We are interested in studying the time evolution of a Gaussian bosonic state $|\psi(0)\rangle$ under $H$. In general, the presence of interacting terms in the Hamiltonian turn $|\psi(t)\rangle$ into an entangled non-Gaussian state at any $t > 0$, and no exact solution for the dynamics can be determined. To approximately derive the time evolution we employ a self-consistent time-dependent mean-field theory (SCTDMFT) approach [30,53–65],

where $H$ is replaced with a quadratic time-dependent Hamiltonian $H_{\mathrm{MF}}(t)$ that preserves the Gaussianity of the state

$$|\psi_{\mathrm{MF}}(t)\rangle = U_{\mathrm{MF}}(t)|\psi(0)\rangle = \mathcal{T}\left[\exp\left(-i\int_0^t dt' H_{\mathrm{MF}}(t')\right)\right]|\psi(0)\rangle . \tag{132}$$

Given the Gaussian nature of $|\psi(t)\rangle_{\mathrm{MF}}$ it can be written in the form

$$|\psi_{\mathrm{MF}}(t)\rangle = Q(t)|0\rangle \qquad Q(t) = C\exp\left[\frac{1}{2}\boldsymbol{a}^\dagger\Lambda(t)\boldsymbol{a} + \boldsymbol{\lambda}(t)\cdot\boldsymbol{a}\right] , \tag{133}$$

where $|0\rangle$ is the boson vacuum. This shows that $|\psi_{\mathrm{MF}}(t)\rangle$ is a "vacuum" for the operators

$$d_i(t) = Q(t)a_i Q(t)^{-1} . \tag{134}$$

Given the form of $Q(t)$, $d_i(t)$ is a linear combination of operators $a$, $a^\dagger$, plus a constant. This also implies $[d_i(t), d_j(t)] = 0$, $[d_i(t), d_j^\dagger(t)] \in \mathbb{C}$. Inverting (134) we get

$$a_i = M_{ij}(t)d_j(t) + N_{ij}(t)d_j^\dagger(t) + \langle\psi_{\mathrm{MF}}(t)|a_i|\psi_{\mathrm{MF}}(t)\rangle , \tag{135}$$

where $M$, $N$ are matrices. We can now express $H$ as a function of the $d(t)$ bosons and apply Wick's theorem to efficiently write $H$ only in terms of normal ordered operators with respect to $|\psi_{\mathrm{MF}}(t)\rangle$. We split the result of this operation in the two contributions

$$H = N[H_{\geq 3}(t)] + H_{\mathrm{MF}}(t) , \tag{136}$$

where $H_{\geq 3}(t)$ indicates terms in $H$ that contain 3 or more $d(t)$, $d^\dagger(t)$ and $N$ is the normal ordering operator. Equations (132) and (136) self-consistently define $H_{\mathrm{MF}}(t)$, which generally takes a simple form when re-expressed back in terms of $a$, $a^\dagger$ and their 1 and 2-point functions. Using normal ordering arguments, it is easy to see from (136) that

1. Energy is conserved in the sense that

$$\frac{d}{dt}\langle\psi_{\mathrm{MF}}(t)|H|\psi_{\mathrm{MF}}(t)\rangle = i\langle\psi_{\mathrm{MF}}(t)|[H_{\mathrm{MF}}(t), H]|\psi_{\mathrm{MF}}(t)\rangle = 0 . \tag{137}$$

2. The equations of motion (EOMs) of 1 and 2-point functions of bosons, from which every expectation value can be obtained by Wick's contractions, are given by

$$\frac{d}{dt}\langle\psi_{\mathrm{MF}}(t)|O|\psi_{\mathrm{MF}}(t)\rangle = i\langle\psi_{\mathrm{MF}}(t)|[H_{\mathrm{MF}}(t), O]|\psi_{\mathrm{MF}}(t)\rangle = i\langle\psi_{\mathrm{MF}}(t)|[H, O]|\psi_{\mathrm{MF}}(t)\rangle , \tag{138}$$

   where $O$ contains one or two $a$, $a^\dagger$ operators. The last equality makes apparent that the SCTDMFT truncates the infinite BBGKY hierarchy of EOMs [56,59,60] by considering the expectation value of the exact term $[H, O]$ over a Gaussian state. This allows to express the result only in terms of Wick's contractions, thus closing the otherwise infinite set of coupled differential equations in the hierarchy.

Importantly, there always is an early time window in which the SCTDMFT is expected to be quantitatively accurate. This follows from the fact that the EOMs in (138) neglect the contribution of connected $n$-point functions of bosons for $n > 2$, which however vanish at $t = 0$ because $|\psi(0)\rangle$ is Gaussian. In many cases of interest, like when interaction strengths are small, the growth of the connected $n$-point functions is expected to be slow and the SCTDMFT is quantitatively accurate up to parametrically large times.

## B Truncation of the equations of motion hierarchy in the rotating frame

In Section 2.3 the truncation of the BBGKY hierarchy of EOMs in a large class of spin-models was performed in terms of bosonic HP variables by employing the SCTDMFT. In Section 2.3.6 we have explicitly applied this method to the case of the long-range transverse field Ising chain (LRTFIC). In this appendix we describe an alternative approximate description of the time-evolution in the LRTFIC, which can can be straightforwardly generalized to the whole class of models of Section 2.3. The basic requirement is to consider only initial states that exhibit vanishing or very small connected correlations functions of spins.

Our starting point is the rotating-frame formalism of Section 2.3 based on the Hamiltonian $\tilde{h}(t)$ of (62) and the requirement (64) which defines the rotation angles $\theta(t)$ and $\phi(t)$. Here we discuss directly the more general non-translationally invariant case and thus write the LRTFIC $\tilde{h}(t)$ as

$$
\begin{aligned}
\tilde{h}(t) = & -\frac{1}{2s} \sum_{i,j} \sum_{\mu,\nu} J_{ij} R_i^{x\mu}(t) R_j^{x\nu}(t) \sigma_{0,i}^{\mu} \sigma_{0,j}^{\nu} \\
& - \sum_i \sum_{\mu} \left[ h_i + \dot{\phi}_i(t) \right] R_i^{z\mu}(t) \sigma_{0,i}^{\mu} - \sum_i \dot{\theta}_i(t) \sigma_{0,i}^{y} ,
\end{aligned}
\tag{139}
$$

where $J_{ij}$ have been defined in (34). The site-dependence of the angles $\theta$, $\phi$ and of the matrix $R$ is absent in presence of translational invariance. In the following we use notations such that $\langle O \rangle$ denotes the expectation value of an operator $O$ in the state $|\tilde{\psi}(t)\rangle = \tilde{U}(t)|\psi_i\rangle$ defined in (63). The equation of motion for the one- and two-point functions of spins are

$$
\frac{d}{dt} \langle \sigma_{0,i}^{\alpha} \rangle = \sum_j \sum_{\nu,\gamma} T_{ij}^{\nu\gamma}(\alpha) \langle \sigma_{0,i}^{\gamma} \sigma_{0,j}^{\nu} + \sigma_{0,j}^{\nu} \sigma_{0,i}^{\gamma} \rangle + \sum_{\gamma} P_i^{\gamma}(\alpha) \langle \sigma_{0,i}^{\gamma} \rangle
\tag{140}
$$

$$
\frac{d}{dt} \langle \sigma_{0,i}^{\alpha} \sigma_{0,j}^{\beta} \rangle = i \langle \sigma_{0,i}^{\alpha} \left[ \tilde{h}(t), \sigma_{0,j}^{\beta} \right] \rangle + i \langle \left[ \tilde{h}(t), \sigma_{0,i}^{\alpha} \right] \sigma_{0,j}^{\beta} \rangle ,
\tag{141}
$$

where

$$
T_{ij}^{\nu\gamma}(\alpha) \equiv \frac{1}{2s} J_{ij} \sum_{\mu} R_i^{x\mu} R_j^{x\nu} \varepsilon_{\mu\alpha\gamma} \qquad P_i^{\gamma}(\alpha) \equiv \left( h_i + \dot{\phi}_i \right) \sum_{\mu} R_i^{z\mu} \varepsilon_{\mu\alpha\gamma} + \dot{\theta}_i \varepsilon_{y\alpha\gamma} .
\tag{142}
$$

The right-hand side of (141) involves 3-point spin correlation functions. In order to obtain a closed system of equations we assume that the connected part of 3-point functions vanishes at all times, so that

$$
\begin{aligned}
\langle \sigma_{0,i}^{\alpha} \sigma_{0,j}^{\beta} \sigma_{0,\ell}^{\gamma} \rangle \rightarrow & \langle \sigma_{0,i}^{\alpha} \rangle \langle \sigma_{0,j}^{\beta} \sigma_{0,\ell}^{\gamma} \rangle_c + \langle \sigma_{0,j}^{\beta} \rangle \langle \sigma_{0,i}^{\alpha} \sigma_{0,\ell}^{\gamma} \rangle_c + \langle \sigma_{0\ell}^{\gamma} \rangle \langle \sigma_{0,i}^{\alpha} \sigma_{0,j}^{\beta} \rangle_c \\
& + \langle \sigma_{0,i}^{\alpha} \rangle \langle \sigma_{0,j}^{\beta} \rangle \langle \sigma_{0,\ell}^{\gamma} \rangle ,
\end{aligned}
\tag{143}
$$

where $\langle . \rangle_c$ denotes a connected correlator. We then impose $\langle \sigma_i^x \rangle = \langle \sigma_i^y \rangle = 0$ by appropriately choosing the rotation angles $\theta_i$ and $\phi_i$ and thus remain with a closed system of coupled ODEs for the variables $\theta_i$, $\phi_i$, $\langle \sigma_i^z \rangle$ and all the 2-point functions $\langle \sigma_i^{\alpha} \sigma_j^{\beta} \rangle$. The main advantage of truncating the BBGKY hierarchy in the rotating frame lies in the fact that the only non-vanishing 1-point function is $\langle \sigma_i^z \rangle$ and this greatly simplifies the decomposition on the right-hand side of (143). Furthermore, numerical tests using exact diagonalization in the fully connected TFIC suggest that connected correlation functions of the rotated-frame variables $\sigma_{t,i}^{\alpha}$ overall grow slower in time than the connected functions in the fixed-frame spin variables $S_i^{\alpha}$.

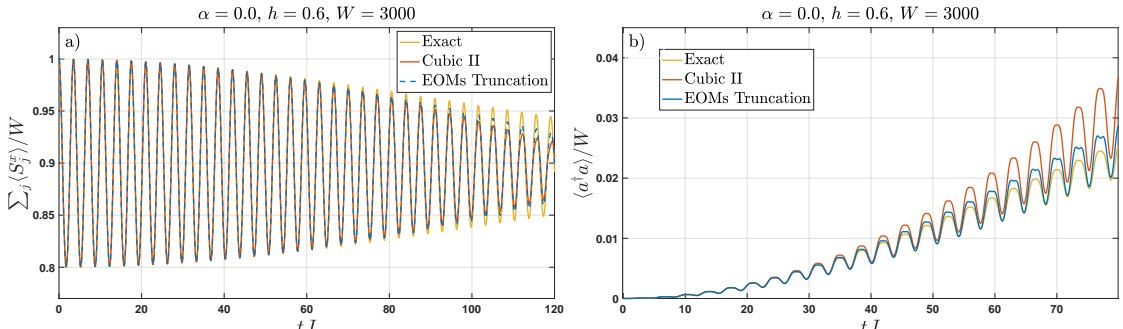

Figure 19: Fully connected TFIC time-evolution: comparison of exact results (Section 2.3.6) with the Cubic II (Section 2.3.6) and EOMs truncation in spin-space (this Appendix) approaches. a) Time evolution of $\langle \sum_j S_j^x \rangle / W$. b) Time evolution of $\langle a^\dagger a \rangle$.

In the specific case of the fully-connected TFIC the approximation just derived naturally acquires a control parameter[9], namely the total number of sites, due to the fact that in the thermodynamic limit the model results in a classical evolution whenever starting from a fully ordered state. In Fig. 19 we plot the result of this method applied to the case of the fully-connected TFIC with finite sizes. Comparison with the exact result and the Cubic II approach from Fig. 8 of Section 2.3.6 show that the EOMs truncation in spin-space yields results which are even slightly better than Cubic II.

## C  Derivation of the characteristic function formula

In this appendix we derive equation (12) for the characteristic function

$$\chi(\lambda) = \text{Tr}\left[\rho \exp(i\lambda R)\right] , \tag{144}$$

where $\rho$ is a Gaussian density matrix

$$\rho = \frac{1}{Z} \exp\left[\frac{1}{2} a^\dagger W a + w^\dagger \cdot a\right], \quad W = W^\dagger, \quad w^\dagger = w^T \Sigma^x . \tag{145}$$

Here the normalization $Z$ enforces $\text{Tr}[\rho] = 1$ and $W$ is restricted to be negative definite. The definition (7) of the vector $a$ always allows to choose $W$ such that

$$\Sigma^x W \Sigma^x = W^T . \tag{146}$$

The 1-point functions and the correlation matrix $\Delta$ in this Gaussian state are [69, 95]

$$\omega = \text{Tr}[\rho\, a] = -W^{-1} w ,$$

$$\Delta = \text{Tr}\left[\rho\,(a - \omega)(a^\dagger - \omega^\dagger)\right] - \frac{1}{2}\Sigma^z = -\frac{1}{2}\coth\left(\frac{1}{2}\Sigma^z W\right)\Sigma^z , \tag{147}$$

where $\Delta = \Delta^\dagger$ is positive definite. For the quadratic observable $R$ we consider the general class

$$R = \frac{1}{2} a^\dagger G a + g^\dagger \cdot a, \quad G = G^\dagger, \quad g^\dagger = g^T \Sigma^x, \quad \text{Det}(G) \neq 0 . \tag{148}$$

Similarly to (146) we choose without loss of generality $\Sigma^x G \Sigma^x = G^T$. We will consider the special case $\text{Det}(G) = 0$ at the end of this appendix.

---

[9]Even for small values of $s$.

We start by defining displacement operators $D_\beta$ associated with vectors of complex constants $\boldsymbol{\beta}$

$$D_\beta \equiv \exp\left[ \boldsymbol{a}^\dagger \Sigma^z \boldsymbol{\beta} \right], \qquad D_\beta^{-1} = D_\beta^\dagger \,, \tag{149}$$

which have the following properties

$$D_\beta^{-1} \boldsymbol{a}_i \, D_\beta = \boldsymbol{a}_i + \boldsymbol{\beta}_i \,, \quad D_\beta D_\tau = D_{\beta+\tau} \exp\left[ -\frac{1}{2} \boldsymbol{\beta}^\dagger \Sigma^z \boldsymbol{\tau} \right] . \tag{150}$$

The displacement operators are a complete orthogonal set on the space of bounded operators [69]. For operators $O$ such that $\mathrm{Tr}\left[ O D_\beta \right]$ is well defined this translates into

$$O = \frac{1}{\pi^\ell} \int d^2\boldsymbol{\beta} \, \mathrm{Tr}\left[ O D_\beta \right] D_\beta^\dagger \qquad \mathrm{Tr}\left[ D_\tau D_\beta \right] = \pi^\ell \delta^{2\ell} \left( \boldsymbol{\tau} + \boldsymbol{\beta} \right) , \tag{151}$$

where $d^2\boldsymbol{\beta} = \prod_{i=1}^\ell d\mathrm{Re}(\beta_i) \, d\mathrm{Im}(\beta_i)$. We can use this to express the characteristic function in the form

$$\chi(\lambda) = \left( \frac{1}{\pi^\ell} \right)^2 \int d^2\boldsymbol{\beta} \int d^2\boldsymbol{\tau} \, \mathrm{Tr}\left[ \rho \, D_\beta \right] \mathrm{Tr}\left[ \exp\left( i\lambda R \right) D_\tau \right] \mathrm{Tr}\left[ D_\beta^\dagger D_\tau^\dagger \right]$$

$$= \frac{1}{\pi^\ell} \int d^2\boldsymbol{\beta} \, \mathrm{Tr}\left[ \rho \, D_\beta \right] \mathrm{Tr}\left[ \exp\left( i\lambda R \right) D_{-\beta} \right] . \tag{152}$$

The first factor is known, see e.g. [69],

$$\mathrm{Tr}\left[ \rho \, D_\beta \right] = \exp\left[ -\frac{1}{2} \boldsymbol{\beta}^\dagger \Sigma^z \Delta \Sigma^z \boldsymbol{\beta} + \boldsymbol{\omega}^\dagger \Sigma^z \boldsymbol{\beta} \right], \tag{153}$$

where $\Delta$ and $\boldsymbol{\omega}$ are defined in (147). This reduces the problem to determining the second factor, which is of the form

$$f(\boldsymbol{\beta}, H, \boldsymbol{h}) = \mathrm{Tr}\left[ e^{i\left( \frac{1}{2} \boldsymbol{a}^\dagger H \boldsymbol{a} + \boldsymbol{h}^\dagger \cdot \boldsymbol{a} \right)} D_\beta \right], \tag{154}$$

where $H$, $\boldsymbol{h}$ have absorbed the $\lambda$ factor in front of $G$, $\boldsymbol{g}$. Shifting the boson operators by a constant $\boldsymbol{s} = -H^{-1}\boldsymbol{h}$ gives

$$f(\boldsymbol{\beta}, H, \boldsymbol{h}) = \exp\left( -i\frac{1}{2} \boldsymbol{h}^\dagger H^{-1} \boldsymbol{h} - \boldsymbol{h}^\dagger H^{-1} \Sigma^z \boldsymbol{\beta} \right) f(\boldsymbol{\beta}, H, 0) . \tag{155}$$

We can now use a splitting formula derived in Appendix D to write the second factor as

$$f(\boldsymbol{\beta}, H, 0) = \mathrm{Tr}\left[ e^{i\left( \frac{1}{2} \boldsymbol{a}^\dagger H \boldsymbol{a} + \boldsymbol{A}^\dagger \cdot \boldsymbol{a} \right)} \right] \exp\left( iB \right) ,$$

$$\boldsymbol{A}^\dagger = \left( -\boldsymbol{\beta}^\dagger \Sigma^z \right) \left[ \exp\left( i \Sigma^z H \right) - \mathbb{I} \right]^{-1} \left( \Sigma^z H \right) ,$$

$$B = \frac{1}{2} \left( \boldsymbol{A}^\dagger H^{-1} \right) \left[ H - \Sigma^z \sin\left( \Sigma^z H \right) \right] \left( H^{-1} \boldsymbol{A} \right) . \tag{156}$$

Noting that $\boldsymbol{A}$ retains the symmetry $\boldsymbol{A}^\dagger = \boldsymbol{A}^T \Sigma^x$ of $\boldsymbol{\beta}$ we can perform a final shift to a new set of canonical bosons $\boldsymbol{b}$

$$\mathrm{Tr}\left[ e^{i\left( \frac{1}{2} \boldsymbol{a}^\dagger H \boldsymbol{a} + \boldsymbol{A}^\dagger \cdot \boldsymbol{a} \right)} \right] = e^{-\frac{i}{2} \boldsymbol{A}^\dagger H^{-1} \boldsymbol{A}} \, \mathrm{Tr}\left[ e^{\frac{i}{2} \boldsymbol{b}^\dagger H \boldsymbol{b}} \right] . \tag{157}$$

Putting everything together we obtain

$$f(\boldsymbol{\beta}, H, \boldsymbol{h}) = Z_G \exp\left[ -\frac{1}{2} \boldsymbol{\beta}^\dagger \Sigma^z \Delta_G \Sigma^z \boldsymbol{\beta} + \boldsymbol{\omega}_G^\dagger \Sigma^z \boldsymbol{\beta} \right], \tag{158}$$

where

$$\boldsymbol{\omega}_G = -G^{-1}\boldsymbol{g} \,, \qquad \Delta_G(\lambda) = -\frac{1}{2}\coth\left(i\lambda\frac{1}{2}\Sigma^z G\right)\Sigma^z \,,$$

$$Z_G(\lambda) = \exp\left(-i\lambda\frac{1}{2}\boldsymbol{g}^\dagger G^{-1}\boldsymbol{g}\right)\mathrm{Tr}\left[e^{i\lambda\frac{1}{2}\boldsymbol{b}^\dagger G\boldsymbol{b}}\right] \,. \tag{159}$$

The first complication in evaluating the trace in $Z_G(\lambda)$ is that in general $\boldsymbol{b}^\dagger G\boldsymbol{b}$ cannot be diagonalized by a transformation to canonical bosons. In Appendix D we review how to diagonalize these generic quadratic forms by introducing a set of "almost canonical" bosons, which provide us with a generic formula to calculate the trace needed. The second complication is that the trace in $Z_G(\lambda)$ is in general not well defined. However, we know from Section 1.1 that $\chi(\lambda)$ is bounded and well-defined for any $R$, given the presence of the physical density matrix $\rho$. This allows one to introduce an infinitesimal regularization when evaluating $Z_G(\lambda)$. The result follows directly from the trace formula of Appendix D and is given by

$$Z_G(\lambda) = \exp\left(-i\lambda\frac{1}{2}\boldsymbol{g}^\dagger G^{-1}\boldsymbol{g}\right)\mathrm{Det}\left[2\Sigma^z\sinh\left(-i\lambda\frac{1}{2}\Sigma^z G\right)\right]^{-1/2} \,. \tag{160}$$

Substituting (153) and the expression (158) for $f(-\boldsymbol{\beta}, \lambda G, \lambda\boldsymbol{g})$ back into (152) we obtain

$$\chi(\lambda) = Z_G\frac{1}{\pi\ell}\int d^2\boldsymbol{\beta}\,\exp\left[-\frac{1}{2}\boldsymbol{\beta}^\dagger\Sigma^z(\Delta+\Delta_G)\Sigma^z\boldsymbol{\beta} + \left(\boldsymbol{\omega}^\dagger - \boldsymbol{\omega}_G^\dagger\right)\Sigma^z\boldsymbol{\beta}\right] =$$

$$= \frac{Z_G}{\sqrt{\mathrm{Det}(\Delta+\Delta_G)}}\exp\left[-\frac{1}{2}\left(\boldsymbol{\omega}^\dagger - \boldsymbol{\omega}_G^\dagger\right)\left(\Delta+\Delta_G\right)^{-1}\left(\boldsymbol{\omega} - \boldsymbol{\omega}_G\right)\right] \,, \tag{161}$$

which is the result (12) stated in the main text. The last equality follows from standard Gaussian integrals [69] when we transform the integration from complex to real variables **z** by

$$\boldsymbol{\beta} = S\,\mathbf{z} \qquad S \equiv \frac{1}{\sqrt{2}}\begin{pmatrix} \mathbb{I}_{\ell\times\ell} & i\,\mathbb{I}_{\ell\times\ell} \\ \mathbb{I}_{\ell\times\ell} & -i\,\mathbb{I}_{\ell\times\ell} \end{pmatrix} = \left(S^\dagger\right)^{-1} \,. \tag{162}$$

The convergence of the integral for any choice of the Hermitian matrix $G$ in (148) is a consequence of the fact that $S^\dagger(\Delta+\Delta_G)S$ is a complex symmetric matrix with positive-definite real part[10].

So far we have assumed that $G$ is invertible. If $G$ is singular we introduce a regulator $\varepsilon$ such that $\det(G_\varepsilon) \neq 0$. We then replace

$$\left(\Delta+\Delta_{G_\varepsilon}\right)^{-1} = -i\lambda\,G_\varepsilon + G_\varepsilon\,M_{G_\varepsilon}\,G_\varepsilon \,, \tag{163}$$

where $M_{G_\varepsilon}$ is non-singular in the limit $\varepsilon \to 0$. Given that

$$\Sigma^z\sinh\left(-i\lambda\frac{1}{2}\Sigma^z G_\varepsilon\right) = G_\varepsilon\,L_{G_\varepsilon} \qquad L_{G_\varepsilon} \text{ non-singular for } \varepsilon \to 0 \,, \tag{164}$$

one can easily verify that all terms formally ill-defined in (161) for $\varepsilon \to 0$ exactly cancel.

Some comments are in order:

1. Equation (161) generalizes the analogous formula for the trace of the product of two physical Gaussian density matrices $\rho_1$ and $\rho_2$ specified by Hermitian negative-definite matrices [69, 97–99].

---

[10]The positive-definiteness of the real part ensures convergence and the formula can be proved for complex matrices by analytic continuation from the real matrices formula [96]

2. A result similar to (161) has been derived in [100] for the specific case $R = \sum_i a_i^\dagger a_i$, by employing coherent states to replace the evaluation of the trace in (144) with two Gaussian integrals. In order to handle the general case with the method of [100] one needs to first work out the decomposition

$$\exp(i\lambda R) = \exp(a_i^\dagger A_{ij} a_j^\dagger) \exp(a_i^\dagger B_{ij} a_j) \exp(a_i C_{ij} a_j) \exp(D_i a_i + \tilde{D}_j a_j^\dagger). \tag{165}$$

This can be done with the algebraic methods presented in Appendix D.

3. A result similar to (161) in the case of specific observables relevant for the Gaussian boson sampling problem has been derived in [101].

# D   Diagonalisation of quadratic forms of bosons and splitting formulas

We recall the notations

$$\Sigma^x = \begin{pmatrix} 0 & \mathbb{I}_{\ell\times\ell} \\ \mathbb{I}_{\ell\times\ell} & 0 \end{pmatrix}, \qquad \Sigma^y = i \begin{pmatrix} 0 & -\mathbb{I}_{\ell\times\ell} \\ \mathbb{I}_{\ell\times\ell} & 0 \end{pmatrix}, \qquad \Sigma^z = \begin{pmatrix} \mathbb{I}_{\ell\times\ell} & 0 \\ 0 & -\mathbb{I}_{\ell\times\ell} \end{pmatrix}. \tag{166}$$

Arranging the boson operators in a vector $\boldsymbol{a}^\dagger = \{a_1^\dagger, \ldots, a_\ell^\dagger, a_1, \ldots, a_\ell\}$ we have

$$\left[ a_i, a_j^\dagger \right] = \Sigma^z_{i,j}. \tag{167}$$

Consider now a quadratic form

$$\boldsymbol{a}^\dagger W \boldsymbol{a}, \qquad \Sigma^x W^T \Sigma^x = W, \qquad \Sigma^z W \text{ diagonalizable}, \tag{168}$$

where the second identity can be enforced without loss of generality due to the definition of $\boldsymbol{a}$. As the bilinear form (168) generally cannot be diagonalized by a canonical transformation we follow [102] and consider left and right eigenvectors of $\Sigma^z W$ such that

$$\Sigma^z W \boldsymbol{R}_a = E_a \boldsymbol{R}_a, \qquad \boldsymbol{L}_a^\dagger \Sigma^z W = E_a \boldsymbol{L}_a^\dagger, \qquad \boldsymbol{L}_a^\dagger \cdot \boldsymbol{R}_b = \delta_{a,b}, \qquad \Sigma^z W = \sum_{a=1}^{2\ell} E_a \boldsymbol{R}_a \boldsymbol{L}_a^\dagger. \tag{169}$$

By virtue of the second identity in (168) the complex eigenvalues come in pairs $(E_a, -E_a)$ and we can choose without loss of generality

$$E_{\ell+j} = -E_j, \qquad D \equiv \mathrm{diag}(E_1, \ldots, E_\ell, -E_1, \ldots, -E_\ell). \tag{170}$$

We now define operators

$$\boldsymbol{\gamma}_j = (\boldsymbol{L}_j^\dagger)_k \, \boldsymbol{a}_k, \qquad \overline{\boldsymbol{\gamma}}_j = \boldsymbol{a}_n^\dagger \Sigma^z_{n,p} (\boldsymbol{R}_k)_p \Sigma^z_{k,j}, \tag{171}$$

where the arrangement of eigenvalues in (170) implies

$$\overline{\boldsymbol{\gamma}}_j = \boldsymbol{\gamma}_{j+\ell} := \overline{\boldsymbol{\gamma}}_j, \qquad \overline{\boldsymbol{\gamma}}_{j+\ell} = \boldsymbol{\gamma}_j := \gamma_j, \qquad j = 1, \ldots, \ell. \tag{172}$$

It is easy to check that they fulfil bosonic commutation relations

$$\left[ \boldsymbol{\gamma}_i, \overline{\boldsymbol{\gamma}}_j \right] = \Sigma^z_{i,j}, \tag{173}$$

and that they bring (168) to a very convenient normal form

$$a^\dagger W a = \overline{\gamma} D \Sigma^z \gamma = 2 \sum_{i=1}^{\ell} E_i \left( \overline{\gamma}_i \gamma_i + \frac{1}{2} \right) . \tag{174}$$

If $W = W^\dagger$ and all eigenvalues of $\Sigma^z W$ are real one can choose $\gamma_i^\dagger = \overline{\gamma}_i$[11] [102]. A sufficient condition for this is requiring a positive or negative definite $W$ [103]. The transformation (171) can alternatively be expressed in the form

$$F a_j F^{-1} = \gamma_j , \qquad F = e^{\frac{1}{2} a^\dagger A a} \qquad \left( e^{-\Sigma^z A} \right)_{j,k} = (L_j^\dagger)_k . \tag{175}$$

Given the bosonic Fock states

$$|n\rangle = \frac{1}{\sqrt{n_1! n_2! \dots n_\ell!}} (a_1^\dagger)^{n_1} \dots (a_\ell^\dagger)^{n_\ell} |0\rangle , \tag{176}$$

we now construct left and right Fock states as follows

$$|n\rangle_R \equiv F |n\rangle = \frac{1}{\sqrt{n_1! n_2! \dots n_\ell!}} (\overline{\gamma}_1)^{n_1} \dots (\overline{\gamma}_\ell)^{n_\ell} |0\rangle_R ,$$

$$_L\langle n| \equiv \langle n| F^{-1} = {}_L\langle 0| (\gamma_1)^{n_1} \dots (\gamma_\ell)^{n_\ell} \frac{1}{\sqrt{n_1! n_2! \dots n_\ell!}}. \tag{177}$$

Here the left and right vacua respectively fulfil

$$\gamma_j |0\rangle_R = 0 = {}_L\langle 0| \overline{\gamma}_j , \quad j = 1, \dots, \ell , \tag{178}$$

and by construction we have

$$_L\langle n| m\rangle_R = \prod_{j=1}^{\ell} \delta_{n_j, m_j} , \qquad \overline{\gamma}_m \gamma_m |n\rangle_R = n_m |n\rangle_R . \tag{179}$$

The left and right Fock states can be used to compute traces as follows

$$\mathrm{Tr}[O] = \mathrm{Tr}[F^{-1} O F] = \sum_n \langle n| F^{-1} O F |n\rangle = \sum_n {}_L\langle n| O |n\rangle_R . \tag{180}$$

Combining (174) with (180) and (179) we have

$$\mathrm{Tr}\left[ \exp\left( \frac{1}{2} a^\dagger W a \right) \right] = \sum_n {}_L\langle n| \exp\left[ \sum_{i=1}^{\ell} E_i \left( \overline{\gamma}_i \gamma_i + \frac{1}{2} \right) \right] |n\rangle_R = \sum_n \exp\left[ \sum_{i=1}^{\ell} E_i \left( n_i + \frac{1}{2} \right) \right]. \tag{181}$$

Assuming $\mathrm{Re}(E_i) < 0 \ \forall \, i$ the sum over the $n_i$'s converges to

$$\mathrm{Tr}\left[ \exp\left( \frac{1}{2} a^\dagger W a \right) \right] = \prod_{i=1}^{\ell} \frac{1}{e^{-E_i/2} - e^{E_i/2}} = \mathrm{Det}\left[ \Sigma^z \left( e^{-\frac{1}{2} \Sigma^z W} - e^{\frac{1}{2} \Sigma^z W} \right) \right]^{-1/2} . \tag{182}$$

In the above discussion we have implicitly assumed that the states $|0\rangle_R$ and $|0\rangle_L$ are well-defined in the sense that they have a regular expansion in the original basis states $|n\rangle$. We have

$$|0\rangle_R = F |0\rangle = C_R \exp\left( M_{ij}^R a_i^\dagger a_j^\dagger \right) |0\rangle \qquad |0\rangle_L = \left( F^{-1} \right)^\dagger |0\rangle = C_L \exp\left( M_{ij}^L a_i^\dagger a_j^\dagger \right) |0\rangle , \tag{183}$$

---

[11]To achieve this the additional freedom in choosing the overall sign within a pair $\pm(|E_a|, -|E_a|)$ must be used.

where $M^R$ and $M^L$ are matrices derived from the $A$ in (175) and $C^R, C^L \in \mathbb{C}$. These states are well-defined and normalizable only if all singular values $\sigma_i^R$, $\sigma_i^L$ of $M^R$ and $M^L$ are bounded by $\sigma_i^R < 1$, $\sigma_i^L < 1$ [104]. We note that in many cases these inequalities can be satisfied by using the freedom in the arrangement of the eigenvalues in (170), namely the signs in the definitions of the pairs $(\pm |E_a|, \mp |E_a|)$. The trace formula (181) is only valid when a choice of the pairs signs exists such to satisfy $\sigma_i^{R,L} < 1 \ \forall \ i = 1, \ldots, \ell$.

The diagonalization just introduced allows us to easily prove a splitting formula which is used in Appendix C. Given a matrix $H$ that satisfies[12] $H = \Sigma^x H^T \Sigma^x$ and a generic vector $\boldsymbol{h}$ the following decompositions hold

$$\exp\left[\frac{1}{2}\boldsymbol{a}^\dagger H \boldsymbol{a} + \boldsymbol{h}^\dagger \cdot \boldsymbol{a}\right] = \exp[C_L]\exp\left[\boldsymbol{h}_L^\dagger \cdot \boldsymbol{a}\right]\exp\left[\frac{1}{2}\boldsymbol{a}^\dagger H \boldsymbol{a}\right]$$

$$= \exp\left[\frac{1}{2}\boldsymbol{a}^\dagger H \boldsymbol{a}\right]\exp\left[\boldsymbol{h}_R^\dagger \cdot \boldsymbol{a}\right]\exp[C_R] , \qquad (184)$$

where

$$\boldsymbol{h}_{R,L}^\dagger = \pm \boldsymbol{h}^\dagger (\Sigma^z H)^{-1}\left[\exp(\pm\Sigma^z H) - \mathbb{I}\right] ,$$

$$C_R = C_L = -\frac{1}{2}\boldsymbol{h}^\dagger H^{-1}\left(H - \Sigma^z \sinh(\Sigma^z H)\right)\left(H^{-1}\Sigma^x (\boldsymbol{h}^\dagger)^T\right) . \qquad (185)$$

To prove this we start by diagonalizing the quadratic form that appears on the right-hand side of (184) by the transformation to $\gamma, \overline{\gamma}$ operators. By calling $Q$ the matrix whose columns are the right-eigenvectors $\boldsymbol{R}_a$ of (169) we obtain

$$\exp\left[\frac{1}{2}\boldsymbol{a}^\dagger H \boldsymbol{a}\right]\exp\left[\boldsymbol{h}_R^\dagger \cdot \boldsymbol{a}\right] = \exp\left[\frac{1}{2}\overline{\gamma}\mathcal{E}\gamma\right]\exp\left[\boldsymbol{h}_R^\dagger Q \gamma\right]$$

$$= \prod_{i=1}^{\ell}\exp\left(\frac{E_i}{2}\right)\exp\left(E_i\overline{\gamma}_i\gamma_i\right)\exp\left(g_i\gamma_i + f_i\overline{\gamma}_i\right), \qquad (186)$$

where $E_i = \mathcal{E}_{ii} = \mathcal{E}_{i+\ell,i+\ell} = E_{i+\ell}$, $g_i \equiv (\boldsymbol{h}_R^\dagger Q)_i$, $f_i \equiv (\boldsymbol{h}_R^\dagger Q)_{(i+\ell)}$ for $i = 1, \ldots, \ell$. We now aim to apply the Baker-Campbell-Hausdorff (BCH) formula to the individual factors on the r.h.s. in (186). We do this by employing a faithful $3 \times 3$ matrix representation of the bosonic algebra generated by $\gamma_i$ and $\overline{\gamma}_i$ [105]. Denoting by $e_{ij}$ the $3 \times 3$ matrix with entries $e_{ij}^{\alpha\beta} = \delta_{i,\alpha}\delta_{j,\beta}$ we use the identifications

$$\overline{\gamma}_i\gamma_i \rightarrow e_{22} \qquad \overline{\gamma}_i \rightarrow e_{23} \qquad \gamma_i \rightarrow e_{12} \qquad \mathbb{1} \rightarrow e_{13} . \qquad (187)$$

Using the explicit matrix representation and the BCH formula we find

$$\exp[E_i e_{22}]\exp[g_i e_{12} + f_i e_{23}] = \begin{pmatrix} 1 & g_i & 1/2\, f_i\, g_i \\ 0 & \exp E_i & f_i \exp E_i \\ 0 & 0 & 1 \end{pmatrix} =$$

$$= \exp\left[E_i e_{22} + \frac{g_i E_i}{\exp(E_i) - 1}e_{12} - \frac{f_i E_i}{\exp(-E_i) - 1}e_{23} + \frac{1}{4}f_i g_i \frac{E_i - \sinh(E_i)}{\sinh(E_i/2)^2}e_{13}\right] . \qquad (188)$$

Substituting the $e_{ij}$ for $\gamma$ and $\overline{\gamma}$ and using $Q(\Sigma^z \mathcal{E})Q^{-1} = \Sigma^z H$ we obtain (185).

Another splitting formula concerns purely quadratic forms. Given two matrices $W_i \ i = 1, 2$ respecting $W_i = \Sigma^x W_i^T \Sigma^x$, the following identity holds

$$\exp\left[\frac{1}{2}\boldsymbol{a}^\dagger W_3 \boldsymbol{a}\right] \equiv \exp\left[\frac{1}{2}\boldsymbol{a}^\dagger W_1 \boldsymbol{a}\right]\exp\left[\frac{1}{2}\boldsymbol{a}^\dagger W_2 \boldsymbol{a}\right] \qquad (189)$$

---

[12]Without loss of generality, see Appendix C.

$$\exp\left[\Sigma^z W_3\right] = \exp\left[\Sigma^z W_1\right]\exp\left[\Sigma^z W_2\right] . \tag{190}$$

Here also $W_3$ respects the property $W_3 = \Sigma^x W_3^T \Sigma^x$. To prove this, we use the matrix $S$ of (162) to pass to vectors of real bosons $\mathbf{z} = S^\dagger \mathbf{a}$ with commutation relations $\left[\mathbf{z}, \mathbf{z}^T\right] = -\Sigma^y$ and notice that the BCH structure of nested commutators appearing implicitly in (189) at the bosonic operators level is automatically transferred to the matrix level.

We remark that the two splitting formulas just discussed and the trace formula (182) are sufficient to derive in a purely algebraic way our characteristic function formula (12), as opposed to the method of Appendix C. This follows in the obvious way by identifying $W_1$ and $W_2$ respectively with $W$ and $i\lambda G$ of Appendix C, while the first splitting formula takes care of the linear parts.

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
