# Peer review of "Out-of-equilibrium full-counting statistics in Gaussian theories of quantum magnets"

_SciPost Physics_

## Round 1 · Referee Report · Anonymous (Referee 1) · 2024-5-19

Strengths

  1. this topic of this work is timely
  2. the results of this work is useful to the study of this field
  3. the paper is well organized and clearly written

Report

This paper by Senese et. al. reports a simple formula to calculate the full counting statistics of Gaussian states (i.e. probability distribution of the measurement outcomes of observables in subsystems) from the two-body correlators. Based on such a formula, the authors leverage the well-established spin wave theory to calculate the full counting statistics in various spin systems. As the starting point for this study is the Gaussian states, it is expected (at least not surprising) that the full counting statistics can be expressed as a quadratic form of bosonic variables. Despite that, full counting statistics is a timely topic recently. Figuring out the exact expression, as well as several examples to demonstrate the power of this expression, is important to this field. Therefore, I find this work, in principle, suitable to be published in SciPost.

Nevertheless, I would like to invite the authors to address the following point before the paper gets published.
How well the quantum system are described by a Gaussian state basically determines the applicability of the formula proposed by the authors. Indeed, in a few cases, especially in the scenarios of quench dynamics, the authors found large discrepancy between the results calculated with their formula and the strict results. While several of these examples are helpful, it is better if the authors can provide some more general principles and guiding rules for when the formula works the best or fails.

Another related point is that the authors mainly focus on the examples when the quantum systems can thermalize or is already at thermal equilibrium. Recently, full counting statistics in integrable systems also receive quite a lot of interest. I also recommend the authors to comment and discuss the validity of their results in those integrable systems which do not fully thermalize to a Gibbs ensemble.

Recommendation

Ask for minor revision

  • validity: high
  • significance: good
  • originality: good
  • clarity: high
  • formatting: excellent
  • grammar: excellent

Author:  Riccardo Senese  on 2024-10-18  [id 4873]

(in reply to Report 1 on 2024-05-19)
Category:
answer to question

We reply to all the points raised by the referee in the PDF attached.

Attachment:

Reply_to_referees_4rl9pbs.pdf

---

## Round 1 · Referee Report · Anonymous (Referee 2) · 2024-6-15

Strengths

1- detailed comparison of mean-field full counting statistics with exact results 2-comparison of different mean-field approaches for bosons 3-derivation of analytical formula for FCS of quadratic observables for bosons 4-spin-wave theory around a time-dependent direction

Weaknesses

1-The paper lacks a bit of structure as derivations and results as well as equilibrium and non-equilibrium results are mixed.
2-Extreme value statistics of staggered magnetization in 2d/3d Heisenberg model lacks explanation. This could be an artifact.

Report

In their manuscript "Out-of Equilibrium Full-Counting Statistics in Gaussian Theories of Quantum Magnets" the authors derive an analytical formula for the generating function of quadratic operators in Gaussian theories of bosons and apply it to different mean-field theories of quantum magnets in equilibrium and after a quantum quench. The work provides detailed derivations and compares the accuracy of the full counting statistics of the mean-field theory with other published methods, e.g. Monte Carlo results for the 2d Heisenberg model. The relative merits of different mean-field approximations (Schwinger-Boson mean-field theory, modified spin-wave theory) are assessed. The authors focus on the 2d and 3d Heisenberg model and the long-range transverse-field Ising model. The validity of calculating second and fourth cumulants in mean-field theory and generating the FCS from this is investigated.

The phenomenological fit of the PDF for the 2d and 3d Heisenberg model to an extreme-value distribution is an interesting observation, which calls for an explanation. The fact that for both 2d and 3d Heisenberg model the PDF of the staggered magnetization follows a Gumbel distribution is rather suspicious, since the former model has no long-range order at finite temperature while the latter has. Why is the PDF in Fig. 1 at zero temperature not a semicircular distribution like in Fig. 3 of the supplemental material of Ref. [20] ? Is this an artifact of the spin-wave expansion around an ordered state with well-defined orientation of the spins ? The PDF for the 2d Heisenberg antiferromagnet in section 3 appears to have the correct Gaussian shape, but this is the same model as in Fig. 1, only the mathematical treatment is different.

The long-range transverse-field Ising chain has a finite-temperature Kosterlitz-Thouless floating phase at alpha=2 (See e.g. PRB 64, 184106 (2001); Journal of Computational Physics, vol. 228, 7 (2009) and J. Stat. Mech. (2020) 063105).
Here, indeed one might expect non-Gaussian magnetization fluctuations inside a relatively narrow temperature window. This is definitely the case for the related 2d XY model, see Bramwell et al. Nature 396, 552 (1998). An interesting question is whether the authors' spin-wave approximation can reproduce this regime.

In the section on quantum quenches (before Eq. 32) it is not stated what the Hamiltonian after the quench is.

In my view, the paper is suitable for publication in Scipost Physics provided the authors can give a more in-depth explanation for the occurrence of the extreme-value statistics in the linear spin-wave calculation for the 2d and 3d Heisenberg model in Figs. 1 and 2.

The detailed benchmarks with exact results may be very useful to assess the validity of FCS of mean-field theories in regimes where no other methods are available, especially non-equilibrium settings, opening a new pathway for interpretation of numerous experimental results.

Requested changes

1-explain occurrence of the Gumbel distribution in Fig. 1

Recommendation

Ask for minor revision

  • validity: high
  • significance: high
  • originality: high
  • clarity: high
  • formatting: perfect
  • grammar: perfect

Author:  Riccardo Senese  on 2024-10-18  [id 4872]

(in reply to Report 2 on 2024-06-15)
Category:
answer to question

We reply to all the points raised by the referee in the PDF attached.

Attachment:

Reply_to_referees.pdf

---

## Editorial Decision

resubmitted